# MODEL MERGING SCALING LAWS IN LARGE LANGUAGE MODELS

## ABSTRACT

We study empirical scaling laws for language model merging measured by cross-entropy. Despite its wide practical use, merging lacks a quantitative rule that predicts returns as we add experts or scale the model size. We identify a compact power law that links model size and expert number: the size-dependent floor decreases with model capacity, while the merging tail exhibits clear diminishing returns in the number of experts. The law holds in-domain and cross-domain, tightly fits measured curves across diverse architectures and methods (Average, TA, TIES, DARE), and explains two robust regularities: most gains arrive early, and variability shrinks as more experts are included. Building on this, we present a simple theory that explains why gains fall roughly as $1/k$ and links the floor and tail to properties of the base model and the diversity across domains. This law enables *predictive planning*: estimate how many experts are needed to reach a target loss, decide when to stop adding experts, and trade off scaling the base model versus adding experts under a fixed budget—turning merging from heuristic practice into a computationally efficient, planable alternative to multitask training. This suggests a scaling principle for distributed generative AI: predictable gains can be achieved by composing specialists, offering a complementary path toward AGI-level systems.

## 1 INTRODUCTION

Large language models (LLMs) are often specialized by fine-tuning on different domains, producing multiple domain experts. Model merging combines these experts in weight space to synthesize a single model without retraining. This idea underlies a range of methods: linear rules such as weight averaging (Izmailov et al., 2018; Wortsman et al., 2022), task arithmetic (Ilharco et al.), selective or nonlinear schemes like TIES (Yadav et al., 2023), and DARE (Yu et al., 2024). Merging has proven attractive in practice—it can approximate joint training at a fraction of the cost, supports modular pipelines with adapters, e.g., LoRA (Hu et al.; Mao et al., 2025), and enables composition under privacy or compute constraints.

Despite this promise, merging remains largely empirical. Practitioners experiment with subsets, orders, and normalization rules, often at substantial computational expense. Unlike pretraining, where well-established *scaling laws* guide how loss decreases with model size, data, or compute (Kaplan et al., 2020; Hoffmann et al., 2022), merging lacks an analogous quantitative account. This gap makes it difficult to anticipate convergence as more experts are added, to compare rules across base sizes, or to make budget-aware design choices.

In this paper, we first introduce a compact, predictive *merging scaling law* that couples model size $N$ with the number of merged experts $k$:

$$\mathbb{E}[L \mid N, k] = \underbrace{L_* + B\,N^{-\beta}}_{\text{floor } L_\infty(N)} + \underbrace{\frac{A_0\,N^{-\gamma}}{k + b}}_{\text{merging tail}}, \qquad \beta, \gamma > 0,\ b \geq 0. \tag{1}$$

Intuitively, larger base models depress the size-dependent floor $L_\infty(N)$ and shrink the tail amplitude $A_0 N^{-\gamma}$; adding experts yields steep early improvements that taper as $1/(k+b)$. The term $L_*$ denotes the irreducible floor that remains even for very large $N$.

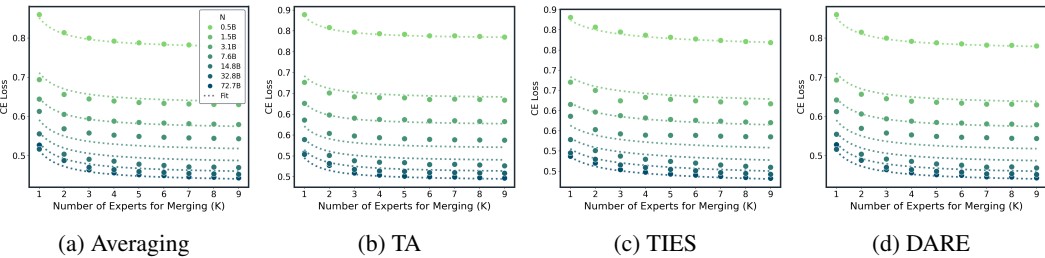

(a) Averaging        (b) TA        (c) TIES        (d) DARE

Figure 1: Model Merging Scaling Law. CE vs. number of merged experts ($k$) at multiple model sizes ($N$) for four merging methods. Dots are real measurements; dotted lines are fits to the unified law $L_\infty(N) + A(N)/(k+b)$. Across methods we see the same pattern: steep early gains that flatten into a $1/(k+b)$ tail, and a uniform downward shift with larger $N$ (both the floor and the tail shrink). Method differences get smaller and smaller as scaling up. $R^2 > 0.98$ over all fitted points

As shown in Fig. 1 and Fig. 2, our experiments on **10,886** merged models, base sizes from 0.5 to 72B, nine domains, four methods: Average/Task Arithmetic (TA)/TIES/DARE, validate this power law and directly compare *merging* with *multitask SFT* under normalized loss and GPU-hours. Empirically, merging approaches multitask SFT performance while using negligible GPU-hours, and method gaps compress as $k$ and $N$ grow. These findings position merging as a practical, budget-friendly alternative to comprehensive multitask training and highlight the significance of the proposed merging scaling law for forecasting returns and planning budgets.

This study reveals a consistent power law for LLM merging that aligns with the later sections: (i) *larger models are easier to merge*, floors decrease with $N$ and tails shrink (Fig. 5); (ii) *most gains arrive early*, with a clear elbow at small $k$ (Sec. 3.3.3); (iii) *mixing domains helps pooled generalization* under the same floor+tail scaling (Sec. 3.3.2); (iv) *method differences are small at scale*, with both means and variability converging (Sec. 3.3.4); (v) *order sensitivity fades quickly* as $k$ grows (Sec. 4.3); and (vi) *the law transfers across backbones* with the same diminishing-returns shape (Sec. 4.4).

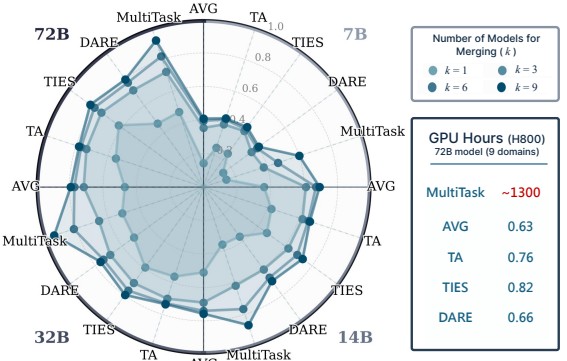

Figure 2: Overview of Merging vs MultiTask. The the polar axis represents the normalized negative loss.

In summary, this work provides:

(1) **Unified scaling law:** We introduce a compact floor+tail law that links base size and expert count, and show it applies consistently in both in-domain and cross-domain settings.

(2) **Large-scale validation:** Across extensive experiments covering diverse domains, model sizes, 10,886 models, and merging methods, the law tightly fits measured curves, variance contracts with more experts, and method gaps compress as scale increases.

(3) **Theory:** We derive a universal inverse-$k$ tail and variance under equal-normalized composition, and connect the floor and tail terms to base curvature and cross-domain dispersion.

(4) **Operational recipe:** A lightweight three-point fit predicts the full curve and recommends an efficient expert count, converting trial-and-error into budgetable planning; results are robust to candidate-pool size and transfer across architectures.

## 2   BACKGROUND, RELATED WORK, AND SETUP

**Notation.** Let $N$ denote the size of the base model, $\mathcal{M}$ denotes a set of expert models, and let $k$ be the number of expert models to be merged. We denote the base model by $\theta_0$. A task vector $v$ is defined as the parameter difference between the base model and a domain-adapted model, which may be either the full parameter difference or a low-rank adaptation such as an adapter/LoRA module (Hu

et al., 2022) restricted to its subspace. Unless otherwise stated, we employ *equal-weight merging*, where all task vectors are assigned the same importance. For fixed $N$ and $k$, the *expected loss* refers to the average performance over all possible $k$-element subsets of experts drawn from $\mathcal{M}$, while *variance* measures the variability of the loss.

## 2.1 BACKGROUND

**Model Merging**    Model merging is the integration of multiple independently trained models into a single cohesive model by aggregating their parameters (Matena & Raffel, 2022; Jin et al., 2022; Wang et al., 2025a). Existing work performs merging either (i) within *modular subspaces*, most commonly adapters or LoRA (Hu et al., 2022), enabling plug-and-play composition

Table 1: Unified view of model merging recipes

| Method | $\Psi(v)$ | $c$ | $\alpha$ | Add. Params |
|---|---|---|---|---|
| Average | $v$ | 1 | $\alpha = 1/k$ | - |
| TA | $v$ | 0.8 | $\alpha = 1/k$ | - |
| TIES | Trim, Elect, Disjoint | 1 | $\alpha = 1/k$ | $d = 1.0$ |
| DARE | $m \odot v/(1-p)$ | 1 | $\alpha = 1/k$ | $p = 0.2$ |

across domains with minimal interference (Hu et al.; Mao et al., 2025) , or (ii) on the *full parameter space*, e.g., model soups and Fisher/weight-space averaging (Izmailov et al., 2018; Wortsman et al., 2022; Davari & Belilovsky, 2024). Recent research has largely concentrated on the full parameter space, where task-vector methods are particularly dominant. The core idea in these approaches is to manipulate *task vectors*—changes relative to the base pre-trained model (Rinaldi et al., 2025; Zhang et al., 2024; Bowen et al., 2024). These methods incur almost no additional computational overhead, while enabling efficient model fusion across multiple tasks. Further gains come from processing task vectors before aggregation, for instance using element-wise masks or gates, like TIES or DARE, to reduce conflicts between experts (Yadav et al., 2023; Yu et al., 2024; Lu et al., 2024), which substantially mitigates conflicts between models. In addition, Ainsworth et al. (2022); Entezari et al. (2021); Stoica et al. (2023) address feature mismatches across models based on observation of permutation invariance. Furthermore, merging methods have been refined with learning-based techniques (Jhunjhunwala et al., 2024; Yan et al., 2025; Akiba et al., 2025; Yang et al., 2023; Cheng et al., 2025; Wang et al., 2024), which expand the hyperparameter search space for merging and yield consistently stronger performance. These approaches cover most practical pipelines and define the primary settings evaluated in this paper. However, most of studies consider limited expert models to merge, and the relation between the number of experts and the effectiveness is underexplored. Wang et al. (2025c); Yadav et al. (2024) examined this relationship from theoretical and empirical perspectives, respectively, identifying factors that influence merging performance, but did not provide a systematic scaling law to guide merging across different domains and model sizes.

**Scaling Law**    Classical scaling laws quantify how loss scales with *model size*, *data*, and *compute*: parameter/data power laws and compute-optimal trade-offs (Kaplan et al., 2020; Hoffmann et al., 2022; Hestness et al., 2017). Extensions study transfer and evaluation efficiency, as well as precision/quantization scaling that augments the usual size–data laws with a precision term (Kumar et al.). Scaling laws provide a predictable, quantitative framework that helps researchers make more informed decisions and prevent the blind allocation of vast resources (Ardalani et al., 2022; Klug & Heckel, 2022; Neumann & Gros, 2022; Geiping et al., 2022). Specifically, scaling laws have been leveraged by Filipovich et al. (2022) to empirically demonstrate that Direct Feedback Alignment (DFA) is not a more compute-efficient training method than backpropagation. Hilton et al. (2023) extend these laws by incorporating sparsity, finding a compute-optimal sparse-dense trade-off that challenges the conventional belief that dense models are always superior for large-scale training. Fernandes et al. (2023) research on scaling laws to multilingual neural machine translation models, revealing that data mixture weights affect the multiplicative factor of the scaling law but not the scaling exponent. These laws guide pretraining, but they do not address *composition in weight space*.

## 2.2 SETUP

**Expert Models**    We use a dual–track design to balance control and realism (details in App. C). *(i) Controlled experts:* Starting from the same base, we train nine domain experts with identical hyperparameters. All base models are from the Qwen2.5 series (0.5B–72B) (Qwen et al., 2025). *(ii) Open-source experts:* We additionally treat diverse HuggingFace checkpoints as experts to test robustness under heterogeneous, partly opaque post-training.

**Data**  We construct our own expert set $\mathcal{M}$ using data from Mixture-of-Thoughts (Face, 2025) and OpenScience[1] , where all solutions are generated by DeepSeek-R1 (DeepSeek-AI et al., 2025) to ensure consistent quality. For mathematics, we sample 93,700 instances and categorize them into five subfields (Algebra, Analysis, Discrete Mathematics and Combinatorics, Geometry and Topology, Number Theory), with 200 medium-difficulty problems per subfield reserved for validation. For science, we combine both datasets, selecting 20,000 training and 200 validation samples from each of Biology, Physics, and Chemistry. For code, we use 82,000 training and 10,000 validation samples from Mixture-of-Thoughts. This construction provides broad domain coverage, balanced validation sets, and consistent standards across all expert models.

**Merging $k$ Experts**  In this paper, we are engaged in four merging methods, which are Average merge, TA, TIES and DARE. We use a unified form to represent all of these merging recipes in Table 1. For a given number of experts $k$, we denote by $\mathcal{K} = \{K \subseteq \mathcal{M} : |K| = k\}$ the collection of all $k$-expert subsets of $\mathcal{M}$. Merging all experts in the $K$ can be written as:

$$\theta = \theta_0 + \sum_{i \in K} \alpha_{i,k}\, \Psi(v_i), \qquad \sum_{i \in K} \alpha_{i,k} = c \qquad (2)$$

with a fixed scale $c > 0$ (often $c = 1$).

**Evaluation.**  We report token-level cross-entropy: per domain, we score 30M held-out tokens and average the loss. For each $k$, we aggregate by averaging CE over all $\binom{|\mathcal{M}|}{k}$ expert subsets (or a uniform random subset when $N > 8$B to control cost, and sampling procedure is provided in Appendix.D). Our project, including data and code, is provided in `https://anonymous.4open.science/r/merging_scaling_law-479B`

## 3   SCALING LAWS WITH MERGING EXPERTS AND MODEL SIZE

In this section, we ask a simple question: *As we merge more experts ($k$) and use larger models ($N$), how does the cross-entropy (CE) loss change?* We study this in two standard setups: *in-domain* (evaluation on the single domain) and *cross-domain* (experts drawn from nine heterogeneous domains and evaluated by macro-averaging over all nine). We use four widely adopted merge rules that scale from small to large models: *Average* (Wortsman et al., 2022), *TA* (Ilharco et al.), *TIES* (Yadav et al., 2023), and *DARE* (Yu et al., 2024). Our grids cover $N \in \{0.5, 1.5, 3, 7, 14, 32, 72\}$B (with **10,886** models in total) and $k \in \{1, \ldots, 9\}$; domains are *algebra, analysis, geometry, discrete, number_theory, code, chemistry, physics, biology*.

**Construction of the expected loss.**  For each backbone size $N$ we start from a single base checkpoint and train $M = 9$ domain-specialist experts (*algebra, analysis, geometry, discrete, number_theory, code, chemistry, physics, biology*). Given a merge rule and a target expert number $k$, there are $\binom{M}{k}$ possible expert subsets. In our scaling experiments, for every pair $(N, k)$, we consider all such subsets or a large uniform sample when $\binom{M}{k}$ is big for the big models, and merge the corresponding experts in weight space. We then evaluate the cross-entropy loss $L(N, k, s)$ of the merged model on the held-out data, where $s$ indexes the subset.

We then define the *expected merge loss* at $(N, k)$ as the empirical average over subsets,

$$\widehat{\mathbb{E}}[L \mid N, k] \;=\; \frac{1}{S_{N,k}} \sum_{s=1}^{S_{N,k}} L(N, k, s),$$

with $S_{N,k}$ the number of sampled subsets.[2] Figure 3 illustrates this construction on a representative case, where light points show individual subset losses $L(N, k, s)$ for $N = 0.5$,B, while the solid curve traces the per-$k$ mean $\widehat{\mathbb{E}}[L \mid N, k]$ that we fit our scaling law to. As $k$ grows, the scatter narrows, but the fitted curve remains smooth, which motivates modelling the *mean* behaviour rather than individual subsets. More results with different models and methods are provided in Appendix F.

---

[1] https://huggingface.co/datasets/nvidia/OpenScience

[2] In the grids we use for this paper, $S_{N,k}$ equals the full $\binom{M}{k}$ whenever feasible; otherwise we use a large uniform sample, which gives visually indistinguishable curves.

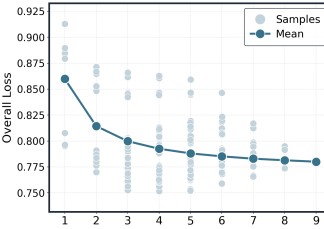 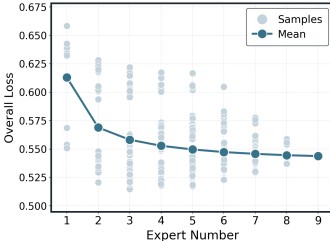 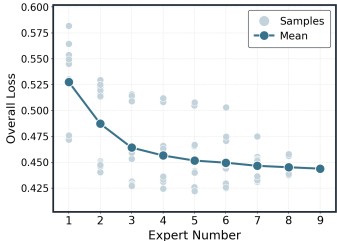

Figure 3: Empirical construction of $\mathbb{E}[L \mid N, k]$ in the cross-domain setting on Qwen-2.5 models at a fixed model size ($N$=0.5B, 7B, 72B). Light points are individual merged models (different expert subsets), and the solid curve is the empirical mean over all subsets at each $k$. The mean loss decreases with $k$ and exhibits diminishing returns, illustrating that our scaling law is defined on the *expected* loss over expert combinations rather than any single merge.

### 3.1 A Unified Empirical Scaling Law

Let $\mathcal{M}$ denote the set of $M$ experts for a given backbone size $N$, and let $K \subseteq \mathcal{M}$ be a subset of size $k$. For a fixed $(N, k)$, choosing $K$ uniformly at random among all $\binom{M}{k}$ subsets and applying a merge rule yields a random merged loss $L$. Throughout this subsection, we therefore study the *conditional expectation* $\mathbb{E}[L \mid N, k]$ over the random choice of $K$.

Empirically, we find that this expected loss admits a simple and interpretable *floor + tail* form with a small finite-$k$ offset:

$$\mathbb{E}[L \mid N, k] = L_\infty(N) + \frac{A(N)}{k + b}, \qquad b \geq 0 \text{ (small)}. \tag{3}$$

Here $L_\infty(N)$ is the limiting "best models can do" as $k \to \infty$, and $A(N)/(k+b)$ is a diminishing-returns term that explains why most gains arrive by small $k$. Both size dependencies are well captured by simple power laws:

$$L_\infty(N) = L_* + B\,N^{-\beta}, \qquad A(N) = A_0\,N^{-\gamma}, \qquad \beta, \gamma \geq 0. \tag{4}$$

*Interpretation.* Bigger models help twice: they lower the floor $L_\infty(N)$ and shrink the tail amplitude $A(N)$, so (i) CE is lower for any fixed $k$, and (ii) fewer experts are needed to get close to the floor.

To fit this power law, we estimate $(L_*, B, \beta, A_0, \gamma, b)$ with weighted nonlinear least squares. Because the empirical variability across runs contracts roughly like $1/k$, we use weights proportional to $k$ when fitting curves in $k$ (this stabilizes early-$k$ noise without over-fitting the tail). All methods and both setups yield near-unity $R^2$ with small, structureless residuals; a tiny $b$ absorbs occasional early-$k$ curvature. Figure 1 plots CE vs. the number of merged experts $k$ at multiple model sizes $N$ for each method; dots are measurements and dotted lines are the fitted $L_\infty(N)+A(N)/(k+b)$ curves. The same visual pattern holds across methods: steep early gains that flatten into a $1/(k+b)$ tail, and a uniform downward shift as $N$ increases.

#### 3.1.1 In-domain (single-domain)

The Figure. 4 shows the Average merging performance in the single algebra domain, and all domains are provided in the Appendix.G.0.1. We can observe that: **(1) Diminishing returns in $k$.** Within each domain, CE decreases monotonically (or near-monotonically) as we merge more experts and follows the $1/(k+b)$ tail predicted by equation 3. Most of the achievable improvement arrives early: there is a clear elbow by $k \approx 5 \sim 6$, after which additional experts yield progressively smaller gains. **(2) Scaling with $N$.** Bigger models help in two orthogonal ways consistent with equation 4: the floor $L_\infty(N)$ drops with $N$ and the tail amplitude $A(N)$ is flat-to-decreasing, so (i) CE is lower at any fixed $k$, and (ii) fewer experts are needed to approach the floor. Math-like domains exhibit shorter tails (earlier saturation), whereas science-like domains benefit more from increasing $k$ before saturating.

Figure 4: Merging Scaling Law in a single algebra domain. (left) CE vs. number of merged experts $(k)$ (middle) C vs. multiple model sizes $(N)$. Dots are real measurements; lines are fits to the unified law $L_\infty(N) + A(N)/(k+b)$ on the single domain. (right) Variance of CE decreases as CE.

### 3.1.2 CROSS-DOMAIN

The Figure.1 shows the power laws across nine domains with expert numbers, and Figure.4 (middle) shows the model size fittings. It is noted that **(1) Same law, pooled over domains.** When merging experts drawn across heterogeneous domains and evaluating by macro-averaged CE, the same floor+tail law equation 3 holds: gains are monotone with $k$, steep early, and flatten into a $1/(k+b)$ tail. The elbow again occurs around $k \approx 5$. **(2) Scaling with $N$.** Increasing model size uniformly shifts curves downward (lower floor) and weakly contracts tails (smaller $A(N)$), mirroring the in-domain behavior: larger models are both better at any fixed $k$ and require fewer experts to approach the pooled floor.

**Summary:** Across both *in-domain* and *cross-domain* settings, the expected merge loss fits the same power law (equation.3). Bigger $N$ lowers the floor and shortens the tail, explaining the monotone gains and early saturation in $k$.

### 3.2 THEORY FOR THE MERGING SCALING LAW

**Assumptions.** Fix a model size $N$. Let $L(\cdot; N)$ be twice continuously differentiable near the base $\theta_0(N)$ with $M(N)$-Lipschitz Hessian $H(N)$ and gradient $g(N)$. Expert/task update vectors $v(N)$ lie in the merged subspace with mean $\mu(N)$, covariance $\Sigma(N)$, and finite sixth moment. We use *equal-normalization* $\alpha_{i,k} = c/k$ (covering uniform averaging, normalized sums, and adapter ensembling); specialized non-uniform weightings can change the tail rate and are outside our scope.

**Theorem 1** (Average-case joint merging law). *Under the assumptions above (equal weights), for each fixed $N$ the population-averaged loss over $k$ merged experts satisfies the second-order law*

$$\mathbb{E}[L \mid N, k] = \underbrace{L(\theta_0; N) + c\, g^\top \mu + \tfrac{1}{2}\, c^2\, \mu^\top H\, \mu}_{L_\infty(N)} + \underbrace{\tfrac{1}{2}\, c^2\, \mathrm{Tr}(H\, \Sigma)}_{A(N)} \cdot \frac{1}{k} + \mathcal{O}_N(k^{-3/2}). \quad (5)$$

*where $H$ denotes an approximation to the **Hessian matrix**, and $\mu, \Sigma$ represent respectively the mean and covariance of task vectors in the merged subspace. In particular, the empirical family equation 3 appears with $b(N) = 0$ at leading order; finite-$k$ effects manifest as a small positive offset in practice. Parameterizing $L_\infty(N), A(N)$ by equation 4 yields the practical joint model $\mathbb{E}[L \mid N, k] = L_* + BN^{-\beta} + A_0 N^{-\gamma}/(k + b_0)$.*

**Corollary 1** (Variance shrinkage). *Let $a_N \triangleq g(N) + H(N)\, c\, \mu(N)$. Under the same assumptions and $a_N^\top \Sigma(N)\, a_N > 0$,*

$$\mathrm{Var}\big(L(\theta_0 + \Delta\theta_k; N)\big) = \Theta\Big(\frac{1}{k}\Big), \qquad \mathrm{sd} = \mathcal{O}\Big(\frac{1}{\sqrt{k}}\Big).$$

*If $a_N^\top \Sigma(N)\, a_N = 0$, the variance contracts faster, at $\Theta(1/k^2)$.*

**Proof:** The detailed proof is provided in Appendix A and B.

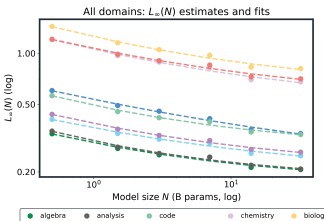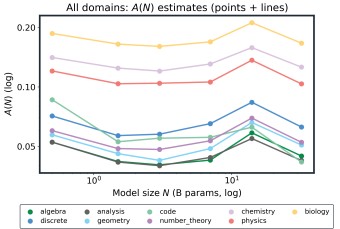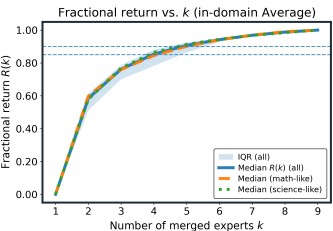

Figure 5: **Larger models are easier to merge.** (Left) Per-domain floors $L_\infty(N)$ fall monotonically with model size $N$. (Middle) Tail amplitude $A(N)$ is small and overall flat-to-decreasing with $N$. **Most of the gain comes from the first few experts.** (Right) Median fractional return $R(k)$ with IQR band; $k{=}5$ and $k{=}6$ cross the $85\%/90\%$ thresholds, respectively. This means only 60% of experts in the expert pool can get over 90% performance.

### 3.3 CORE FINDINGS FOR MERGING

#### 3.3.1 LARGER MODELS MAKE MERGING EASIER

**Setup:** We study the *in-domain* case across nine domains and define "easier to merge" as: at a fixed number of experts $k$, (i) CE is lower, and (ii) the number of experts needed to get $\varepsilon$-close to the domain floor is smaller. Following the unified law in §3.1, we estimate the *floor* $L_\infty(N)$ and the *tail amplitude* $A(N)$ from joint $(N, k)$ fits and summarize them in Fig. 5.

**Findings.** Figure 5 (left) shows that *floors* decay cleanly with model size $N$ across all domains (power-law trend), while Fig. 5 (middle) shows *tails* are small and overall decreasing with oscillation in terms of $N$. Together these two effects explain why larger models are easier to merge: at any fixed $k$ the CE is lower and fewer experts are required to approach the floor. As a headline number, at $k{=}9$ the domain-averaged CE drops from $0.739$ (@0.5B) to $0.430$ (@32B), a **41.9%** reduction. Domains with shorter tails (math-like) saturate earlier; science-like domains benefit more from increasing $k$ but still follow the same floor+tail pattern. Per-domain parameters and worked examples for the experts-to-floor budget are provided in App. I.1.

#### 3.3.2 MIXING DOMAINS HELPS GENERALIZATION

**Findings & why.** As seen in Fig. 1 and 5 (left), cross-domain merging follows the *same* law as in-domain: gains are monotone in $k$, steep early, and flatten into a $1/(k{+}b)$ tail, with an elbow around $k{\approx}5$. Larger $N$ uniformly shifts the pooled curves downward, mirroring the lower floor and smaller tail amplitude from Sec. 3.3.1. The diversity of donors reduces domain-specific bias (lower $L_\infty$) while averaging attenuates variance and leaves a short tail governed by $A(N)/(k{+}b)$. The rare late-$k$ uptick in Fig. 5(middle) is a smaller nonlinearity, and the effect disappears in overall loss, confirming that cross-domain generalization *inherits* the same scaling with diminishing-returns tail.

#### 3.3.3 MOST OF THE GAIN COMES FROM THE FIRST FEW EXPERTS

**Setup.** We quantify the "return" from merging $k$ experts at a fixed $(N, d)$ by the fraction of realized improvement $R(N, d, k)$ computed from the monotone envelope of the measured CE curve (see App. I.2). We summarize two views in Fig. 5: *(left)* the median $R(k)$ over all $(N, d)$ with an IQR band; *(right)* a heatmap of the smallest $k$ that reaches a target return (here $90\%$).

**Findings & why.** As shown in Fig. 5, most of the improvement arrives early: the median curve crosses $85\%$ by $k{=}5$ and $90\%$ by $k{=}6$, and the $k_{90}$ heatmap concentrates in $\{5, 6\}$ across domains and model sizes. Math-like domains tend to saturate slightly earlier, while science-like domains keep a longer—but still flattening—tail. This "early elbow" follows directly from the unified law $L(N, k) = L_\infty(N) + A(N)/(k{+}b)$: the marginal gain $\Delta_k \approx A(N)/[(k{+}b)(k{+}1{+}b)]$ decays roughly as $k^{-2}$, so returns diminish sharply beyond the first few experts.

#### 3.3.4 MERGE METHODS DIFFER LITTLE AT LARGE SCALE

**Setup.** We compare four merge methods, Average, TA ($\lambda{=}0.8$), TIES ($\lambda{\in}\{0.5, 1\}$), and DARE (density 0.2), under the same protocol as before, reporting macro-averaged CE across nine domains

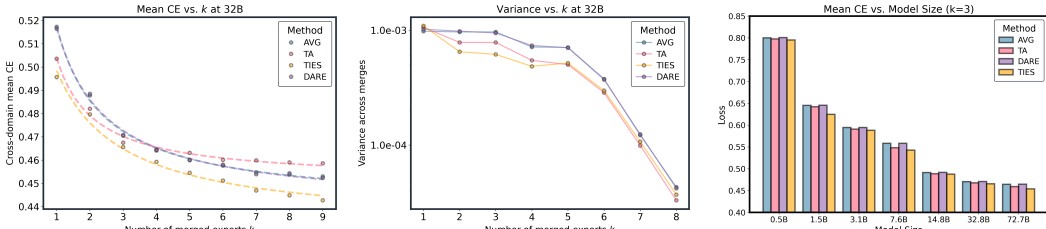

Figure 6: **Method sensitivity is little at scale.** *Left:* Mean CE vs. $k$ at $N{=}32$B—all methods follow the power law; the early-$k$ lead of TA/TIES(0.5) is small ($\sim$1–2%) and narrows by $k{\gtrsim}8$. *Right:* Variance vs. $k$ at $N{=}32$B, near-$1/k$ contraction; TIES/TA $<$ Average at small $k$, and all methods meet near the variance floor by $k{\approx}8$. Curves show measurements (markers) and floor+tail fits (lines) with a shared small $b$ per method.

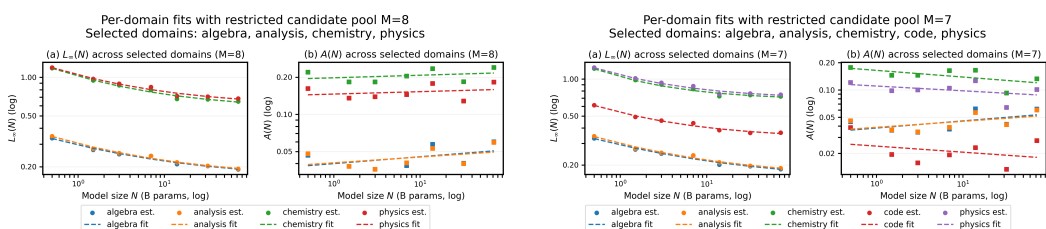

Figure 7: **Effect of candidate-pool size.** Two restricted-pool fits of the unified law (decreasing the number of candidates from M=9 to 8 and 7). *Left (M=8), Right (M=7):* floors $L_\infty(N)$ are tight across domains; tails $A(N)$ show weak or no shrinkage with $N$.

and fitting each curve with the unified law. Figure 6a shows mean CE vs. $k$ at $N{=}32$B; Fig. 6b shows the corresponding merge-to-merge variance.

**Findings & why.** As $k$ grows (and especially at larger $N$), method gaps in *mean* CE compress quickly: in Fig. 6a, small early advantages (TA/TIES at $k{\leq}3$) shrink to a tight band by $k{\approx}8$ (differences $\lesssim 2\%$). Variance exhibits the same convergence (Fig. 6b), contracting near $\sim 1/k$ and approaching a small floor where all methods meet. This behavior follows directly from the shared scaling form: the diminishing-returns tail $A(N)/(k{+}b)$ makes early steps method-sensitive, while the common floor $L_\infty(N)$ dominates at larger $k$ and $N$, leaving only second-order differences. The results are consistent with the observations of (Yadav et al., 2024), further confirming their findings.

## 4 FURTHER ANALYSIS AND CHECKS

Beyond establishing the unified law in §3.1, we stress–test it along practical axes that affect day-to-day merging: how large the candidate pool is, whether mixing domains helps, how sensitive results are to order/selection, and whether findings transfer across backbones. Throughout, we keep the evaluation protocol fixed and re-estimate the same $L_\infty(N) + A(N)/(k{+}b)$ family. The main text reports trends and takeaways; per-domain numbers and fit diagnostics are in the Appendix.

### 4.1 DOES A BIGGER CANDIDATE POOL HELP?

**Setup.** We repeat the cross-domain analysis while *restricting* the pool of available donor domains to $M \in \{8, 7\}$ (DARE; identical $(N, k)$ grids), then refit the unified law. Figure 7 contrasts the fitted floor $L_\infty(N)$ and tail $A(N)$ for $M{=}8$ vs. $M{=}7$.

**Findings & why.** The law itself is *stable* to pool size: floors remain tight power laws in $N$ with negligible change across $M$ (Fig. 7, both panels). The effect of a larger pool shows up almost entirely in the *tail*: moving from $M{=}8$ to $M{=}7$ makes $A(N)$ *flat-to-decreasing* with $N$ on science-like domains (chemistry/physics) while leaving math-like domains nearly unchanged. Intuitively, a slightly more diverse pool supplies complementary donors and reduces residual cross-domain mismatch, shrinking the $A(N)/(k{+}b)$ term; this yields the clearest gains at moderate-to-large $k$ and larger $N$. In short, a bigger pool chiefly helps by tightening the tail rather than shifting the floor.

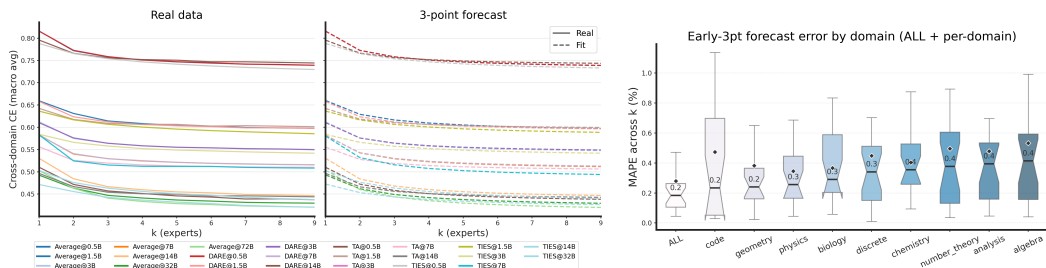

Figure 8: **Predicting the $k$-curve from three points.** *Left:* ground truth (markers) versus a floor+tail fit using only $k=\{1, 2, 4\}$ (lines) across representative domains/methods. *Right:* forecast error as MAPE across $k$ (lower is better) and the distribution of recommended $k^\star$ under $\Delta=1\%$, concentrated at $5-6$. Together these show that three points suffice to recover the full curve and yield a practical early-stop $k^\star$.

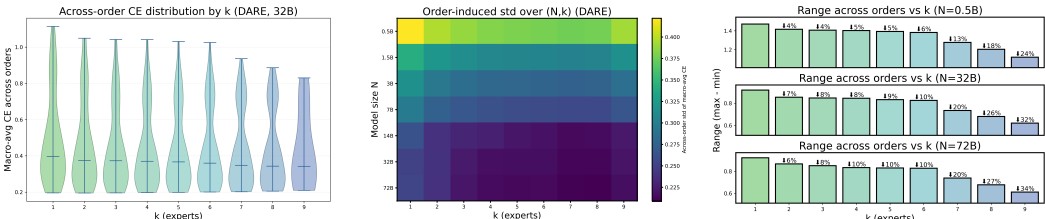

Figure 9: **Order sensitivity contracts with $k$ (DARE).** *Left:* At $N=32$B, the distribution of macro CE across merge orders (violins) tightens quickly as $k$ increases; the whisker length shrinks by $\sim 83\%$ from $k=1$ to $k=8$, while the median curve is monotone in $k$. *Middle:* Heatmap of across-order std over $(N, k)$ shows a robust left-to-right decay at all scales, consistent with a near-$1/(k+b)$ tail (larger $N$ is also slightly darker, i.e., more stable). *Right:* Worst–best spread (range) decreases steadily with $k$; the annotated relative reductions vs. $k=1$ are about $\{24\%, 32\%, 34\%\}$ at $k=9$ for $N \in \{0.5, 32, 72\}$B, respectively. In absolute CE terms (32B), the range drops from $\approx 0.086$ at $k=1$ to $\approx 0.015$ at $k=8$ (Appx.).

## 4.2 CAN THREE POINTS PREDICT THE WHOLE $k$-CURVE? (YES)

**Setup.** For each series, either a single *(domain, N)* in-domain curve or a *(method, N)* cross-domain curve, we fit the unified law $L(k) = L_\infty(N) + \frac{A(N)}{k+b}$ using only the first three points $k \in \{1, 2, 4\}$, then forecast the full $k \in \{1, \ldots, 9\}$ trajectory and the value at a target $k$.

**Findings & why.** *Three points suffice.* Across domains and methods, the early-$k$ slope plus the long-tail shape are captured well by $L_\infty + A/(k+b)$, so fitting on $\{1, 2, 4\}$ closely tracks the full curve in Fig. 8(left). The implied $k^\star$ concentrates around $5 \sim 6$ in Fig. 8(right), aligning with the elbow found in §3.3.3. Intuitively, the model's floor $L_\infty$ anchors the late regime while $A$ controls the early drop; those two degrees of freedom are identifiable from three well-spaced points, yielding stable forecasts without overfitting. This can guide us to make decisions for the merging in practice.

## 4.3 DOES MERGE ORDER MATTER?

**Setup.** We permute donor orders under DARE, and at each $(N, k)$ summarize the across-order dispersion of the macro-averaged CE by standard deviation, range, and coefficient of variation; we also fit a parsimonious tail $\text{Std}_{\text{order}}(N, k) \approx c_0(N) + c_1(N)/(k+b)$.

**Findings.** *Order effects fade fast.* Fig. 9 (left) shows that, at 32B, both the interquartile mass and the whiskers collapse as $k$ grows (about $83\%$ shrinkage in whisker length by $k=8$). Fig. 9 (middle) confirms that this contraction holds *for all* base sizes and follows the same $1/(k+b)$ pattern that governs the mean, with larger $N$ being slightly more stable. Fig. 9 (right) quantifies worst–best differences: the relative range reduction at $k=9$ is $\approx 24\%$ (0.5B), $32\%$ (32B), and $34\%$ (72B), and in absolute CE the 32B spread falls from $\sim 0.086$ (@$k=1$) to $\sim 0.015$ (@$k=8$). Practically, once $k \geq 6$

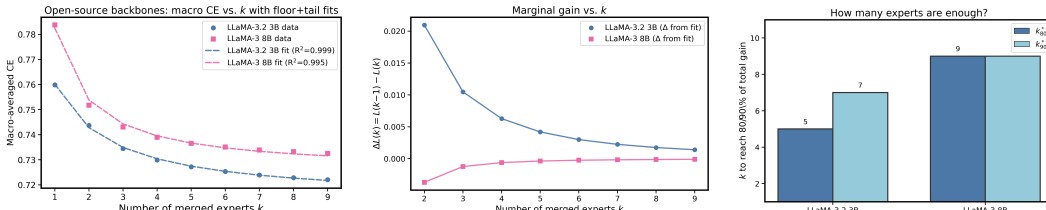

Figure 10: **Cross-backbone validation on LLaMA.** *Left:* Macro CE vs. $k$ on LLaMA-3.2 3B and LLaMA-3 8B, with floor+tail fits $L_\infty + \frac{A}{k+b}$ showing the same inverse tail. *Middle:* Marginal gain $\Delta L(k)$ decays smoothly with $k$, consistent with the $1/(k+b)$ form. *Right:* Experts-to-target $k^\star_{80/90}$ concentrates at small $k$, echoing that most gains come early.

the across-order spread is small compared to early-$k$ method gaps and to the floor itself, so curating a specific merge order yields little benefit.

### 4.4 DOES THE SAME LAW HOLD ON OTHER BACKBONES?

**Setup.** We replicate the cross-domain protocol from §3.1 on two open-source backbones, LLaMA-3.2 3B and LLaMA-3 8B. For each backbone, we merge experts sampled across nine domains, report the macro-averaged CE for $k \in \{1, \ldots, 9\}$, and fit the same floor+tail form $L(k) = L_\infty + \frac{A}{k+b}$ with a small $b$. To complement the main *curve fit*, we also visualize the *marginal gain* $\Delta L(k) = L(k-1) - L(k)$ and the *experts-to-target* bars $k^\star_{80/90}$ (the smallest $k$ that reaches 80/90% of the total $k=1 \rightarrow 9$ improvement).

**Findings.** Both backbones follow the *same* law with a clear $1/(k+b)$ tail: CE decreases monotonically in $k$, gains are steep at small $k$ and flatten thereafter (Fig. 10a). The *marginal gain* curves (Fig. 10b) decay smoothly with $k$, illustrating diminishing returns consistent with the inverse tail; *experts-to-target* bars (Fig. 10c) show that both backbones need only a *small* number of experts to secure most of the improvement (80/90% reached at modest $k$), in line with our main-scale observations. Absolute CE levels differ slightly, LLaMA-3.2 3B sits lower and shows a marginally steeper early slope, reflecting *backbone capability* (series 3.2 vs. 3.0), not a change of law. In short, the merging scaling law *transfers across architectures*: same functional form, same diminishing-returns shape, and similar fitted parameter ranges. Additional results on Gemma 2 also follow the same functional form, provided in the Appendix. J

**Note 1:** We also provide the detailed downstream performance to answer the question "*Do downstream metrics follow the same trend?*" in the Appendix. J. In short, aggregated task metrics (mean metric over benchmarks and expert subsets) improve with $k$ and then plateau, exhibiting the same floor+tail shape as our CE scaling law.

**Note 2:** We further extend the cross-domain experiments to a 16-domain pool on the LLaMA-3B-Instruct backbone (original 9 domains plus Japanese, medical, house-arrangement, Korean, emotion, elementary school mathematics, and Java-code experts), and the aggregated cross-entropy still follows the same floor+tail scaling law (see Appendix K).

## 5 CONCLUSION

This paper presented a simple, predictive *merging scaling law* that links model size and the number of merged experts via a floor+tail power law. This law unifies a broad set of empirical regularities: larger bases lower the size-dependent floor, most improvement arrives at small $k$, variance contracts with additional experts, method gaps compress at scale, and merge order quickly becomes inconsequential. The same power law form holds in-domain and cross-domain, and transfers across architectures and representative merging methods. Beyond description, the law is prescriptive. A lightweight fit from a few early points forecasts the full loss–vs.–$k$ curve and recommends an efficient expert count, enabling budget-aware decisions about when to stop adding experts and how to trade off scaling the base model versus increasing $k$. Together, these results elevate merging from heuristic practice to a computationally efficient, planable alternative to multitask fine-tuning.

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

LIMITATIONS.

Our study centers on cross-entropy and equal-normalized composition; extending to other objectives and adaptive weighting is an important next step. While the law is robust across datasets, methods, and backbones we tested, probing extreme scales, additional modalities, and broader downstream metrics (robustness, safety, calibration) remains future work. On the theoretical side, refining the link between floor/tail parameters, curvature anisotropy, and domain dispersion, and developing selection/ordering policies that exploit these quantities, could further tighten predictions and automate practical merging at scale.

## A  DETAILED PROOF OF THEOREM 1

We fix a model size $N$ and omit $(N)$ when clear. Following Assumption 3.2: (i) $L$ is twice continuously differentiable near $\theta_0$ with an $M$-Lipschitz Hessian; (ii) task vectors $v_i$ are i.i.d. with mean $\mu$ and covariance $\Sigma$, and $\mathbb{E}\|v_i\|^6 < \infty$; (iii) equal-weight normalisation $\alpha_{i,k} = c/k$.

**Decomposition.**  Let

$$\Delta\theta_k(S) = \sum_{i \in S} \tfrac{c}{k} v_i = c\,\mu + \varepsilon_k(S), \qquad \varepsilon_k(S) := \tfrac{c}{k} \sum_{i \in S} (v_i - \mu).$$

Expectation $\mathbb{E}[\cdot]$ is with respect to the uniform random $k$-subset $S$ (the same orders follow for i.i.d. sampling with replacement) and $\varepsilon$ means the sampling error.

**Lemma 1** (Moments of the mean-corrected step). $\mathbb{E}[\varepsilon_k] = 0$ and $\mathbb{E}[\varepsilon_k \varepsilon_k^\top] = \frac{c^2}{k}\Sigma$. Moreover, $\mathbb{E}\|\varepsilon_k\|^3 = \mathcal{O}(k^{-3/2})$ under $\mathbb{E}\|v_i\|^6 < \infty$.

*Proof.* Linearity gives $\mathbb{E}[\varepsilon_k] = 0$. For the second moment, averaging $k$ i.i.d. centred vectors yields covariance $c^2\Sigma/k$. The $p = 3$ Marcinkiewicz–Zygmund (Ortega-Cerdà & Saludes, 2007; Ibragimov & Sharakhmetov, 1999) inequality gives

$$\mathbb{E}\|\varepsilon_k\|^3 \le \frac{C_3\,c^3}{k^{3/2}} \Big(\mathbb{E}\|\xi_1\|^2\Big)^{3/2} + \frac{C_3'\,c^3}{k^2}\mathbb{E}\|\xi_1\|^3 = \mathcal{O}\Big(\frac{1}{k^{3/2}}\Big),$$

for $\xi_i := v_i - \mu$, hence the stated rate after multiplying by $c^3$. $\qquad\square$

**Step 1: Taylor expand at $\theta_0 + c\mu$.**  Define $\phi(\delta) := L(\theta_0 + c\mu + \delta)$. Let $a := \nabla\phi(0) = \nabla L(\theta_0 + c\mu)$ and $H_S := \nabla^2\phi(0) = \nabla^2 L(\theta_0 + c\mu)$. The Hessian is $M$-Lipschitz, hence the second-order Taylor formula with remainder

$$\phi(\delta) = \phi(0) + a^\top\delta + \tfrac{1}{2}\delta^\top H_S \delta + R_S(\delta), \qquad |R_S(\delta)| \le \tfrac{M}{6}\|\delta\|^3. \tag{6}$$

Plugging $\delta = \varepsilon_k(S)$ and taking expectation, using Lemma 1,

$$\mathbb{E}\big[L(\theta_k(S))\big] = L(\theta_0 + c\mu) + a^\top\mathbb{E}[\varepsilon_k] + \tfrac{1}{2}\mathbb{E}\big[\varepsilon_k^\top H_S \varepsilon_k\big] + \mathbb{E}[R_S(\varepsilon_k)]$$

$$= L(\theta_0 + c\mu) + \tfrac{1}{2}\operatorname{Tr}\big(H_S\,\mathbb{E}[\varepsilon_k \varepsilon_k^\top]\big) + \mathbb{E}[R_S(\varepsilon_k)]$$

$$= L(\theta_0 + c\mu) + \tfrac{1}{2}c^2 \operatorname{Tr}\big(H_S\Sigma\big) \cdot \frac{1}{k} + \mathcal{O}\Big(\frac{1}{k^{3/2}}\Big). \tag{7}$$

Thus, at the *asymptote point* $\theta_0 + c\mu$ the averaged curve has a $1/k$ tail with coefficient $\tfrac{1}{2}c^2\operatorname{Tr}(H_S\Sigma)$, up to $\mathcal{O}(k^{-3/2})$.

**Step 2: relate $(L_\infty(N), A(N))$ used in the main text to the above.**  In the main text we present the $k \to \infty$ intercept and tail amplitude at the *base* $\theta_0$, using a PSD curvature surrogate $H$ (e.g., GGN/Fisher) evaluated at $\theta_0$:

$$L_\infty(N) := L(\theta_0) + c\,g^\top\mu + \tfrac{1}{2}c^2\,\mu^\top H\,\mu, \qquad A(N) := \tfrac{1}{2}c^2\operatorname{Tr}(H\Sigma),$$

where $g = \nabla L(\theta_0)$.

To connect these to equation 7, apply Taylor at $\theta_0$ with the same Lipschitz-$M$ control:

$$L(\theta_0+c\mu) = L(\theta_0)+c\,g^\top\mu+\tfrac{1}{2}c^2\,\mu^\top H\,\mu + \rho_0, \qquad |\rho_0| \leq \frac{M}{6}\,c^3\|\mu\|^3 + \underbrace{\tfrac{1}{2}c^2\,|\mu^\top(\nabla^2 L(\theta_0) - H)\mu|}_{\text{curvature surrogate error}}.$$
(8)

Similarly, since $\|H_S - \nabla^2 L(\theta_0)\|_{\text{op}} \leq M\,c\|\mu\|$ (Hessian Lipschitz along the segment),

$$\text{Tr}(H_S\Sigma) = \text{Tr}(H\Sigma) + \eta_0, \qquad |\eta_0| \leq \|(H_S - H)\|_{\text{op}}\,\text{Tr}(\Sigma) \leq M\,c\|\mu\|\,\text{Tr}(\Sigma) + |\text{Tr}((\nabla^2 L(\theta_0) - H)\Sigma)|.$$
(9)

Combining equation 7–equation 9,

$$\mathbb{E}\big[L(\theta_k(S))\big] = \underbrace{L(\theta_0) + c\,g^\top\mu + \tfrac{1}{2}c^2\,\mu^\top H\,\mu}_{L_\infty(N)} + \underbrace{\tfrac{1}{2}c^2\,\text{Tr}(H\Sigma)}_{A(N)}\cdot\frac{1}{k} + R_{N,k},$$
(10)

with an explicit error bound

$$|R_{N,k}| \leq \underbrace{|\rho_0|}_{\mathcal{O}(\|\mu\|^3)\,+\,\text{surrogate}} + \underbrace{\frac{1}{2}c^2\frac{|\eta_0|}{k}}_{\mathcal{O}(\|\mu\|/k)\,+\,\text{surrogate}} + \underbrace{C\,k^{-3/2}}_{\text{from }\mathbb{E}[R_S(\varepsilon_k)]},$$
(11)

where $C$ depends on $M$, $c$ and the sixth-moment bound of $v_i$. Hence,

$$\boxed{\mathbb{E}[L \mid N,k] = L_\infty(N) + \frac{A(N)}{k} + \mathcal{O}_N\Big(\frac{1}{k^{3/2}}\Big) + \mathcal{O}_N(\|\mu\|^3) + \mathcal{O}_N\Big(\frac{\|\mu\|}{k}\Big) + \text{(error)}.}$$

**Interpretation of the approximation terms.** The $\mathcal{O}(k^{-3/2})$ term is the genuine *averaging* remainder from Step 1. The $\mathcal{O}(\|\mu\|^3)$ and $\mathcal{O}(\|\mu\|/k)$ terms arise from *using base-point coefficients* $(g, H)$ to parameterise the intercept and tail: when $\|\mu\|$ is moderate (typical in practice for adapter/LoRA merging or small $c$), these terms are dominated by the leading $1/k$ tail. Any persistent curvature-surrogate mismatch at $\theta_0$ is absorbed into the (fitted) $L_\infty(N)$ and $A(N)$ in the empirical model; it does not change the $1/k$ rate.

**Conclusion (Theorem 1 in $\approx$ form).** Collecting the above, for each fixed $N$,

$$\mathbb{E}[L \mid N,k] \approx L_\infty(N) + \frac{A(N)}{k},$$

with a quantitative remainder given by equation 11. Equivalently, at the granularity used in the main text,

$$\mathbb{E}[L \mid N,k] = L_\infty(N) + \frac{A(N)}{k} + \mathcal{O}_N\Big(\frac{1}{k^{3/2}}\Big),$$

where the $N$-dependent constants (including the small base-point/curvature-surrogate discrepancies) are absorbed into $L_\infty(N), A(N)$—exactly the form fitted in our 2D scaling law. $\qquad\square$

## B  DETAILED PROOF OF COROLLARY 1

We continue with the setting and notation of Appendix A. Fix a model size $N$ and omit $(N)$ when clear. Based on the equation 6, the second-order expansion at $\theta_0 + c\mu$:

$$L(\theta_0 + c\mu + \delta) = L(\theta_0 + c\mu) + a^\top\delta + \tfrac{1}{2}\delta^\top H_S\delta + R_S(\delta), \qquad |R_S(\delta)| \leq \tfrac{M}{6}\|\delta\|^3, \quad (12)$$

with $a := \nabla L(\theta_0 + c\mu)$ and $H_S := \nabla^2 L(\theta_0 + c\mu)$. Besides Lemma 1 (which gave $\mathbb{E}[\varepsilon_k] = 0$, $\text{Cov}(\varepsilon_k) = \frac{c^2}{k}\Sigma$, and $\mathbb{E}\|\varepsilon_k\|^3 = \mathcal{O}(k^{-3/2})$), we will need $p=4,6$ moment bounds. By Marcinkiewicz–Zygmund / Rosenthal inequalities (Ortega-Cerdà & Saludes, 2007),

$$\mathbb{E}\|\varepsilon_k\|^p = \mathcal{O}(k^{-p/2}), \qquad p \in \{2,4,6\}.$$
(13)

Then we make a variance decomposition. By equation 12 with $\delta = \varepsilon_k(S)$,

$$L(\theta_k(S)) = C + \underbrace{a^\top\varepsilon_k}_{L_1} + \underbrace{\tfrac{1}{2}\varepsilon_k^\top H_S\varepsilon_k}_{L_2} + \underbrace{R_S(\varepsilon_k)}_{L_3}, \qquad C := L(\theta_0 + c\mu).$$

Hence

$$\mathrm{Var}\big(L(\theta_k(S))\big) = \mathrm{Var}(L_1) + \mathrm{Var}(L_2) + \mathrm{Var}(L_3)$$
$$+ 2\mathrm{Cov}(L_1, L_2) + 2\mathrm{Cov}(L_1, L_3) + 2\mathrm{Cov}(L_2, L_3). \qquad (14)$$

We bound the six pieces one by one.

**(i) Linear term:** $\mathrm{Var}(L_1)$. Since $\mathrm{Var}[\varepsilon_k] = 0$ and $\mathrm{Cov}(\varepsilon_k) = \frac{c^2}{k}\Sigma$,

$$\mathrm{Var}(L_1) = \mathrm{Var}(a^\top \varepsilon_k) = a^\top \mathrm{Cov}(\varepsilon_k)a = \frac{c^2}{k} a^\top \Sigma a. \qquad (15)$$

**(ii) Quadratic term:** $\mathrm{Var}(L_2)$. Using $(x^\top A x)^2 \le \|A\|_F^2 \|x\|^4$,

$$\mathbb{E}[L_2^2] = \tfrac{1}{4}\mathbb{E}\big[(\varepsilon_k^\top H_S \varepsilon_k)^2\big] \le \tfrac{1}{4}\|H_S\|_F^2 \,\mathbb{E}\|\varepsilon_k\|^4 = \mathcal{O}\Big(\frac{1}{k^2}\Big)$$

by equation 13 with $p{=}4$. Moreover $\mathbb{E}[L_2] = \tfrac{1}{2}\mathbb{E}[\varepsilon_k^\top H_S \varepsilon_k] = \tfrac{1}{2}\mathrm{Tr}(H_S \mathrm{Cov}(\varepsilon_k)) = \tfrac{1}{2}\frac{c^2}{k}\mathrm{Tr}(H_S\Sigma)$, so $|\mathbb{E}[L_2]| = \mathcal{O}(1/k)$, hence $|\mathbb{E}[L_2]|^2 = \mathcal{O}(1/k^2)$. Therefore

$$\mathrm{Var}(L_2) = \mathbb{E}[L_2^2] - \mathbb{E}[L_2]^2 = \mathcal{O}\Big(\frac{1}{k^2}\Big). \qquad (16)$$

**(iii) Remainder:** $\mathrm{Var}(L_3)$. By equation 12, $|L_3| \le \frac{M}{6}\|\varepsilon_k\|^3$, so $\mathbb{E}[L_3^2] \le \left(\frac{M}{6}\right)^2 \mathbb{E}\|\varepsilon_k\|^6 = \mathcal{O}\Big(\frac{1}{k^3}\Big)$ by equation 13 with $p{=}6$, hence

$$\mathrm{Var}(L_3) \le \mathbb{E}[L_3^2] = \mathcal{O}\Big(\frac{1}{k^3}\Big). \qquad (17)$$

**(iv) Covariances.** By Cauchy–Schwarz and the above variance bounds,

$$|\mathrm{Cov}(L_1, L_2)| \le \sqrt{\mathrm{Var}(L_1)\mathrm{Var}(L_2)} = \mathcal{O}\Big(\frac{1}{k^{3/2}}\Big), \qquad (18)$$

$$|\mathrm{Cov}(L_1, L_3)| \le \sqrt{\mathrm{Var}(L_1)\mathrm{Var}(L_3)} = \mathcal{O}\Big(\frac{1}{k^2}\Big), \qquad (19)$$

$$|\mathrm{Cov}(L_2, L_3)| \le \sqrt{\mathrm{Var}(L_2)\mathrm{Var}(L_3)} = \mathcal{O}\Big(\frac{1}{k^{5/2}}\Big). \qquad (20)$$

Then combining equation 14–equation 20,

$$\mathrm{Var}\big(L(\theta_k(S))\big) = \frac{c^2}{k} a^\top \Sigma a + \mathcal{O}\Big(\frac{1}{k^2}\Big). \qquad (21)$$

Here $\mathcal{O}(1/k^2)$ is a *one-sided upper bound* on the remainder. If $\alpha > 0$, which is non-degenerate case, there exist constants $C_1, C_2 > 0$ and $k_0$ such that for all $k \ge k_0$,

$$\frac{C_1}{k} \le \mathrm{Var}\big(L(\theta_k(S))\big) \le \frac{C_2}{k},$$

hence

$$\mathrm{Var}\big(L(\theta_k(S))\big) = \Theta\Big(\frac{1}{k}\Big), \qquad \mathrm{sd}\big(L(\theta_k(S))\big) = \mathcal{O}\Big(\frac{1}{\sqrt{k}}\Big).$$

For the degenerate linear term, where $a^\top \Sigma a = 0$, the linear contribution vanishes and equation 16–equation 20 give the uniform bound

$$\mathrm{Var}\big(L(\theta_k(S))\big) = \mathcal{O}\Big(\frac{1}{k^2}\Big).$$

Moreover, whenever $H_S$ is nonzero on the range of $\Sigma$ and the fourth central moments of $v_i$ are not all degenerate along that range (a mild condition satisfied in our experiments), the quadratic fluctuation has *nonzero* variance constant, so the bound is tight:

$$\mathrm{Var}\big(L(\theta_k(S))\big) = \Theta\Big(\frac{1}{k^2}\Big).$$

$\square$

Table 2: Training Hyperparameters

| Hyperparameter | Value |
|---|---|
| Batch Size | 16 |
| Learning Rate | $1 \times 10^{-5}$ |
| Warmup Ratio | 0.05 |
| Number of Epochs | 2 |
| Maximum Sequence Length | 16,384 |
| Optimizer | Adam (with offloading) |
| Precision | bfloat16 |
| Gradient Checkpointing | Enabled |
| Zero Redundancy Optimizer Stage | 3 |

## C  EXPERT MODEL DETAILS

For evaluation, we evaluate model performance using *(token-level) cross-entropy loss*. For each domain, we randomly sample 30M tokens from the corresponding validation set. Let $x_t$ denote the $t$-th token sequence in the evaluation set and $p_\theta(x_t)$ the probability assigned by the model parameterized by $\theta$. The domain-specific loss is defined as the average negative log-likelihood:

$$\mathcal{L}_{\text{overall}} = -\frac{1}{\sum_{i \in \mathcal{M}} T_i} \sum_{i \in \mathcal{M}} \sum_{t=1}^{T_i} \log p_\theta(x_t | x_{t-1}, .., x_1),$$

where $T_i$ is the number of tokens in domain $i$. Even for a given $k$, we have $\binom{|\mathcal{M}|}{k}$ possible selections to merge. Each such choice yields a potentially distinct merged model. This indicates that the loss is not only a function of $k$ but also depends on which specific domains are included. Therefore, for a fixed $k$, we enumerate all $\binom{|\mathcal{M}|}{k}$ possible subsets of domain experts and compute the expected loss over them.[3]

**Note:** We isolate weight-space merging and its scaling; complementary *model fusion* baselines (e.g., InfiGFusion, InfiFPO Wang et al. (2025b); Gu et al. (2025)) are different ways as they require data and additional training.

## D  SAMPLING ALGORITHM

---
**Algorithm 1** Diverse Permutation Generation

---
**Require:** $k \in \mathbb{N}$, base sequence $\mathbf{s} = [1, 2, \ldots, 9]$
**Ensure:** Set of $k$ diverse permutations $\mathcal{P} = \{\pi_1, \ldots, \pi_k\}$
 1: Initialize $\mathcal{P} \leftarrow \{\mathbf{s}\}$
 2: **if** $k \geq 2$ **then**
 3:    $\mathcal{P} \leftarrow \mathcal{P} \cup \{\text{reverse}(\mathbf{s})\}$
 4: **end if**
 5: **for** $i = 3$ to $k$ **do**
 6:    Generate candidate set $\mathcal{C}$ by random shuffling of $\mathbf{s}$ ($|\mathcal{C}| = 1000$)
 7:    $\pi^* \leftarrow \arg\max_{\pi \in \mathcal{C}} \min_{\pi' \in \mathcal{P}} d_H(\pi, \pi')$
 8:    $\mathcal{P} \leftarrow \mathcal{P} \cup \{\pi^*\}$
 9: **end for**
10: **return** $\mathcal{P}$

---

[3]Since the computational overhead of model merging grows with model size, we mitigate the cost by randomly sampling a subset of all possible combinations when the model size exceeds 8B parameters. The complete sampling procedure is detailed in the next section.

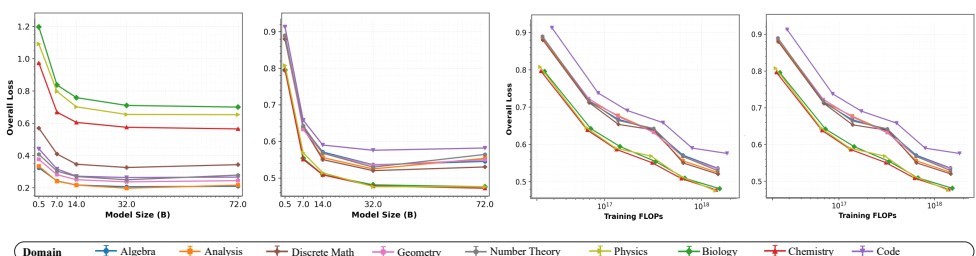

Figure 12: Expert Post-training Scaling Law. Expert models performance improves as we increase the model size, computational budget used for post-training.

We employ Algorithm 1 to perform sampling over model merge combinations, where $d_H$ denotes the Hamming distance. Figure 11 illustrates a comparison between curves obtained via our sampling strategy (where $k = 15$) and those obtained from full merging combinations on the 0.5B model. The results demonstrate that the sampled curves closely align with the full ones, both in terms of overall trend and numerical values.

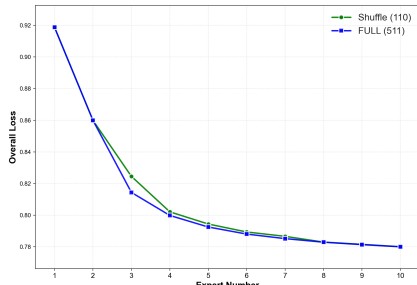

Figure 11: Results for different numbers of merged experts on the 0.5B model. The base model is also considered one expert.

## E    SCALING LAWS FOR EXPERT MODEL TRAINING

In addition to investigating the scaling laws of model merging, we further examine the scaling behavior of expert models during the post-training stage. Specifically, we conduct a systematic analysis across different domains to understand how post-training affects expert models. Our study focuses on characterizing the relationship between the magnitude of the loss and three key factors: model size, the number of training tokens, and the overall computational budget. This analysis provides new insights into the scaling laws that govern post-training dynamics and highlights their potential applicability across diverse domains.

Figure 12 illustrates the performance of expert models, measured in terms of loss, as a function of model size and computational budget. Overall, we observe a consistent trend across domains: larger models and greater computation generally yield improved performance. This observation aligns with the well-established *language modeling scaling law* (Kaplan et al., 2020). Nevertheless, an important distinction arises across domains. For instance, the performance curve in the *Biology* domain exhibits substantially higher loss values compared to that in *Geometry*, even under comparable training conditions. This suggests that the model's pre-existing knowledge reserves differ across domains, leading to heterogeneous post-training dynamics despite identical training configurations. Such domain-specific disparities may further induce instability when merging expert models trained on heterogeneous knowledge bases.

## F    EMPIRICAL CONSTRUCTION OF $\mathbb{E}[L \mid N, k]$

In this section, Figures illustrate the expected loss of different representative cases, where light points show individual subset losses $L(N, k, s)$ for different model sizes, $N$B, while the solid curve traces the per-$k$ mean $\widehat{\mathbb{E}}[L \mid N, k]$ that we fit our scaling law to. As $k$ grows, the scatter narrows, but the fitted curve remains smooth, which motivates modelling the *mean* behaviour rather than individual subsets.

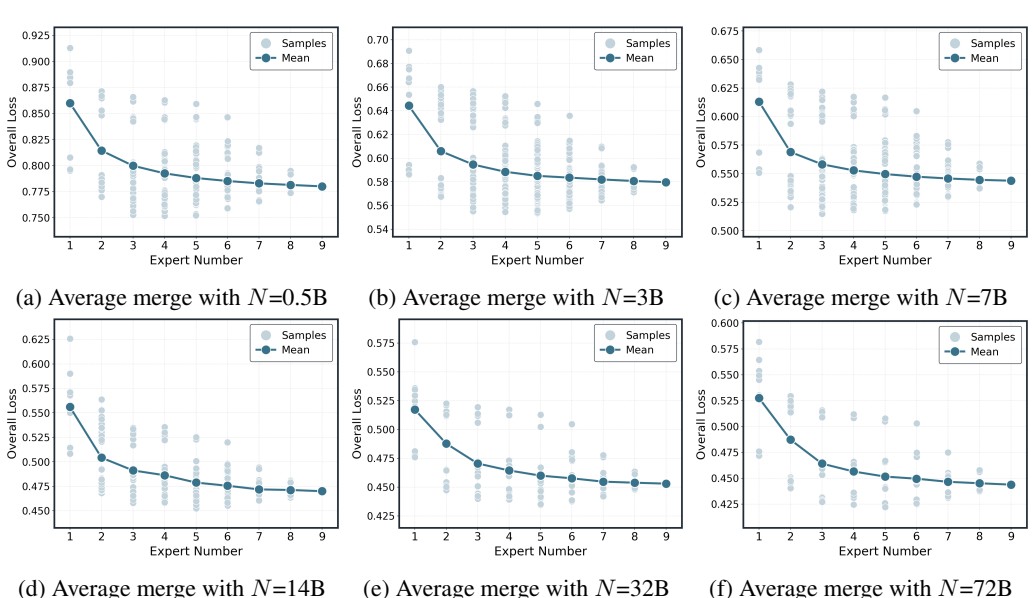

(a) Average merge with $N$=0.5B  (b) Average merge with $N$=3B  (c) Average merge with $N$=7B

(d) Average merge with $N$=14B  (e) Average merge with $N$=32B  (f) Average merge with $N$=72B

Figure 13: Empirical construction of $\mathbb{E}[L \mid N, k]$ in the cross-domain setting. Each figure shows average merging on Qwen 2.5 at a fixed model size. Light points are individual merged models (different expert subsets and seeds), and the solid curve is the empirical mean over all subsets at each $k$.

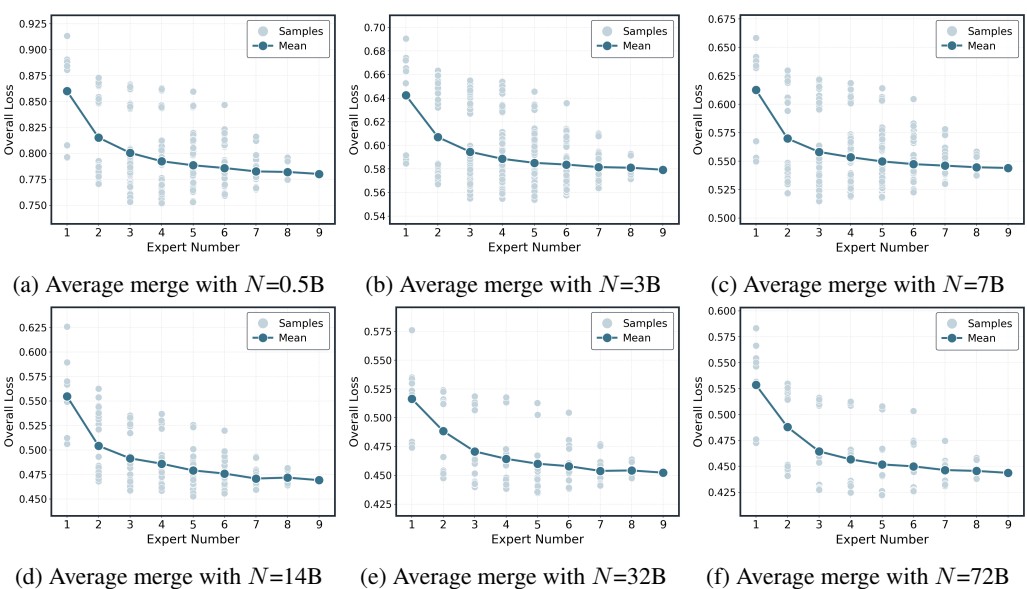

(a) Average merge with $N$=0.5B  (b) Average merge with $N$=3B  (c) Average merge with $N$=7B

(d) Average merge with $N$=14B  (e) Average merge with $N$=32B  (f) Average merge with $N$=72B

Figure 14: Empirical construction of $\mathbb{E}[L \mid N, k]$ in the cross-domain setting. Each figure shows DARE merging on Qwen 2.5 at a fixed model size.

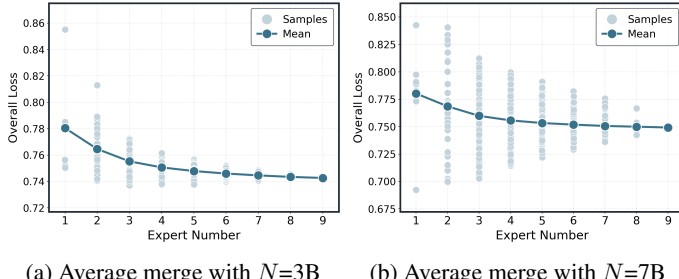

(a) Average merge with $N$=3B      (b) Average merge with $N$=7B

Figure 15: Empirical construction of $\mathbb{E}[L \mid N, k]$ in the cross-domain setting. Each figure shows TA merging on LLaMA-3.1/3.2 at a fixed model size.

Table 3: Joint $(N, k)$ fit for Average (per-domain parameters).

| domain | Lstar | B | beta | A0 | gamma | b0 | R2 |
|---|---|---|---|---|---|---|---|
| algebra | 0.18092 | 0.11453 | 0.42335 | 0.052334 | 0.0086009 | 0.096378 | 0.984 |
| analysis | 0.18784 | 0.11445 | 0.46899 | 0.054877 | 0.02738 | 0.1375 | 0.988 |
| biology | 0.63884 | 0.6201 | 0.37247 | 0.1588 | 1.4702e-11 | 0.022561 | 0.990 |
| chemistry | 0.50824 | 0.54954 | 0.34262 | 0.12219 | 2.15e-08 | 1.668e-14 | 0.990 |
| code | 0.28292 | 0.20851 | 0.41186 | 0.082102 | 0.13678 | 0.43453 | 0.986 |
| discrete | 0.2052 | 0.3295 | 0.26766 | 0.066181 | 4.7525e-12 | 9.8614e-05 | 0.992 |
| geometry | 0.20278 | 0.16029 | 0.35431 | 0.052369 | 1.3982e-12 | 0.0087202 | 0.987 |
| number_theory | 0.21726 | 0.16818 | 0.38339 | 0.055823 | 6.8172e-09 | 0.0070628 | 0.992 |
| physics | 0.54195 | 0.52847 | 0.33756 | 0.1111 | 0.0038941 | 9.3222e-17 | 0.987 |

# G   APPENDIX: IN-DOMAIN FITS

## G.0.1   IN-DOMAIN (SINGLE-DOMAIN EVIDENCE)

**Diminishing returns in** $k$**.** CE decreases near-monotonically with $k$ and follows the $1/(k+b)$ tail. At 0.5B the macro in-domain CE drops from $\approx 0.816$ at $k$=1 to $\approx 0.739$ at $k$=9 ($-9.5\%$); at 32B it drops from $\approx 0.493$ to $\approx 0.430$ ($-12.8\%$). Most gains arrive by $k \approx 5$ (math-like domains saturate sooner; science-like domains carry longer tails).

**Scaling with** $N$**.** Both the floor $L_\infty(N)$ and the tail amplitude $A(N)$ shrink with $N$; at fixed $k$=9, macro CE moves from $\approx 0.739$ (@0.5B) to $\approx 0.430$ (@32B), about $-42\%$. Per-domain joint fits (Average) give tight exponents (e.g., $\beta \in [0.33, 0.42]$) and high $R^2$ (Table 3).

**Where the details live.** Full per-domain parameters for Average/TA/TIES (incl. $b$), plus 72B forecasts, are reported in Appx. G. The 72B extrapolation is modest: at $k$=9 the median in-domain CE is forecast to drop another $\sim 6$–$10\%$ from 32B to 72B.

## G.1   MEAN CE: JOINT $(N, k)$ FITS

Table 3 reports the per-domain parameters of the joint law $L_{\infty,d}(N) = L_{*,d} + B_d N^{-\beta_d}$ and $A_d(N) = A_{0,d} N^{-\gamma_d}$ with the finite-$k$ offset $b_{0,d}$. All numbers come from weighted nonlinear least squares (weights $\propto k$). $R^2$ is computed on held-in $k$ grid points.

## G.2   VARIANCE: JOINT $(N, k)$ FITS BY METHOD

We fit $\mathrm{Var}[L_d \mid N, k] = V_{*,d} + B_d^{(\mathrm{var})} N^{-\beta_d^{(\mathrm{var})}} + \frac{A_{0,d}^{(\mathrm{var})} N^{-\gamma_d^{(\mathrm{var})}}}{k + b_{0,d}^{(\mathrm{var})}}$ with $V_{*,d} \approx 0$. Below we list parameters and $N$=72B predictions for $k \in \{1, 3, 5, 9\}$.

**Average.** Parameters:

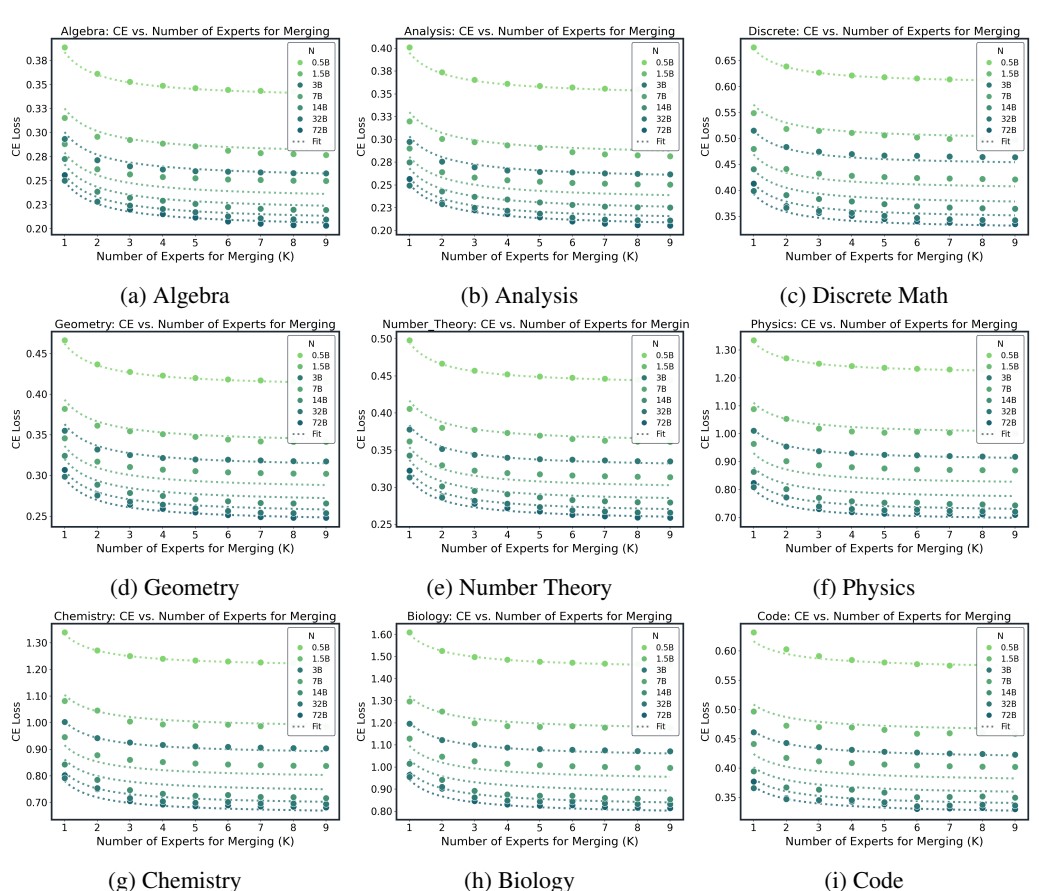

Figure 16: Merging Scaling Law with the Averaging Method

Table 4: Joint $(N, k)$ fit for TA (per-domain parameters).

| domain | Lstar | B | beta | A0 | gamma | b0 | R2 |
|---|---|---|---|---|---|---|---|
| algebra | 0.1912 | 0.10481 | 0.48613 | 0.031756 | 2.0539e-12 | 3.0949e-12 | 0.993 |
| analysis | 0.19859 | 0.10452 | 0.53812 | 0.032072 | 0.020433 | 8.3408e-12 | 0.994 |
| biology | 0.67453 | 0.6048 | 0.39438 | 0.10437 | 2.0948e-10 | 6.7298e-13 | 0.994 |
| chemistry | 0.5471 | 0.52754 | 0.3698 | 0.079144 | 1.4331e-15 | 5.2296e-13 | 0.994 |
| code | 0.29195 | 0.19378 | 0.4604 | 0.061683 | 0.11845 | 0.41132 | 0.993 |
| discrete | 0.26439 | 0.26479 | 0.36064 | 0.045787 | 5.5863e-10 | 1.153e-15 | 0.997 |
| geometry | 0.21888 | 0.14605 | 0.40757 | 0.034849 | 3.6096e-12 | 6.4127e-08 | 0.995 |
| number_theory | 0.23532 | 0.15 | 0.45207 | 0.037155 | 2.7958e-12 | 4.9617e-11 | 0.997 |
| physics | 0.57646 | 0.50399 | 0.36559 | 0.073691 | 1.0052e-07 | 5.0247e-15 | 0.993 |

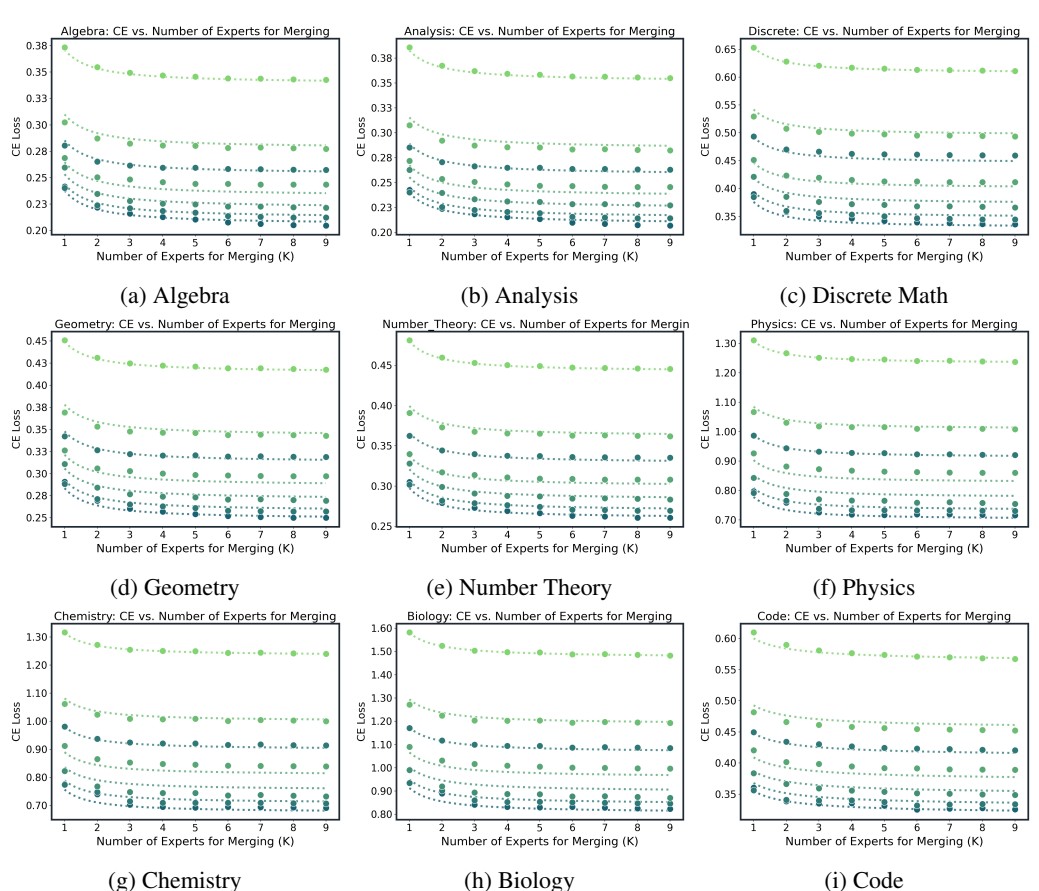

Figure 17: Merging Scaling Law with the TA Method

Table 5: Joint $(N, k)$ fit for TIES (per-domain parameters).

| domain | Lstar | B | beta | A0 | gamma | b0 | R2 |
|---|---|---|---|---|---|---|---|
| algebra | 0.18929 | 0.10752 | 0.46554 | 0.035371 | 0.011425 | 0.19757 | 0.975 |
| analysis | 0.19237 | 0.10912 | 0.50434 | 0.050902 | 0.016536 | 0.58856 | 0.980 |
| biology | 0.6077 | 0.60414 | 0.38384 | 0.37301 | 0.0080666 | 1.1634 | 0.990 |
| chemistry | 0.48423 | 0.53452 | 0.35563 | 0.30644 | 0.0069314 | 1.2221 | 0.989 |
| code | 0.26877 | 0.21391 | 0.38297 | 0.079839 | 0.11961 | 1.1999 | 0.986 |
| discrete | 0.22555 | 0.30917 | 0.28993 | 0.037062 | 2.9507e-10 | 1.2161e-08 | 0.988 |
| geometry | 0.21017 | 0.15222 | 0.38672 | 0.051128 | 5.5086e-10 | 0.39637 | 0.983 |
| number_theory | 0.22585 | 0.15954 | 0.41453 | 0.046348 | 1.0173e-09 | 0.27291 | 0.987 |
| physics | 0.53415 | 0.51524 | 0.34897 | 0.15923 | 0.00073252 | 0.50358 | 0.987 |

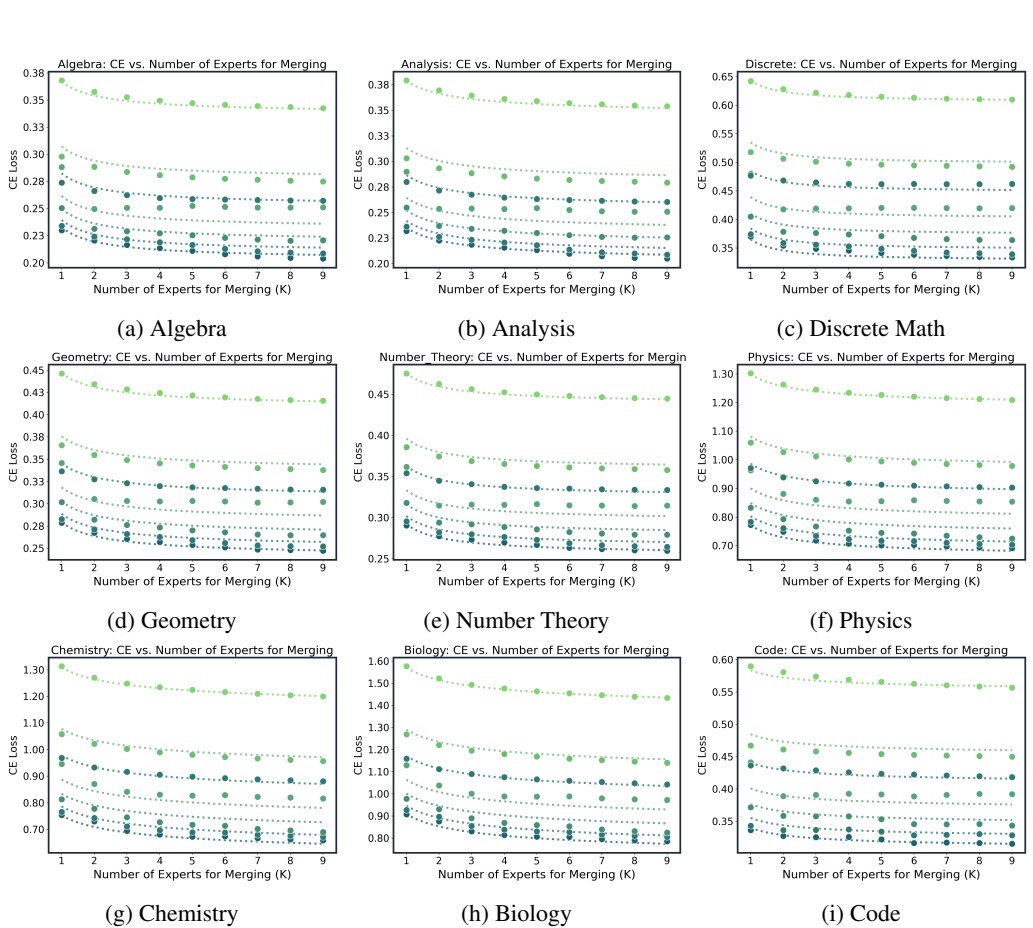

Figure 18: Merging Scaling Law with the TIES Method

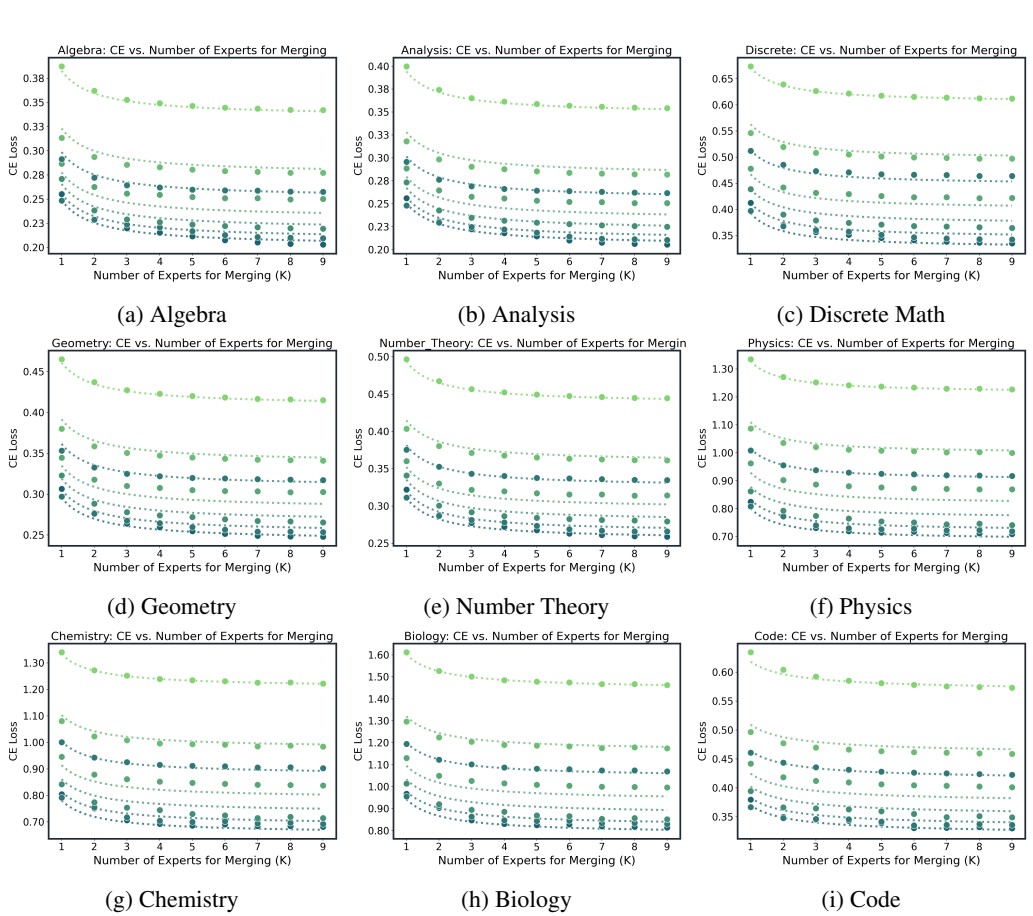

Figure 19: Merging Scaling Law with the DARE Method

Table 6: Variance fit parameters, Average.

| domain | ls | b | beta | a0 | gamma | b0 | r2 |
|---|---|---|---|---|---|---|---|
| algebra | 1.36e-18 | 5.57e-19 | 3 | 0.00159 | 0.0178 | 1.89e-11 | 0.844 |
| discrete | 5.80e-34 | 1.23e-22 | 1.94 | 0.00254 | 0.00496 | 2.29e-19 | 0.862 |
| analysis | 8.12e-29 | 2.49e-20 | 3 | 0.00146 | 0.0283 | 1.57e-12 | 0.861 |
| geometry | 2.30e-27 | 9.82e-22 | 2.1 | 0.00192 | 0.032 | 1.82e-16 | 0.851 |
| code | 3.25e-23 | 2.74e-22 | 0.067 | 0.0085 | 0.254 | 0.912 | 0.782 |
| number_theory | 2.92e-20 | 1.53e-23 | 2.08e-07 | 0.00248 | 0.0273 | 5.79e-13 | 0.86 |
| chemistry | 7.08e-31 | 9.84e-24 | 1.99 | 0.0393 | 0.127 | 0.205 | 0.891 |
| physics | 5.15e-21 | 1.09e-27 | 1.76 | 0.0229 | 0.119 | 0.125 | 0.903 |
| biology | 1.43e-19 | 1.77e-33 | 1.9 | 0.0556 | 0.151 | 0.272 | 0.879 |

Table 7: Variance at $N=72B$, Average.

| domain | $k=1$ | $k=3$ | $k=5$ | $k=9$ |
|---|---|---|---|---|
| algebra | 0.00148 | 4.93e-04 | 2.96e-04 | 1.64e-04 |
| analysis | 0.0013 | 4.32e-04 | 2.59e-04 | 1.44e-04 |
| biology | 0.0229 | 0.00889 | 0.00552 | 0.00314 |
| chemistry | 0.0189 | 0.00711 | 0.00438 | 0.00247 |
| code | 0.0015 | 7.34e-04 | 4.86e-04 | 2.90e-04 |
| discrete | 0.00249 | 8.30e-04 | 4.98e-04 | 2.77e-04 |
| geometry | 0.00168 | 5.59e-04 | 3.35e-04 | 1.86e-04 |
| number_theory | 0.0022 | 7.34e-04 | 4.41e-04 | 2.45e-04 |
| physics | 0.0122 | 0.00441 | 0.00269 | 0.00151 |

**TA.** Parameters:

Table 8: Variance fit parameters, TA.

| domain | ls | b | beta | a0 | gamma | b0 | r2 |
|---|---|---|---|---|---|---|---|
| algebra | 7.61e-31 | 9.32e-22 | 4.05e-08 | 0.00109 | 1.62e-06 | 3.41e-16 | 0.821 |
| discrete | 5.05e-36 | 9.52e-33 | 0.784 | 0.0017 | 1.14e-10 | 9.33e-23 | 0.819 |
| analysis | 2.63e-44 | 2.67e-26 | 0.585 | 9.98e-04 | 2.01e-06 | 7.55e-23 | 0.84 |
| geometry | 8.01e-30 | 4.73e-21 | 3.03e-08 | 0.00133 | 0.00682 | 5.56e-16 | 0.848 |
| code | 1.78e-33 | 2.30e-05 | 3 | 0.00525 | 0.206 | 0.514 | 0.816 |
| number_theory | 4.23e-45 | 1.72e-27 | 1.62 | 0.00171 | 5.25e-07 | 8.13e-21 | 0.845 |
| chemistry | 1.75e-24 | 3.78e-21 | 2.18 | 0.0266 | 0.112 | 1.88e-11 | 0.93 |
| physics | 6.05e-20 | 2.15e-21 | 0.812 | 0.0169 | 0.0995 | 1.70e-28 | 0.936 |
| biology | 2.27e-19 | 2.19e-23 | 1.91 | 0.0372 | 0.137 | 4.71e-12 | 0.924 |

Table 9: Variance at $N=72B$, TA.

| domain | $k=1$ | $k=3$ | $k=5$ | $k=9$ |
|---|---|---|---|---|
| algebra | 0.00109 | 3.64e-04 | 2.19e-04 | 1.21e-04 |
| analysis | 9.98e-04 | 3.33e-04 | 2.00e-04 | 1.11e-04 |
| biology | 0.0207 | 0.00691 | 0.00414 | 0.0023 |
| chemistry | 0.0165 | 0.0055 | 0.0033 | 0.00183 |
| code | 0.00144 | 6.18e-04 | 3.94e-04 | 2.28e-04 |
| discrete | 0.0017 | 5.66e-04 | 3.40e-04 | 1.89e-04 |
| geometry | 0.0013 | 4.32e-04 | 2.59e-04 | 1.44e-04 |
| number_theory | 0.00171 | 5.70e-04 | 3.42e-04 | 1.90e-04 |
| physics | 0.011 | 0.00368 | 0.00221 | 0.00123 |

**TIES.** Parameters:

Variance at $N=72B$:

Table 10: Variance fit parameters, TIES.

| domain | ls | b | beta | a0 | gamma | b0 | r2 |
|---|---|---|---|---|---|---|---|
| algebra | 1.35e-27 | 4.08e-32 | 0.863 | 7.48e-04 | 7.09e-11 | 1.94e-09 | 0.801 |
| discrete | 2.48e-34 | 9.61e-29 | 2.98 | 0.00117 | 6.34e-13 | 8.13e-11 | 0.736 |
| analysis | 2.03e-22 | 1.96e-26 | 4.28e-08 | 6.83e-04 | 1.33e-08 | 3.31e-09 | 0.822 |
| geometry | 2.56e-27 | 6.97e-26 | 0.63 | 8.99e-04 | 1.43e-11 | 2.86e-08 | 0.784 |
| code | 2.92e-12 | 1.76e-05 | 3 | 0.00424 | 0.12 | 1.09 | 0.752 |
| number_theory | 2.25e-24 | 3.61e-33 | 1.11e-07 | 0.00114 | 8.10e-12 | 6.51e-11 | 0.796 |
| chemistry | 3.25e-49 | 2.78e-28 | 0.781 | 0.0241 | 0.00132 | 0.446 | 0.816 |
| physics | 2.45e-19 | 5.40e-27 | 3 | 0.0137 | 0.00363 | 0.164 | 0.886 |
| biology | 1.65e-23 | 9.54e-22 | 0.00561 | 0.0344 | 0.0366 | 0.397 | 0.857 |

Table 11: Variance at $N{=}72$B, TIES.

| domain | $k{=}1$ | $k{=}3$ | $k{=}5$ | $k{=}9$ |
|---|---|---|---|---|
| algebra | 7.48e-04 | 2.49e-04 | 1.50e-04 | 8.31e-05 |
| analysis | 6.83e-04 | 2.28e-04 | 1.37e-04 | 7.59e-05 |
| biology | 0.0211 | 0.00867 | 0.00546 | 0.00313 |
| chemistry | 0.0165 | 0.00694 | 0.00439 | 0.00253 |
| code | 0.00121 | 6.20e-04 | 4.17e-04 | 2.51e-04 |
| discrete | 0.00117 | 3.91e-04 | 2.35e-04 | 1.30e-04 |
| geometry | 8.99e-04 | 3.00e-04 | 1.80e-04 | 9.99e-05 |
| number_theory | 0.00114 | 3.81e-04 | 2.28e-04 | 1.27e-04 |
| physics | 0.0116 | 0.00426 | 0.00261 | 0.00147 |

## H  APPENDIX: CROSS-DOMAIN FIT DETAILS

### H.0.1  CROSS-DOMAIN (POOLED EVIDENCE)

Macro-averaged CE over the nine domains follows the same floor+tail law $L_\infty(N) + A(N)/(k+b)$ as in-domain: curves are monotone with steep early gains and a short inverse tail; TA and TIES(0.5) show slightly faster early drops and gaps narrow with $k$. A single mild late-$k$ wobble appears for TIES(1) at 3B and is captured by adding a small bounded term (Appx. H). Scaling with model size mirrors the in-domain trend: at fixed large $k$ (e.g., $k{=}9$, Average), pooled CE improves substantially from small to large bases, reflecting a lower floor and a smaller tail amplitude. Merge-to-merge variance contracts approximately as $1/k$ with smaller amplitude at larger $N$, and TIES/TA exhibit slightly lower variance than Average at small $k$; details and extended forecasts (including 72B) appear in Appx. 6–11. For fitting, we regress the mean law per base size and method (setting the interference term to zero for monotone series) and fit the variance model unweighted; the scale parameter decreases with $N$ and the variance floor is small. Representative figures include Average@32B (mean improving from 0.5173 to 0.4530; variance shrinking from $9.8{\times}10^{-4}$ to $4.3{\times}10^{-5}$), TIES(0.5)@14B (mean $0.5286{\rightarrow}0.4599$), and the bounded late-$k$ uptick for TIES(1)@3B captured by a positive interference term.

### H.1  VARIANCE BEHAVIOR (BOTH SETTINGS)

*Merge-to-merge variability* contracts approximately as

$$\mathrm{Var}[L] \; \propto \; \frac{1}{k}, \tag{22}$$

with three robust regularities: (i) a near-$1/k$ drop that is already pronounced by small $k$ and flattens near a small floor (e.g., chemistry @0.5B, Average: $0.0385{\rightarrow}0.00108$ by $k{=}8$; algebra @0.5B: $2.28{\times}10^{-3}{\rightarrow}1.88{\times}10^{-5}$); (ii) *larger models are stabler*—at fixed $k$, variance is lower for larger $N$ (e.g., physics, Average, $k{=}1$: $0.0239{\rightarrow}0.0128$ from 0.5B to 32B); and (iii) *method ordering* at small $k$ typically satisfies TIES < TA < Average, with gaps vanishing as $k$ grows. We use Eq. equation 22 descriptively (fixing the log–log slope near $-1$), as heavier parameterization yields little additional predictive value while the simple form transfers cleanly across domains, sizes, and methods.

# I APPENDIX FOR CORE QUESTIONS

## I.1 PER-DOMAIN FITS, $k_\varepsilon$ EXAMPLES, AND ROBUSTNESS

**Specification.** For each domain $d$ we fit the joint law $\mathbb{E}[L_d \mid N, k] = L_{*,d} + B_d N^{-\beta_d} + \frac{A_{0,d} N^{-\gamma_d}}{k + b_d}$ by weighted nonlinear least squares (weights $\propto k$). We summarize *floors* via $L_{\infty,d}(N) = L_{*,d} + B_d N^{-\beta_d}$ (log–log regression) and *tails* via $A_d(N) = A_{0,d} N^{-\gamma_d}$.

**Per-domain parameters.** Floors exhibit tight power-law fits with exponents clustered in $[0.33, 0.42]$ and $R^2 \approx 0.98$–$0.99$ across domains. Tails are smaller and noisier; several domains are near-flat in $N$, while *code* shows the clearest decay. Table 12 lists an illustrative subset; full tables for all methods/domains appear in App. G.

| Domain | $\hat{b}$ | $\hat{A}_0$ | $\hat{\gamma}$ | $R^2(A)$ | $\hat{L}_*$ | $\hat{B}$ | $\hat{\beta}$ | $R^2(L)$ |
|---|---|---|---|---|---|---|---|---|
| algebra | 0.000 | 0.0460 | −0.004 | −0.002 | 0.1724 | 0.1248 | 0.379 | 0.983 |
| analysis | 0.000 | 0.0462 | +0.009 | +0.009 | 0.1793 | 0.1255 | 0.417 | 0.990 |
| biology | 0.125 | 0.1741 | −0.006 | +0.007 | 0.6227 | 0.6338 | 0.362 | 0.988 |
| chemistry | 0.075 | 0.1317 | −0.006 | +0.008 | 0.4924 | 0.5639 | 0.331 | 0.988 |
| code | 0.250 | 0.0682 | +0.115 | 0.556 | 0.2705 | 0.2238 | 0.378 | 0.986 |

Table 12: **Joint $(N, k)$ fits (subset, Average).** Floors are tight power laws; tails are small and domain-dependent (clearest decay in *code*).

**Macro evidence.** At $k{=}9$ (Average), macro CE decreases from 0.739 at 0.5B to 0.430 at 32B (−41.9%), consistent with a lower floor and a weakly shrinking tail.

$k_\varepsilon$ **examples ($\varepsilon{=}0.01$).** Using $k_\varepsilon(N, d) = \lceil A_d(N)/\varepsilon - b_d \rceil$ with $A_d(N) = A_{0,d} N^{-\gamma_d}$:

- *code*: $(\hat{b}, \hat{A}_0, \hat{\gamma}) = (0.25, 0.068, 0.115)$ gives $A(0.5\text{B}) \approx 0.074$, $A(32\text{B}) \approx 0.046$, hence $k_\varepsilon(0.5\text{B}) = 8$ and $k_\varepsilon(32\text{B}) = 5$.
- *biology*: $(0.125, 0.174, -0.006)$ (near-flat tail) gives $A(0.5\text{B}) \approx 0.173$, $A(32\text{B}) \approx 0.177$, so $k_\varepsilon$ stays $\approx 18$, yet CE still falls with $N$ due to the lower floor.

**Robustness.** Altering weights (uniform vs. $\propto k$) or censoring tiny high-$k$ points barely changes floor exponents. For extrapolation (e.g., 72B), floors should be treated as the dominant $N$-driver, with tails weakly decreasing/flat; $k_\varepsilon$ then gives a practical "experts-to-saturation" budget.

**Plot inventory.** (i) Macro CE@$k{=}9$ vs. $N$ (log–log) with power-law fit; (ii) two representative floor curves $L_{\infty,d}(N)$ (e.g., *algebra* vs. *biology*); (iii) optional $A_d(N)$ vs. $N$ to visualize tail trends.

## I.2 MOST OF THE GAIN COMES FROM THE FIRST FEW EXPERTS

We quantify the "return" from merging $k$ experts at a fixed $(N, d)$ by the fraction of realized improvement $R(N, d, k)$ computed from the monotone envelope of the measured CE curve (see App. I.2). We summarize two views in Fig. 20: *(left)* the median $R(k)$ over all $(N, d)$ with an IQR band; *(right)* a heatmap of the smallest $k$ that reaches a target return (here 90%). As shown in Fig. 20, most of the improvement arrives early: the median curve crosses 85% by $k{=}5$ and 90% by $k{=}6$, and the $k_{90}$ heatmap concentrates in $\{5, 6\}$ across domains and model sizes. Math-like domains tend to saturate slightly earlier, while science-like domains keep a longer—but still flattening—tail. This "early elbow" follows directly from the unified law $L(N, k) = L_\infty(N) + A(N)/(k+b)$: the marginal gain $\Delta_k \approx A(N)/[(k+b)(k+1+b)]$ decays roughly as $k^{-2}$, so returns diminish sharply beyond the first few experts.

## I.3 ADDITIONAL PLOTS, TABLES, AND DETAILS

For each method (Average, TA, TIES, DARE) and size $N$, we fit $L(N, k) = L_\infty(N) + A(N)/(k+b)$ by weighted least squares (weights $\propto k$) on the pooled CE, averaging over randomized expert orders;

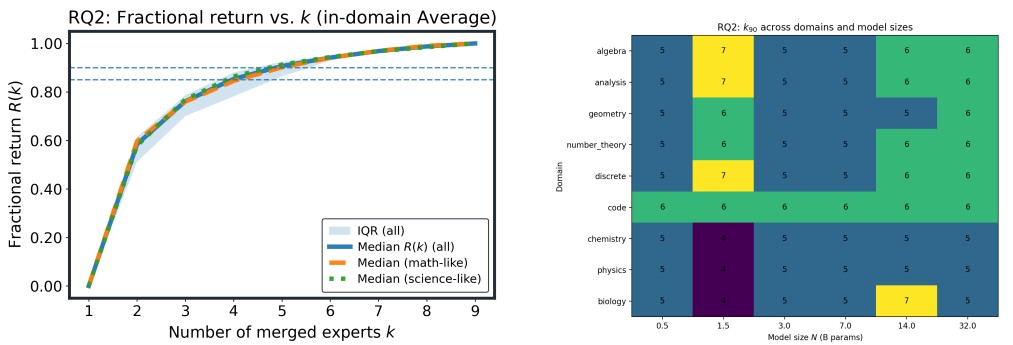

Figure 20: **Most of the gain comes from the first few experts.** *Left:* Median fractional return $R(k)$ with IQR band; $k=5$ and $k=6$ cross the $85\%/90\%$ thresholds, respectively. *Right:* $k_{90}$ across domains and sizes concentrates at $k \in \{5, 6\}$ (about half to two-thirds of this 9-expert pool ($5/9 \approx 56\%$)).

only TIES with the strongest nonlinearity requires an extra bounded term $+D(N)\frac{k}{k+q}$, with small $D$ and stable $q$. We release per-method parameter tables $\{b, A_0, \gamma, L_*, B, \beta\}$ and residual plots versus $k$; companion figures reproduce Fig. 5a for all methods and provide fractional-return curves $R(k)$ and $k_{90}$ heatmaps across $N$. Headline patterns match the main text: most pooled improvement is realized by $k \le 6$, method differences narrow with $k$, and scaling in $N$ lowers both the pooled floor and the tail.

| Method | Qwen-0.5b | Qwen-1.5b | Qwen-3b | Qwen-7b | Qwen-14b | Qwen-32b | Qwen-72b | SUM(GPUh) |
|--------|-----------|-----------|---------|---------|----------|----------|----------|-----------|
| TA | 32s | 68s | 129s | 244s | 383s | 777s | 2686s | 1.20 |
| AVG | 48s | 73s | 168s | 265s | 421s | 843s | 2280s | 1.14 |
| Dare | 30s | 72s | 102s | 251s | 420s | 796s | 2360s | 1.112 |
| Ties | 43s | 77s | 136s | 270s | 507s | 961s | 2967s | 1.38 |

Table 13: The GPU hours when merging 9 domains for different size of models.

| Model Size | Model Name |
|------------|-----------|
| 3B | theprint/ReWiz-Llama-3.2-3B (theprint, 2025) |
| | NousResearch/Hermes-3-Llama-3.2-3B (NousResearch, 2025b) |
| | MergeBench/Llama-3.2-3B-Instruct_coding (MergeBench, 2025a) |
| | MergeBench/Llama-3.2-3B-Instruct_math (MergeBench, 2025c) |
| | MergeBench/Llama-3.2-3B-Instruct_multilingual (MergeBench, 2025d) |
| | meta-llama/Llama-3.2-3B-Instruct (meta llama, 2025b) |
| | ValiantLabs/Llama3.2-3B-ShiningValiant2 (ValiantLabs, 2025) |
| | MergeBench/Llama-3.2-3B-Instruct_safety (MergeBench, 2025e) |
| | MergeBench/Llama-3.2-3B-Instruct_instruction (MergeBench, 2025b) |
| 8B | Undi95/Meta-Llama-3-8B-Instruct-hf (Undi95, 2025b) |
| | Undi95/Llama-3-LewdPlay-8B-evo (Undi95, 2025a) |
| | jondurbin/bagel-8b-v1.0 (jondurbin, 2025) |
| | Weyaxi/Einstein-v6.1-Llama3-8B (Weyaxi, 2025) |
| | VAGOsolutions/Llama-3-SauerkrautLM-8b-Instruct (VAGOsolutions, 2025) |
| | aaditya/OpenBioLLM-Llama3-8B (aaditya, 2025) |
| | Dampfinchen/Llama-3-8B-Ultra-Instruct (Dampfinchen, 2025) |
| | NousResearch/Hermes-3-Llama-3.1-8B (NousResearch, 2025a) |
| | meta-llama/Llama-3.1-8B-Instruct (meta llama, 2025a) |

Table 14: List of open source models on Huggingface.

EXTENDED EVIDENCE

*Small-k mean gaps vs. Average (relative %).* We report $(\text{Avg} - \text{Method})/\text{Avg}$ at $k{=}2$ and the signed gap at $k{=}9$ (lower is better).

| Method | 0.5B | | 14B | |
|---|---|---|---|---|
| | $k{=}2$ | $k{=}9$ | $k{=}2$ | $k{=}9$ |
| TA ($\lambda{=}0.8$) | 0.9% | +0.7% (worse) | 0.6% | +1.4% (worse) |
| TIES ($\lambda{=}0.5$) | 0.9% | −1.1% (better) | 0.6% | −2.2% (better) |

| Method | 32B | |
|---|---|---|
| | $k{=}2$ | $k{=}9$ |
| TA ($\lambda{=}0.8$) | 1.2% | +1.2% (worse) |
| TIES ($\lambda{=}0.5$) | 1.7% | −2.3% (better) |

*Summary.* The method "bandwidth" is consistently narrow across scales: at small $k$, TA and TIES(0.5) are modestly better than Average (typically $1\%{\sim}2\%$ at $k{=}2$), and by $k{=}9$ gaps further compress (TIES(0.5) usually retains a $\approx 1\%{\sim}2\%$ edge, TA is near-tied or slightly worse). Variance shows the same convergence: at $N{=}32\text{B}$, $k{=}2$ the across-merge variance is $9.67{\times}10^{-4}$ (Average), $7.83{\times}10^{-4}$ (TA, $-19\%$), and $6.50{\times}10^{-4}$ (TIES 0.5, $-33\%$); by $k{=}8$ all methods are around $(3-4) \times 10^{-5}$. At $N{=}0.5\text{B}$, $k{=}2$ the pattern holds (Average $1.73{\times}10^{-3}$, TA $-26\%$, TIES 0.5 $-44\%$). A mild late-$k$ uptick for TIES($\lambda{=}1$) at 3B is captured by a small bounded term $D\,\frac{k}{k+q}$; using $\lambda{=}0.5$ restores the standard monotone $1/(k+b)$ tail. Together, these results support the main-text claim: method differences are second-order and shrink quickly with $k$.

# J APPENDIX: DO DOWNSTREAM METRICS FOLLOW THE SAME TREND?

## J.1 OVERALL RESULTS

**Setup.** To test whether the CE scaling law is reflected in end-task quality, we train and use the different merged checkpoints from §3.1 to demonstrate the trend on different backbones. In this section, we post-train on Llama-based and Gemma-based models and evaluate them on a diverse suite of downstream benchmarks, including math, general reasoning, multilingual, coding, and safety. For each backbone and merge method, we:

(i) evaluate all expert subsets for $k{\in}\{1,\ldots,5\}$,

(ii) normalise each metric so that larger is better,

(iii) report the *mean accuracy* obtained by first averaging across tasks and then across all expert subsets at fixed $k$.

Table 15 summarises the resulting trend for three backbones (LLaMA-3.1 8B, LLaMA-3.2 3B, Gemma-2 2B) and two merge rules (Task Arithmetic and TIES).

**Findings.** Across all settings, the aggregated downstream performance improves as we increase the number of merged experts and then saturates, mirroring the floor+tail behaviour observed for CE. For LLaMA-3.1 8B with Task Arithmetic, mean accuracy rises steadily from 0.41 at $k{=}1$ to 0.47 at $k{=}5$, with rapidly diminishing gains after $k{\approx}3$. LLaMA-3.2 3B shows the same qualitative pattern but with a shallower tail: accuracy improves from 0.38 at $k{=}2$ to about 0.39 at $k{=}4$ and then slightly fluctuates within $\pm 0.002$, which we attribute to benchmark variance rather than a systematic degradation. Gemma-2 2B (available only for $k{\geq}2$) and TIES on LLaMA-3.1 8B both display monotone or nearly monotone gains up to $k{\approx}4$, followed by a clear plateau. Taken together, these results indicate that our merging scaling law, defined on the *expected* CE over expert subsets, is qualitatively aligned with the trend of averaged downstream metrics: most of the benefit is captured with a small number of experts, and additional experts yield diminishing returns. A more refined characterisation of when CE and task accuracy may diverge, and how to predict in advance whether a particular merge will work on a given task, is an interesting direction for future work.

| Backbone | Method | $k{=}1$ | $k{=}2$ | $k{=}3$ | $k{=}4$ | $k{=}5$ |
|---|---|---|---|---|---|---|
| LLaMA-3.1 8B | TA | 0.411 | 0.443 | 0.456 | 0.462 | 0.469 |
| LLaMA-3.2 3B | TA | 0.375 | 0.386 | 0.388 | 0.389 | 0.388 |
| Gemma-2 2B | TA | 0.492 | 0.503 | 0.506 | 0.507 | 0.507 |
| LLaMA-3.1 8B | TIES | 0.388 | 0.414 | 0.426 | 0.436 | 0.436 |

Table 15: Mean downstream accuracy vs. number of merged experts $k$. Each entry is averaged over all benchmarks and all expert subsets at fixed $k$ for the given backbone and merge methods (higher is better).

| folder | math_500 | gsm8k | ifeval | arc | hellaswag | mmlu | mbppplus | humanevalplus | wildguard_micro_harm | harmbench_micro_asr | wildguard_rta | harmbench_rta |
|---|---|---|---|---|---|---|---|---|---|---|---|---|
| 1 | 0.138 | 0.378 | 0.244 | 0.406 | 0.493 | 0.500 | 0.558 | 0.543 | 0.471 | 0.669 | 0.529 | 0.331 |
| 2 | 0.468 | 0.835 | 0.102 | 0.341 | 0.466 | 0.451 | 0.370 | 0.256 | 0.565 | 0.634 | 0.435 | 0.366 |
| 3 | 0.150 | 0.271 | 0.545 | 0.412 | 0.494 | 0.501 | 0.526 | 0.372 | 0.506 | 0.550 | 0.494 | 0.450 |
| 4 | 0.016 | 0.148 | 0.198 | 0.357 | 0.467 | 0.434 | 0.413 | 0.226 | 0.228 | 0.184 | 0.772 | 0.816 |
| 5 | 0.128 | 0.215 | 0.165 | 0.432 | 0.496 | 0.540 | 0.540 | 0.085 | 0.485 | 0.584 | 0.515 | 0.416 |
| 1-2 | 0.406 | 0.804 | 0.189 | 0.380 | 0.492 | 0.499 | 0.526 | 0.494 | 0.558 | 0.650 | 0.442 | 0.350 |
| 1-3 | 0.168 | 0.539 | 0.368 | 0.440 | 0.510 | 0.527 | 0.563 | 0.549 | 0.523 | 0.719 | 0.477 | 0.281 |
| 1-4 | 0.116 | 0.444 | 0.189 | 0.393 | 0.493 | 0.499 | 0.534 | 0.506 | 0.191 | 0.138 | 0.809 | 0.863 |
| 1-5 | 0.148 | 0.493 | 0.231 | 0.442 | 0.509 | 0.537 | 0.550 | 0.396 | 0.547 | 0.675 | 0.453 | 0.325 |
| 2-3 | 0.442 | 0.819 | 0.268 | 0.378 | 0.495 | 0.497 | 0.511 | 0.354 | 0.557 | 0.653 | 0.443 | 0.347 |
| 2-4 | 0.348 | 0.732 | 0.190 | 0.365 | 0.483 | 0.475 | 0.463 | 0.323 | 0.350 | 0.459 | 0.650 | 0.541 |
| 2-5 | 0.402 | 0.806 | 0.131 | 0.378 | 0.490 | 0.513 | 0.511 | 0.329 | 0.537 | 0.609 | 0.463 | 0.391 |
| 3-4 | 0.010 | 0.219 | 0.290 | 0.395 | 0.490 | 0.480 | 0.537 | 0.317 | 0.200 | 0.163 | 0.800 | 0.838 |
| 3-5 | 0.146 | 0.543 | 0.392 | 0.442 | 0.504 | 0.542 | 0.534 | 0.311 | 0.509 | 0.606 | 0.491 | 0.394 |
| 4-5 | 0.042 | 0.118 | 0.146 | 0.402 | 0.493 | 0.500 | 0.503 | 0.165 | 0.218 | 0.244 | 0.782 | 0.756 |
| 1-2-3 | 0.388 | 0.767 | 0.244 | 0.397 | 0.503 | 0.517 | 0.561 | 0.451 | 0.558 | 0.653 | 0.442 | 0.347 |
| 1-2-4 | 0.316 | 0.704 | 0.202 | 0.379 | 0.495 | 0.503 | 0.537 | 0.451 | 0.343 | 0.463 | 0.657 | 0.538 |
| 1-2-5 | 0.368 | 0.753 | 0.177 | 0.401 | 0.501 | 0.522 | 0.563 | 0.451 | 0.555 | 0.672 | 0.445 | 0.328 |
| 1-3-4 | 0.116 | 0.510 | 0.263 | 0.407 | 0.504 | 0.515 | 0.537 | 0.494 | 0.216 | 0.166 | 0.784 | 0.834 |
| 1-3-5 | 0.176 | 0.503 | 0.333 | 0.447 | 0.512 | 0.542 | 0.563 | 0.433 | 0.565 | 0.725 | 0.435 | 0.275 |
| 1-4-5 | 0.114 | 0.436 | 0.211 | 0.407 | 0.504 | 0.524 | 0.556 | 0.348 | 0.248 | 0.203 | 0.752 | 0.797 |
| 2-3-4 | 0.318 | 0.713 | 0.255 | 0.385 | 0.494 | 0.494 | 0.505 | 0.390 | 0.356 | 0.463 | 0.644 | 0.538 |
| 2-3-5 | 0.388 | 0.759 | 0.237 | 0.400 | 0.500 | 0.526 | 0.529 | 0.372 | 0.535 | 0.647 | 0.465 | 0.353 |
| 2-4-5 | 0.310 | 0.688 | 0.174 | 0.391 | 0.494 | 0.509 | 0.513 | 0.335 | 0.405 | 0.497 | 0.595 | 0.503 |
| 3-4-5 | 0.050 | 0.354 | 0.261 | 0.411 | 0.500 | 0.512 | 0.540 | 0.262 | 0.247 | 0.316 | 0.753 | 0.684 |
| 1-2-3-4 | 0.300 | 0.692 | 0.240 | 0.395 | 0.502 | 0.515 | 0.532 | 0.463 | 0.360 | 0.528 | 0.640 | 0.472 |
| 1-2-3-5 | 0.324 | 0.722 | 0.229 | 0.413 | 0.507 | 0.532 | 0.566 | 0.451 | 0.567 | 0.663 | 0.433 | 0.338 |
| 1-2-4-5 | 0.300 | 0.681 | 0.203 | 0.396 | 0.502 | 0.521 | 0.558 | 0.402 | 0.371 | 0.538 | 0.629 | 0.463 |
| 1-3-4-5 | 0.122 | 0.468 | 0.277 | 0.416 | 0.508 | 0.532 | 0.556 | 0.360 | 0.268 | 0.256 | 0.732 | 0.744 |
| 2-3-4-5 | 0.292 | 0.674 | 0.240 | 0.400 | 0.499 | 0.516 | 0.534 | 0.378 | 0.399 | 0.513 | 0.601 | 0.488 |
| 1-2-3-4-5 | 0.272 | 0.679 | 0.251 | 0.408 | 0.505 | 0.528 | 0.569 | 0.421 | 0.386 | 0.572 | 0.614 | 0.428 |
| overall_k1 | 0.180 | 0.369 | 0.251 | 0.390 | 0.483 | 0.485 | 0.481 | 0.296 | 0.451 | 0.524 | 0.549 | 0.476 |
| overall_k2 | 0.223 | 0.552 | 0.239 | 0.402 | 0.496 | 0.507 | 0.523 | 0.374 | 0.419 | 0.492 | 0.581 | 0.508 |
| overall_k3 | 0.254 | 0.619 | 0.235 | 0.403 | 0.501 | 0.516 | 0.540 | 0.399 | 0.403 | 0.480 | 0.597 | 0.520 |
| overall_k4 | 0.268 | 0.647 | 0.238 | 0.404 | 0.504 | 0.523 | 0.549 | 0.411 | 0.393 | 0.499 | 0.607 | 0.501 |
| overall_k5 | 0.272 | 0.679 | 0.251 | 0.408 | 0.505 | 0.528 | 0.569 | 0.421 | 0.386 | 0.572 | 0.614 | 0.428 |

Table 16: Full downstream results for LLaMA 3.1 8B with TA merging across five domain experts.

## J.2 DETAILED CASES

**Setup.** We report full downstream results for three backbones and two merge rules. For all experiments, we start from a common base model and five domain-specialised experts: (1) *math* (MATH/GSM8K style), (2) *code* (MBPP/HumanEval style), (3) *multilingual* (general language understanding across languages), (4) *safety* (safety and refusal tuning), and (5) *instruction-following* (generic chat/IFEval). In the tables, the column `folder` encodes which experts are merged: for example, `1, 2, ..., 5` are single experts; `1-2` or `3-5` are 2-model merges of the corresponding experts; and `1-2-3-4-5` is the merge of all five experts. Tables 16 and 17 use *Task Arithmetic (TA)* as the merge rule on LLaMA 3.1 8B and LLaMA 3.2 3B, respectively. Table 18 uses TA on Gemma 2 2B. Table 19 uses *TIES* merging on LLaMA 3.1 8B. We evaluate on a heterogeneous benchmark suite covering math reasoning (math_500, GSM8K), code (mbppplus, humanevalplus), general QA and language understanding (IFEval, ARC, HellaSwag, MMLU and multilingual_overall), and safety. Rows `overall_k1`–`overall_k5` in the TA tables aggregate over all expert subsets with fixed $k$ and then average across benchmarks. These aggregated means are the values used in the main-text plots.

## K SCALING BEHAVIOUR WITH 16 DOMAINS

**Setup.** We extend the cross-domain scaling experiment to a larger 16-domain pool on the LLaMA3-3B-Instruct backbone. Starting from the original 9 domains (*algebra, analysis, geometry, discrete, number_theory, code, chemistry, physics, biology*), we add 7 additional experts fine-tuned on *Japanese, medical, house-arrangement, Korean, emotion, elementary school mathematics*, and *Java code* tasks.

| folder | math_500 | gsm8k | ifeval | arc | hellaswag | mmlu | multilingual_overall | mbppplus | humanevalplus | wildguard_micro_harm | harmbench_micro_asr | wildguard_rta | harmbench_rta |
|---|---|---|---|---|---|---|---|---|---|---|---|---|---|
| 1 | 0.048 | 0.256 | 0.177 | 0.349 | 0.437 | 0.458 | 0.415 | 0.466 | 0.415 | 0.607 | 0.756 | 0.393 | 0.244 |
| 2 | 0.274 | 0.682 | 0.196 | 0.329 | 0.423 | 0.444 | 0.398 | 0.386 | 0.287 | 0.625 | 0.656 | 0.375 | 0.344 |
| 3 | 0.070 | 0.194 | 0.351 | 0.350 | 0.431 | 0.425 | 0.402 | 0.410 | 0.293 | 0.567 | 0.663 | 0.433 | 0.338 |
| 4 | 0.002 | 0.036 | 0.161 | 0.330 | 0.429 | 0.454 | 0.404 | 0.423 | 0.220 | 0.146 | 0.109 | 0.854 | 0.891 |
| 5 | 0.016 | 0.061 | 0.187 | 0.356 | 0.429 | 0.441 | 0.409 | 0.421 | 0.213 | 0.697 | 0.784 | 0.303 | 0.216 |
| 1-2 | 0.192 | 0.545 | 0.189 | 0.347 | 0.432 | 0.457 | 0.412 | 0.442 | 0.335 | 0.637 | 0.713 | 0.363 | 0.288 |
| 1-3 | 0.096 | 0.261 | 0.340 | 0.361 | 0.442 | 0.455 | 0.420 | 0.429 | 0.360 | 0.595 | 0.691 | 0.405 | 0.309 |
| 1-4 | 0.076 | 0.240 | 0.214 | 0.344 | 0.435 | 0.460 | 0.413 | 0.455 | 0.305 | 0.427 | 0.513 | 0.573 | 0.488 |
| 1-5 | 0.058 | 0.217 | 0.194 | 0.359 | 0.437 | 0.459 | 0.418 | 0.471 | 0.372 | 0.690 | 0.713 | 0.310 | 0.288 |
| 2-3 | 0.184 | 0.519 | 0.263 | 0.350 | 0.436 | 0.451 | 0.413 | 0.397 | 0.305 | 0.619 | 0.681 | 0.381 | 0.319 |
| 2-4 | 0.124 | 0.519 | 0.185 | 0.336 | 0.430 | 0.455 | 0.407 | 0.434 | 0.262 | 0.526 | 0.631 | 0.474 | 0.369 |
| 2-5 | 0.098 | 0.434 | 0.185 | 0.351 | 0.430 | 0.456 | 0.412 | 0.426 | 0.280 | 0.705 | 0.709 | 0.295 | 0.291 |
| 3-4 | 0.066 | 0.110 | 0.316 | 0.347 | 0.439 | 0.451 | 0.412 | 0.402 | 0.274 | 0.294 | 0.375 | 0.706 | 0.625 |
| 3-5 | 0.046 | 0.064 | 0.266 | 0.361 | 0.439 | 0.447 | 0.415 | 0.426 | 0.280 | 0.654 | 0.663 | 0.346 | 0.338 |
| 4-5 | 0.020 | 0.104 | 0.218 | 0.349 | 0.432 | 0.455 | 0.412 | 0.431 | 0.195 | 0.509 | 0.513 | 0.491 | 0.488 |
| 1-2-3 | 0.160 | 0.467 | 0.257 | 0.357 | 0.439 | 0.457 | 0.418 | 0.447 | 0.372 | 0.623 | 0.731 | 0.377 | 0.269 |
| 1-2-4 | 0.126 | 0.447 | 0.194 | 0.344 | 0.434 | 0.461 | 0.413 | 0.429 | 0.305 | 0.565 | 0.675 | 0.435 | 0.325 |
| 1-2-5 | 0.096 | 0.417 | 0.179 | 0.353 | 0.434 | 0.459 | 0.415 | 0.423 | 0.335 | 0.668 | 0.700 | 0.332 | 0.300 |
| 1-3-4 | 0.068 | 0.246 | 0.279 | 0.354 | 0.440 | 0.457 | 0.417 | 0.434 | 0.317 | 0.442 | 0.528 | 0.558 | 0.472 |
| 1-3-5 | 0.052 | 0.218 | 0.259 | 0.360 | 0.442 | 0.457 | 0.419 | 0.434 | 0.323 | 0.670 | 0.653 | 0.330 | 0.347 |
| 1-4-5 | 0.050 | 0.207 | 0.216 | 0.351 | 0.436 | 0.459 | 0.416 | 0.442 | 0.262 | 0.590 | 0.650 | 0.410 | 0.350 |
| 2-3-4 | 0.134 | 0.399 | 0.251 | 0.347 | 0.436 | 0.455 | 0.413 | 0.418 | 0.280 | 0.489 | 0.519 | 0.511 | 0.481 |
| 2-3-5 | 0.094 | 0.340 | 0.248 | 0.359 | 0.438 | 0.455 | 0.417 | 0.402 | 0.329 | 0.684 | 0.669 | 0.316 | 0.331 |
| 2-4-5 | 0.078 | 0.352 | 0.203 | 0.345 | 0.432 | 0.459 | 0.412 | 0.423 | 0.287 | 0.630 | 0.706 | 0.370 | 0.294 |
| 3-4-5 | 0.038 | 0.080 | 0.285 | 0.353 | 0.438 | 0.454 | 0.415 | 0.418 | 0.244 | 0.475 | 0.525 | 0.525 | 0.475 |
| 1-2-3-4 | 0.106 | 0.409 | 0.242 | 0.350 | 0.438 | 0.459 | 0.416 | 0.442 | 0.323 | 0.542 | 0.594 | 0.458 | 0.406 |
| 1-2-3-5 | 0.090 | 0.394 | 0.227 | 0.360 | 0.439 | 0.459 | 0.419 | 0.439 | 0.335 | 0.666 | 0.666 | 0.334 | 0.334 |
| 1-2-4-5 | 0.086 | 0.372 | 0.214 | 0.349 | 0.435 | 0.461 | 0.415 | 0.452 | 0.317 | 0.636 | 0.688 | 0.364 | 0.313 |
| 1-3-4-5 | 0.040 | 0.204 | 0.250 | 0.355 | 0.440 | 0.458 | 0.418 | 0.431 | 0.329 | 0.563 | 0.597 | 0.437 | 0.403 |
| 2-3-4-5 | 0.084 | 0.272 | 0.226 | 0.352 | 0.437 | 0.457 | 0.416 | 0.405 | 0.280 | 0.598 | 0.597 | 0.402 | 0.403 |
| 1-2-3-4-5 | 0.070 | 0.334 | 0.231 | 0.356 | 0.438 | 0.459 | 0.418 | 0.423 | 0.305 | 0.602 | 0.644 | 0.398 | 0.356 |
| overall_k1 | 0.082 | 0.246 | 0.214 | 0.343 | 0.430 | 0.444 | 0.406 | 0.421 | 0.285 | 0.528 | 0.594 | 0.472 | 0.406 |
| overall_k2 | 0.096 | 0.301 | 0.237 | 0.350 | 0.435 | 0.455 | 0.413 | 0.431 | 0.297 | 0.566 | 0.620 | 0.434 | 0.380 |
| overall_k3 | 0.090 | 0.317 | 0.237 | 0.352 | 0.437 | 0.457 | 0.416 | 0.427 | 0.305 | 0.584 | 0.636 | 0.416 | 0.364 |
| overall_k4 | 0.081 | 0.330 | 0.232 | 0.353 | 0.438 | 0.459 | 0.417 | 0.434 | 0.317 | 0.601 | 0.628 | 0.399 | 0.372 |
| overall_k5 | 0.070 | 0.334 | 0.231 | 0.356 | 0.438 | 0.459 | 0.418 | 0.423 | 0.305 | 0.602 | 0.644 | 0.398 | 0.356 |

Table 17: Full downstream results for LLaMA 3.2 3B with TA merging across five domain experts.

| folder | math_500 | gsm8k | ifeval | arc | hellaswag | mmlu | mbppplus | humanevalplus | wildguard_rta | harmbench_rta |
|---|---|---|---|---|---|---|---|---|---|---|
| 1-2 | 0.288 | 0.569 | 0.417 | 0.372 | 0.431 | 0.488 | 0.437 | 0.366 | 0.826 | 0.806 |
| 1-3 | 0.254 | 0.566 | 0.457 | 0.378 | 0.432 | 0.488 | 0.447 | 0.348 | 0.824 | 0.859 |
| 1-4 | 0.240 | 0.560 | 0.440 | 0.378 | 0.431 | 0.488 | 0.437 | 0.335 | 0.837 | 0.884 |
| 1-5 | 0.264 | 0.531 | 0.463 | 0.386 | 0.437 | 0.490 | 0.447 | 0.354 | 0.817 | 0.828 |
| 2-4 | 0.290 | 0.591 | 0.451 | 0.373 | 0.432 | 0.487 | 0.442 | 0.335 | 0.797 | 0.825 |
| 3-4 | 0.302 | 0.590 | 0.421 | 0.373 | 0.431 | 0.487 | 0.437 | 0.354 | 0.814 | 0.841 |
| 2-3 | 0.276 | 0.569 | 0.442 | 0.381 | 0.434 | 0.488 | 0.439 | 0.348 | 0.812 | 0.834 |
| 2-5 | 0.258 | 0.591 | 0.449 | 0.376 | 0.432 | 0.487 | 0.434 | 0.305 | 0.828 | 0.856 |
| 3-5 | 0.266 | 0.557 | 0.468 | 0.386 | 0.437 | 0.490 | 0.431 | 0.329 | 0.817 | 0.856 |
| 4-5 | 0.252 | 0.550 | 0.451 | 0.386 | 0.437 | 0.489 | 0.450 | 0.341 | 0.840 | 0.866 |
| 1-2-4 | 0.292 | 0.558 | 0.438 | 0.374 | 0.433 | 0.489 | 0.442 | 0.366 | 0.817 | 0.831 |
| 1-2-3 | 0.298 | 0.544 | 0.407 | 0.371 | 0.432 | 0.488 | 0.452 | 0.372 | 0.836 | 0.847 |
| 1-2-5 | 0.286 | 0.536 | 0.438 | 0.384 | 0.435 | 0.489 | 0.434 | 0.378 | 0.820 | 0.819 |
| 1-3-4 | 0.268 | 0.553 | 0.453 | 0.377 | 0.433 | 0.487 | 0.455 | 0.360 | 0.842 | 0.897 |
| 1-3-5 | 0.266 | 0.538 | 0.444 | 0.388 | 0.439 | 0.490 | 0.434 | 0.348 | 0.809 | 0.863 |
| 1-4-5 | 0.264 | 0.522 | 0.449 | 0.385 | 0.438 | 0.489 | 0.434 | 0.360 | 0.833 | 0.888 |
| 2-3-4 | 0.298 | 0.575 | 0.451 | 0.374 | 0.433 | 0.487 | 0.447 | 0.348 | 0.813 | 0.863 |
| 2-3-5 | 0.298 | 0.557 | 0.444 | 0.382 | 0.437 | 0.489 | 0.450 | 0.354 | 0.802 | 0.822 |
| 2-4-5 | 0.260 | 0.560 | 0.449 | 0.381 | 0.437 | 0.489 | 0.444 | 0.348 | 0.818 | 0.847 |
| 3-4-5 | 0.256 | 0.538 | 0.470 | 0.383 | 0.438 | 0.490 | 0.452 | 0.354 | 0.828 | 0.888 |
| 1-2-3-4 | 0.282 | 0.557 | 0.423 | 0.375 | 0.433 | 0.488 | 0.437 | 0.360 | 0.833 | 0.875 |
| 1-2-3-5 | 0.286 | 0.543 | 0.458 | 0.384 | 0.438 | 0.489 | 0.452 | 0.335 | 0.809 | 0.834 |
| 1-2-4-5 | 0.282 | 0.547 | 0.438 | 0.380 | 0.437 | 0.490 | 0.444 | 0.354 | 0.828 | 0.856 |
| 1-3-4-5 | 0.280 | 0.528 | 0.442 | 0.388 | 0.439 | 0.489 | 0.444 | 0.341 | 0.838 | 0.903 |
| 2-3-4-5 | 0.288 | 0.563 | 0.470 | 0.380 | 0.439 | 0.489 | 0.447 | 0.354 | 0.833 | 0.863 |
| 1-2-3-4-5 | 0.270 | 0.542 | 0.457 | 0.381 | 0.439 | 0.489 | 0.439 | 0.341 | 0.826 | 0.872 |

Table 18: Full downstream results for Gemma 2 2B with TA merging across five domain experts.

For each domain, we merge $k \in \{2, 4, 6, 8, 10, 12, 14, 16\}$ experts using TA, sampling multiple random $k$-subsets of experts, and evaluating CE on the corresponding domain. We report the mean CE, together with the empirical variance and standard deviation across random subsets. The overall row represents the macro-average across all 16 domains for each $k$.

**Findings.** As shown in Table 20, CE decreases as the number of merged experts $k$ grows, both per-domain and in the 16-domain macro-average, with clear diminishing returns: most of the improvement is obtained by small $k$, and the gains flatten as $k$ increases from 10 to 16. At the same time, the empirical variance and standard deviation across random expert subsets shrink with $k$, indicating that the merging outcomes become more stable as more experts are combined. Crucially, moving from 9 to 16 domains does not change the qualitative behaviour. The aggregated CE in the 16-domain setting still exhibits the same floor+tail scaling in $k$ as in our main experiments.

| folder | math_500 | gsm8k | ifeval | arc | hellaswag | mmlu | mbppplus | humanevalplus | wildguard_rta | harmbench_rta | | |
|---|---|---|---|---|---|---|---|---|---|---|---|---|
| 1-2 | 0.264 | 0.604 | 0.190 | 0.408 | 0.486 | 0.542 | 0.545 | 0.402 | 0.579 | 0.421 | 0.688 | 0.312 |
| 1-3 | 0.090 | 0.418 | 0.205 | 0.421 | 0.487 | 0.544 | 0.526 | 0.335 | 0.610 | 0.390 | 0.747 | 0.253 |
| 1-4 | 0.110 | 0.466 | 0.233 | 0.406 | 0.488 | 0.541 | 0.529 | 0.335 | 0.387 | 0.613 | 0.463 | 0.537 |
| 1-5 | 0.080 | 0.418 | 0.233 | 0.416 | 0.485 | 0.542 | 0.529 | 0.311 | 0.614 | 0.386 | 0.734 | 0.266 |
| 2-4 | 0.084 | 0.418 | 0.205 | 0.392 | 0.483 | 0.531 | 0.511 | 0.299 | 0.431 | 0.569 | 0.497 | 0.503 |
| 3-4 | 0.084 | 0.418 | 0.205 | 0.398 | 0.482 | 0.533 | 0.521 | 0.244 | 0.393 | 0.607 | 0.422 | 0.578 |
| 2-3 | 0.210 | 0.604 | 0.203 | 0.398 | 0.481 | 0.538 | 0.545 | 0.305 | 0.590 | 0.410 | 0.716 | 0.284 |
| 2-5 | 0.064 | 0.137 | 0.246 | 0.389 | 0.479 | 0.535 | 0.497 | 0.287 | 0.595 | 0.405 | 0.719 | 0.281 |
| 3-5 | 0.064 | 0.137 | 0.246 | 0.406 | 0.481 | 0.536 | 0.521 | 0.244 | 0.558 | 0.442 | 0.766 | 0.234 |
| 4-5 | 0.066 | 0.359 | 0.196 | 0.393 | 0.480 | 0.531 | 0.508 | 0.250 | 0.403 | 0.597 | 0.466 | 0.534 |
| 1-2-4 | 0.246 | 0.596 | 0.216 | 0.396 | 0.488 | 0.539 | 0.540 | 0.341 | 0.458 | 0.542 | 0.500 | 0.500 |
| 1-2-3 | 0.248 | 0.609 | 0.194 | 0.411 | 0.487 | 0.544 | 0.524 | 0.372 | 0.582 | 0.418 | 0.697 | 0.303 |
| 1-2-5 | 0.242 | 0.600 | 0.194 | 0.403 | 0.485 | 0.541 | 0.537 | 0.305 | 0.575 | 0.425 | 0.719 | 0.281 |
| 1-3-4 | 0.002 | 0.130 | 0.216 | 0.337 | 0.429 | 0.456 | 0.399 | 0.195 | 0.714 | 0.286 | 0.709 | 0.291 |
| 1-3-5 | 0.082 | 0.435 | 0.249 | 0.418 | 0.487 | 0.542 | 0.550 | 0.378 | 0.609 | 0.391 | 0.725 | 0.275 |
| 1-4-5 | 0.110 | 0.478 | 0.222 | 0.404 | 0.486 | 0.539 | 0.529 | 0.268 | 0.399 | 0.601 | 0.519 | 0.481 |
| 2-3-4 | 0.246 | 0.579 | 0.202 | 0.393 | 0.483 | 0.532 | 0.537 | 0.317 | 0.453 | 0.547 | 0.516 | 0.487 |
| 2-3-5 | 0.170 | 0.577 | 0.203 | 0.396 | 0.481 | 0.537 | 0.529 | 0.293 | 0.591 | 0.409 | 0.725 | 0.275 |
| 2-4-5 | 0.242 | 0.577 | 0.316 | 0.396 | 0.481 | 0.537 | 0.510 | 0.286 | 0.591 | 0.409 | 0.725 | 0.275 |
| 3-4-5 | 0.074 | 0.394 | 0.218 | 0.399 | 0.482 | 0.532 | 0.508 | 0.250 | 0.417 | 0.583 | 0.456 | 0.544 |
| 1-2-3-4 | 0.224 | 0.594 | 0.207 | 0.397 | 0.488 | 0.539 | 0.516 | 0.366 | 0.473 | 0.527 | 0.531 | 0.469 |
| 1-2-3-5 | 0.254 | 0.593 | 0.205 | 0.404 | 0.485 | 0.541 | 0.534 | 0.317 | 0.589 | 0.411 | 0.744 | 0.256 |
| 1-2-4-5 | 0.228 | 0.585 | 0.203 | 0.391 | 0.486 | 0.537 | 0.542 | 0.354 | 0.455 | 0.545 | 0.547 | 0.453 |
| 1-3-4-5 | 0.094 | 0.465 | 0.224 | 0.402 | 0.487 | 0.540 | 0.529 | 0.250 | 0.419 | 0.581 | 0.509 | 0.491 |
| 2-3-4-5 | 0.214 | 0.590 | 0.166 | 0.387 | 0.482 | 0.530 | 0.529 | 0.280 | 0.439 | 0.561 | 0.528 | 0.472 |
| 1-2-3-4-5 | 0.208 | 0.586 | 0.187 | 0.390 | 0.486 | 0.538 | 0.529 | 0.311 | 0.450 | 0.550 | 0.541 | 0.459 |

Table 19: Full downstream results for LLaMA 3.1 8B with TIES merging across five domain experts.

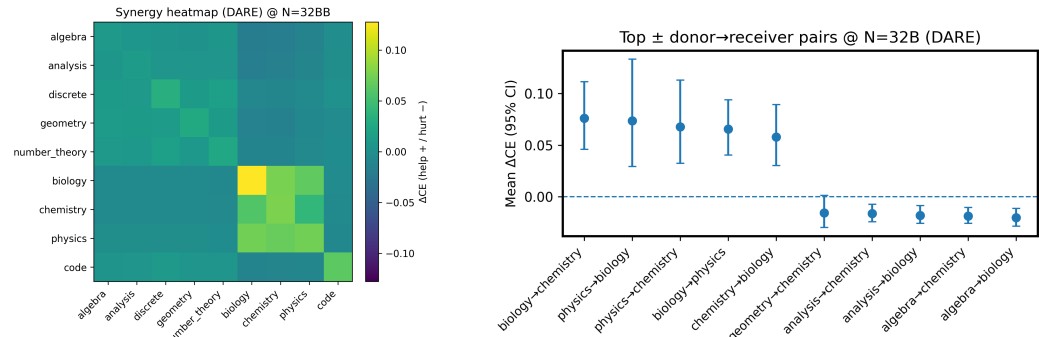

Figure 21: **Cross-domain synergy (DARE, 32B).** Left: synergy heatmap $S_{d \to e}$ (red = help, blue = hurt) showing science↔science and math↔math blocks; cross-block entries are weakly negative; *code*→(*discrete, geometry*) is mildly positive. Right: representative top $\pm$ pairs (donor→receiver) highlight actionable donor choices for target domains.

## L  APPENDIX: CROSS-DOMAIN SYNERGY

We quantify *donor–receiver* interactions by adding one expert at a time in the cross-domain setting (randomized orders) and recording the marginal change in macro CE for each evaluation domain, aggregating into a $9 \times 9$ *synergy matrix* $S_{d \to e}$. Using **DARE** at $N{=}32B$ as a representative case (Fig. 21), the heatmap reveals a structured, non-random pattern: science↔science pairs (physics, biology, chemistry) are strongly positive, math↔math pairs are moderately positive, and cross-block interactions are weakly negative at scale; *code* provides mild benefits to *discrete* and *geometry*. This structure is consistent with feature/skill overlap—closer domains supply complementary signal, while distant domains may dilute it—and persists across base sizes with slightly stronger block contrast for larger $N$. In practice, to help a science target, prioritize donors from {physics, biology, chemistry}; for math targets, stay within the math block or include *code*. We report the full matrix values, rank-ordered donor→receiver pairs, and size-wise trends in the released tables and replicate the qualitative structure for other methods (TA, TIES) with minor early-$k$ differences that narrow as $k$ grows.

**Details under DARE** We compute a $9 \times 9$ synergy matrix $S_{d \to e}$ by parsing each DARE trajectory from `results_dare_{0.5,1.5,3,7,14,32,72}B.csv`: at step $t$ (sequence

| Domain / Stat | Number of experts $k$ | | | | | | | |
|---|---|---|---|---|---|---|---|---|
| | 2 | 4 | 6 | 8 | 10 | 12 | 14 | 16 |
| **Code** | | | | | | | | |
| CE | 0.501872213 | 0.499083328 | 0.493360451 | 0.490470956 | 0.485253937 | 0.4851 | 0.48658 | 0.4914 |
| Var | 7.68E-05 | 0.000147146 | 0.000289833 | 0.000453827 | 0.000232502 | 0.00017 | 0 | |
| Std | 0.008763931 | 0.012130371 | 0.017024468 | 0.021303225 | 0.01524803 | 0.0133 | 0.0063 | |
| **Biology** | | | | | | | | |
| CE | 1.254403344 | 1.187543099 | 1.149616918 | 1.122063748 | 1.091473348 | 1.0853 | 1.06932 | 1.0607 |
| Var | 0.003412081 | 0.006524263 | 0.007148748 | 0.006041633 | 0.004593592 | 0.0027 | 0.000714 | |
| Std | 0.058413017 | 0.080772909 | 0.08455027 | 0.077727944 | 0.067776041 | 0.0521 | 0.02673 | |
| **Physics** | | | | | | | | |
| CE | 1.104021163 | 1.040059072 | 1.006337177 | 0.982421894 | 0.956452194 | 0.9501 | 0.93743 | 0.9293 |
| Var | 0.002224675 | 0.004339474 | 0.004616592 | 0.003374005 | 0.00256163 | 0.0014 | 0.0003 | |
| Std | 0.047166458 | 0.065874686 | 0.067945509 | 0.058086183 | 0.050612551 | 0.0378 | 0.0197 | |
| **Chemistry** | | | | | | | | |
| CE | 1.065878806 | 1.011048543 | 0.981000202 | 0.958958325 | 0.932410791 | 0.9276 | 0.91455 | 0.9067 |
| Var | 0.001910628 | 0.004432202 | 0.004765348 | 0.003875651 | 0.00334207 | 0.0019 | 0.0005 | |
| Std | 0.043710732 | 0.066574781 | 0.069031498 | 0.062254723 | 0.05781064 | 0.044 | 0.0232 | |
| **Geometry** | | | | | | | | |
| CE | 0.499684074 | 0.463583462 | 0.436880708 | 0.422598944 | 0.409463482 | 0.3991 | 0.3922 | 0.3839 |
| Var | 0.000569954 | 0.001058108 | 0.001218494 | 0.001190094 | 0.00067622 | 0.0005 | 0.000259 | |
| Std | 0.023873707 | 0.032528574 | 0.034906937 | 0.034497743 | 0.026004223 | 0.0238 | 0.01609 | |
| **Analysis** | | | | | | | | |
| CE | 0.420332789 | 0.390687826 | 0.368092274 | 0.3561941 | 0.34518962 | 0.3362 | 0.3312 | 0.3247 |
| Var | 0.000428451 | 0.000731257 | 0.000854753 | 0.000915523 | 0.000574783 | 0.00044 | 0.000207 | |
| Std | 0.020699068 | 0.027041758 | 0.029236164 | 0.030257614 | 0.023974637 | 0.0211 | 0.01439 | |
| **Number theory** | | | | | | | | |
| CE | 0.538458845 | 0.502845037 | 0.476979184 | 0.462241288 | 0.447491769 | 0.4345 | 0.42617 | 0.4182 |
| Var | 0.000612441 | 0.00113994 | 0.001340721 | 0.0015108 | 0.000863911 | 0.00067 | 0.00029 | |
| Std | 0.024747554 | 0.033762994 | 0.036615854 | 0.038869017 | 0.029392358 | 0.026 | 0.01723 | |
| **Discrete** | | | | | | | | |
| CE | 0.694258124 | 0.652957155 | 0.62294102 | 0.607414432 | 0.591427917 | 0.5777 | 0.569 | 0.5592 |
| Var | 0.000846054 | 0.001480839 | 0.001805925 | 0.001803155 | 0.0011049 | 0.00086 | 0.0004 | |
| Std | 0.029087002 | 0.038481669 | 0.042496179 | 0.042463568 | 0.033240039 | 0.0293 | 0.0202 | |
| **Algebra** | | | | | | | | |
| CE | 0.419097721 | 0.386756245 | 0.362299539 | 0.349713671 | 0.337287239 | 0.3268 | 0.3204 | 0.3130 |
| Var | 0.000505634 | 0.000875391 | 0.001036457 | 0.001059614 | 0.000619406 | 0.0005 | 0.00023 | |
| Std | 0.0224863 | 0.029587007 | 0.032194052 | 0.032551707 | 0.024887874 | 0.2271 | 0.01537 | |
| **Overall (16-domain macro-average)** | | | | | | | | |
| CE | 0.7774 | 0.7331 | 0.7051 | 0.6874 | 0.6685 | 0.6603 | 0.6509 | 0.6437 |
| Var | 0.0009 | 0.0017 | 0.0021 | 0.0018 | 0.0012 | 0.0006 | 0.0024 | |
| Std | 0.0310 | 0.0418 | 0.0461 | 0.0424 | 0.0357 | 0.0251 | 0.0156 | |

Table 20: Cross-entropy (CE), variance, and standard deviation for LLaMA-3.x 3B Instruct in the 16-domain setting (original 9 domains plus Japanese, medical, house-arrangement, Korean, emotion, elementary school mathematics, and Java code). For each domain and each number of merged experts $k \in \{2, 4, 6, 8, 10, 12, 14, 16\}$, we report the mean CE across random expert subsets, along with empirical variance and standard deviation. The overall row represents the macro-average across all 16 domains.

model length $t$), adding donor $d_t$ yields a marginal gain $\Delta L_e^{(t)} = L_e^{(t-1)} - L_e^{(t)}$ on evaluation domain $e$, and $S_{d \to e}$ averages these deltas over all occurrences (typically 11~13 per pair at 32B). Using domain blocks $\mathcal{M} = \{$algebra, analysis, discrete, geometry, number_theory$\}$ and $\mathcal{S} = \{$biology, chemistry, physics$\}$, block means $\overline{S}_{A \to B} = \frac{1}{|A||B|} \sum_{d \in A, e \in B} S_{d \to e}$ are: at 7B, $\overline{S}_{\mathcal{M} \to \mathcal{M}} = 0.009$, $\overline{S}_{\mathcal{S} \to \mathcal{S}} = 0.117$, $\overline{S}_{\mathcal{M} \to \mathcal{S}} = 0.014$, $\overline{S}_{\mathcal{S} \to \mathcal{M}} = -0.003$; at 14B, $0.016, 0.077, -0.011, -0.005$; at 32B, $0.012, 0.073, -0.013, -0.005$. The strongest off-diagonal positive pairs at 32B are biology→chemistry (+0.076), physics→biology (+0.074), physics→chemistry (+0.068), chemistry→biology (+0.066), biology→physics (+0.054); the largest negatives are algebra→physics (−0.026), geometry→chemistry (−0.020), discrete→chemistry (−0.018), algebra→biology (−0.016), number_theory→biology (−0.015). Donor strengths (row-sums, off-diagonal) rank physics 0.124 > biology 0.107 > chemistry 0.063 > discrete 0.025 $\gtrsim$ number_theory 0.021, with algebra and geometry weakest (−0.032, −0.005); receiver susceptibilities (column-sums) rank biology 0.133 > chemistry 0.087 > physics 0.059, while code is slightly negative (−0.029). We release the full matrices and top-pair lists for 7B/14B/32B/72B as CSVs (`out/rq6_synergy_matrix_32B_DARE.csv`, `out/rq6_top_pairs_32B_DARE.csv`); Fig. 21 visualizes the 32B heatmap and top pairs that these numbers summarize.

## L.1 DETAILS FOR ORDER SENSITIVITY AND $1/(k+b)$ FIT

From each DARE CSV we derive $k$ (hyphen count $+1$) and collect macro `Avg.` CE across all permutations to compute, per $(N, k)$, the across-order std, range, and CV; we then fit $\text{Std}_{\text{order}}(N, k) = c_0(N) + \frac{c_1(N)}{k+b}$ by grid-search over a small $b \in [0, 2]$ with linear least squares for $(c_0, c_1)$. Table 21 shows that dispersion collapses from $k=1$ to $k=8$ at 0.5B/32B/72B (std drops $\sim 79\%-81\%$, range $\sim 83\%$), and Table 22 reports fitted $(\hat{b}, c_0, c_1)$ and $R^2$ across sizes, where a single small offset $\hat{b} \approx 2$ with $c_0 \approx 0$ explains most of the decay; these are the statistics underlying the violin/heatmap/bar visualizations in Fig. 9.

| $N$ (B) | $k$ | mean CE | std (across orders) | range (max–min) | CV |
|---|---|---|---|---|---|
| 0.5 | 1 | 0.8164 | 0.0388 | 0.1122 | 0.048 |
| 0.5 | 8 | 0.7810 | 0.0081 | 0.0185 | 0.011 |
| 32 | 1 | 0.5207 | 0.0313 | 0.0865 | 0.060 |
| 32 | 8 | 0.4634 | 0.0060 | 0.0148 | 0.013 |
| 72 | 1 | 0.4638 | 0.0364 | 0.1056 | 0.072 |
| 72 | 8 | 0.4247 | 0.0076 | 0.0179 | 0.018 |

Table 21: **Across-order dispersion of `Avg.` CE at $k=1$ vs. $8$ (DARE).** Order sensitivity drops rapidly with $k$ at all $N$ (std $\sim -79\%-81\%$, range $\sim -83\%$).

| $N$ (B) | $\hat{b}$ | $c_0$ | $c_1$ | $R^2$ |
|---|---|---|---|---|
| 0.5 | 2.00 | $-0.002$ | 0.033 | 0.94 |
| 1.5 | 2.00 | $+0.002$ | 0.028 | 0.90 |
| 3 | 2.00 | $+0.003$ | 0.023 | 0.88 |
| 7 | 2.00 | $+0.002$ | 0.021 | 0.92 |
| 14 | 2.00 | $+0.003$ | 0.019 | 0.91 |
| 32 | 2.00 | $+0.001$ | 0.017 | 0.75 |
| 72 | 2.00 | $+0.002$ | 0.023 | 0.69 |

Table 22: $\text{Std}_{\text{order}}(N, k) \approx c_0 + c_1/(k+b)$ **fits (DARE).** A small shared offset $b \approx 2$ with $(c_0, c_1)$ per size explains the decay; $c_0$ is near zero (floor) and $c_1$ shrinks up to mid-scale.

## L.2 DETAILS FOR CROSS-BACKBONE/OPEN-SOURCE REPLICATION

For each backbone, every CSV row corresponds to one merge order (tokenized in the `model` field) evaluated on a domain with CE in `CE Loss`. We compute macro CE per order by averaging `CE Loss` over the nine domains, derive $k$ as the length of the `model` token list, and then average across orders with the same $k$ to obtain a per-backbone series $\{(k, \bar{L}_k)\}_{k=1}^9$. We fit $L(k) = L_\infty + \frac{A}{k+b}$ by least squares with a small grid over $b \in [0, 1]$; the best $b$ and $(L_\infty, A)$, along with $R^2$ and the end-point values $L(k=1)/L(k=9)$, are reported in Table 23. These numbers back Fig. 10 and show near-unity $R^2$ and small residuals, confirming that the same $1/(k+b)$ tail holds on LLaMA backbones.

| Backbone | $R^2$ | $b$ | $L_\infty$ | $A$ | $L(k=1)$ | $L(k=9)$ |
|---|---|---|---|---|---|---|
| LLaMA-3.2 3B | 0.9989 | 0.6875 | 0.7137 | 0.0783 | 0.7599 | 0.7221 |
| LLaMA-3 8B | 0.9955 | 0.0000 | 0.7252 | 0.0573 | 0.7837 | 0.7325 |

Table 23: **Fitted floor+tail parameters on LLaMA backbones (appendix).** Least-squares fits to macro-averaged CE vs. $k$; both series achieve near-unity $R^2$ with a shared $1/(k+b)$ tail.

# M   MIXING DOMAINS HELPS GENERALIZATION

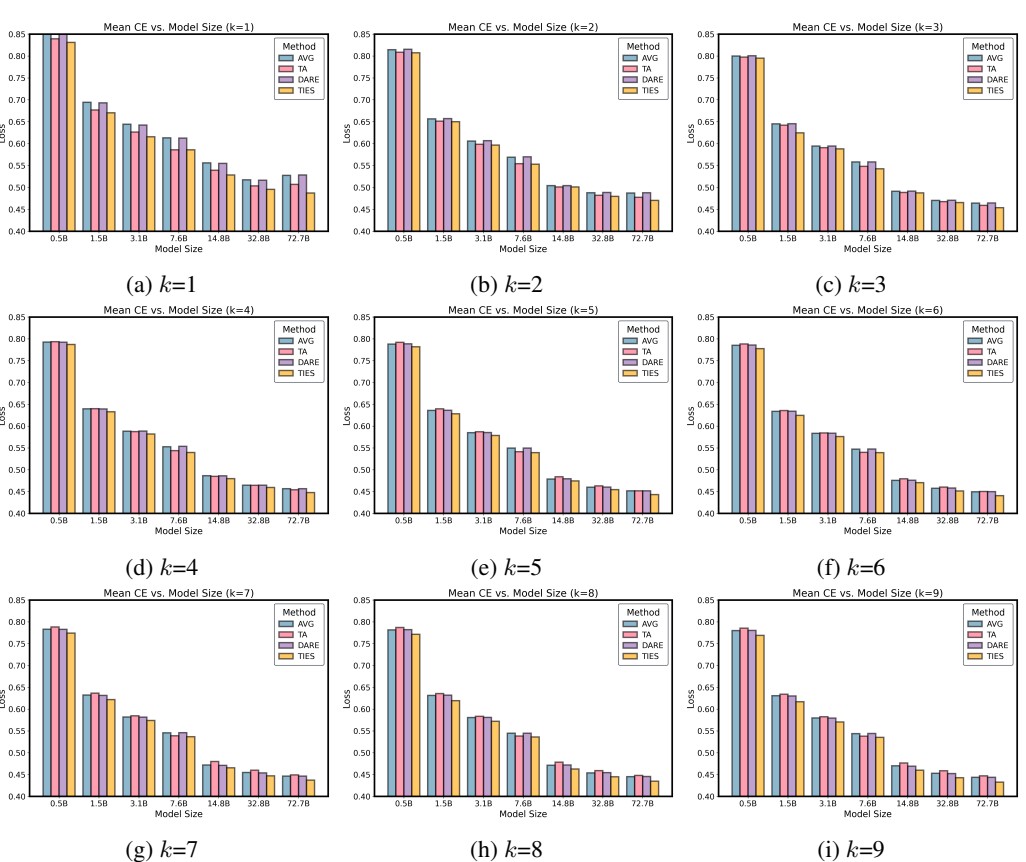

(a) $k$=1         (b) $k$=2         (c) $k$=3

(d) $k$=4         (e) $k$=5         (f) $k$=6

(g) $k$=7         (h) $k$=8         (i) $k$=9

Figure 22: Mean CE Loss vs. Model Size with Different $k$

