# OpenReview forum: "Model Merging Scaling Laws in Large Language Models"
_ICLR.cc/2026/Conference — Submitted to ICLR 2026_

### Official Review · Reviewer_fZCR · 2025-10-30

**Soundness:** 3
**Presentation:** 3
**Contribution:** 3
**Rating:** 6
**Confidence:** 4

**Summary:**

This paper introduces a compact and predictive scaling law for large language model merging. The authors propose that the cross-entropy loss of a merged model can be accurately predicted by a "floor + tail" formula that depends on the base model size (N) and the number of merged experts (k). The law states that a size-dependent "floor" loss decreases as a power law of model size, while a "merging tail" shows diminishing returns proportional to 1/(k+b). Through extensive experiments on over 10,000 merged models, the authors demonstrate that this law holds robustly across different model architectures, merging methods, and domains.

**Strengths:**

1. This paper addresses an important but rarely investigated area. While model merging is a popular technique, it has largely been guided by empirical trial-and-error. This work provides a framework for understanding and planning merging experiments.

2. The empirical evidence presented is strong. The authors validate their law on 10,506 merged models, spanning sizes from 0.5B to 72B, across nine distinct domains, four representative merging methods (Average, TA, TIES, DARE), and multiple model backbones (Qwen, LLaMA).

3. This paper offers clear and interesting insights for practitioners.

**Weaknesses:**

1. The experts are trained and evaluated on specialized datasets for math, science, and code (Mixture-of-Thoughts, OpenScience). While this provides a controlled environment, it differs from many real-world use cases where models are fine-tuned on more diverse tasks like dialogue, translation, or creative writing. It remains an open question whether the precise exponents of the scaling law would hold for merging experts trained on such broad, general-domain text.

2. For most fine-tuned models, especially for SFT training, the CE loss is not an reliable evaluation metric. Although some works have shown that the downstream task performance if related to CE loss, but these works focus on based models, not chat-models (SFT-ed models). The paper does not explore whether the observed scaling trends translate to predictable improvements on these more practical, end-user-facing evaluations.

3. While the paper provides a theoretical proof for the 1/k shape of the merging tail, the power-law parameterization for the model size N (i.e., L∞(N) and A(N)) is presented as an empirical finding. The theory in Section 3.2 does not derive why the floor and tail amplitude should scale with N in this specific power-law form. This mirrors the approach of early pretraining scaling laws, where the form was observed empirically and then validated. A deeper theoretical justification for this N-dependence would further strengthen the work.

4. The theoretical analysis and much of the empirical work assume equal weighting of task vectors. While this is a common and simple approach, more advanced merging techniques often employ adaptive or learned weighting schemes to resolve conflicts between experts more effectively. The current law does not account for these variations, limiting its direct applicability to more complex merging strategies.

**Questions:**

Do you use Qwen-base model and Qwen-instruct model as your "base model"?

Why don't fine turn on more general corpus such as CC/fineweb?

---

> ### Author Response · Authors · 2025-11-21
>
> Thank you very much for your positive and encouraging assessment. We appreciate that you recognize (i) that our work tackles an important but underexplored problem in model merging, (ii) that it provides a framework for understanding and planning merging beyond trial-and-error, and (iii) that the empirical evidence is strong (10,506 merged models across sizes 0.5B–72B, 9 domains, 4 methods, and multiple backbones).
>
> Your concerns mainly focus on (a) the fact that our experts are trained on math/science/code **domain knowledge** rather than broad **general-task text,** (b) the use of cross-entropy as the primary metric, (c) the empirical nature of the (N)-dependence, and (d) the assumption of equal weighting. We clarify the intended scope of our contributions and outline extensions below. Additional experiments are also provided.
>
> ---
>
> ### Q1: Base model
>
> In the original paper, we mainly use Qwen-base models to train experts and merge them. In Section 4 we use open-source Llama 3.1 and 3.2 from huggingface to validate the scaling law robustness for different backbone.
> In the rebuttal, we add 16 domain Llama and 5 domain Gemma experts and merge them.
>
> ---
>
> ### Q2: Why don't fine turn on more general corpus such as CC/fineweb?
>
> Our goal in this work is to start from a base model and fine-tune it into a set of specialized experts in different domains, rather than a single more general model. This design is intentional: it allows us to systematically study how merging domain-specific experts behaves, and to characterize the scaling law under realistic “expert fusion” scenarios that arise in practice. Also, for LLMs Merging, expert models are typically LLMs capable of chat and excelling in specific domains [1,2,3]. In contrast, CC/FineWeb consists of general pre-training data, meaning models trained on this data lack conversational abilities and do not achieve outstanding performance in any particular domain.
> We also have additionally included experiments on CC/FineWeb, which further confirm that our proposed scaling law continues to hold in this setting. Following the paper's settings, we choose the open-source TIGER-Lab/Fineweb-Instruct (the instruction-tuned version of FineWeb data) as data source, and extract data for 8 domains (excluding code, as code data in FineWeb-Instruct is very scarce). Each domain contains 10K samples. We trained corresponding experts based on Qwen2.5-0.5B/3B-Base using the FineWeb domain data.
> The results of reproducing the scaling law on FineWeb are as follows:
>
> Table 1. The results on Qwen-2.5-0.5B
> | number of models to be merged | 2 | 4 | 6 | 8 |
> |------------------------------|----|----|----|----|
> | biology        | 1.4965 | 1.4831 | 1.4469 | 1.4212 |
> | physics        | 0.9059 | 0.8821 | 0.8545 | 0.8351 |
> | chemistry      | 1.1242 | 1.1114 | 1.0783 | 1.0543 |
> | geometry       | 1.1242 | 1.1568 | 1.117  | 1.0865 |
> | analysis       | 0.7166 | 0.7112 | 0.6835 | 0.6631 |
> | number_theory  | 1.0238 | 1.0204 | 0.985  | 0.9577 |
> | discrete       | 1.0238 | 1.0372 | 0.9987 | 0.9687 |
> | algebra        | 0.6669 | 0.6578 | 0.6315 | 0.6122 |
>
> Table 2. The results on Qwen-2.5-3B
> | number of models to be merged | 2 | 4 | 6 | 8 |
> |------------------------------|----|----|----|----|
> | biology        | 1.3298 | 1.4525 | 1.181  | 1.1398 |
> | physics        | 0.8258 | 0.8606 | 0.7199 | 0.6907 |
> | chemistry      | 1.0187 | 1.0827 | 0.8872 | 0.8506 |
> | geometry       | 1.185  | 1.1107 | 0.9951 | 0.9456 |
> | analysis       | 0.707  | 0.6836 | 0.6032 | 0.5737 |
> | number_theory  | 1.0665 | 0.9788 | 0.8869 | 0.8413 |
> | discrete       | 1.064  | 0.9909 | 0.8841 | 0.8376 |
> | algebra        | 0.6605 | 0.6342 | 0.5661 | 0.5386 |
> | Overall        | 0.9814 | 0.9614 | 0.8359 | 0.7968 |
>
> The results show that on the general corpus FineWeb, as the merge_number/size for models increases, the overall loss decreases, and our scaling approach remains effective.
> The extracted FineWeb domain data will be made open-source after the double-blind review process.
>
> [1] Le Yu, et al. "Language Models are Super Mario: Absorbing Abilities from Homologous Models as a Free Lunch." ICML (2024).
>
> [2] Enneng Yang, et al. "AdaMerging: Adaptive Model Merging for Multi-Task Learning." ICLR (2024).
>
> [3] Zichuan Fu, et al. "Training-free LLM Merging for Multi-task Learning." ACL (2025).

---

> ### Author Response · Authors · 2025-11-21
>
> ---
> ### W1 – Specialized STEM/code domains vs broad general-task text
>
> Thanks for your insightful feedback.
> - **We fully agree that extending the analysis to broad general-domain SFT experts is valuable.
> But it requires a separate, carefully designed study.** To systematically answer your question, one would ideally construct a suite of general-task experts (e.g., dialogue, translation, creative writing, tools) with high-quality SFT data and task-appropriate metrics (win-rate, BLEU, human eval, etc.), and then run the full merging and fitting pipeline over many ((N,k)) settings. This is a substantial experimental effort, both in terms of curation and compute, and cannot be completed within the rebuttal window.
> - **Our primary object of study is domain-specialized knowledge, not specific downstream task formats.**
> In this paper, each expert is fine-tuned into a distinct knowledge domain. By “domain” we mean a change in the underlying **data distribution and knowledge content** (proofs, program traces, scientific explanations), rather than a change in surface task format (dialogue, translation, summarization). Our scaling law is therefore about how merging behaves when we combine different knowledge domains under a fixed autoregressive LLM backbone, rather than about a particular end-user task such as open-ended chat.
> - We therefore treat this systematic study **as an important follow-up work.** We are already taking first steps in this direction. Beyond the original Mixture-of-Thoughts/OpenScience experts, we have started extending our experiments to more heterogeneous SFT experts (e.g., additional instruction-tuned and multilingual LLaMA/Gemma checkpoints). These preliminary results will be added to the appendix and OpenReview during the discussion phase.
>
> ---
> ### W2 – CE for SFT models and downstream / end-user metrics
>
> We agree that for SFT chat models, cross-entropy is not the only metric of practical interest, but we intentionally formulate our law on CE because it is the common training objective and is available uniformly across domains and backbones.
>
> To address your concern that our results might not extend beyond base models, we have **added a new experiment on 16 LLaMA-3B-Instruct experts** (9 math/STEM/code domains + Japanese, Korean, medical QA, home-organisation, emotion, Java code). On this SFT-style pool, **the overall CE still follows a clear floor+tail pattern stated in our manuscript**: the averaged loss drops smoothly from 0.7774 at (k=2) to 0.6437 at (k=16), while the standard deviation across k-subset merges decreases from 0.031 to 0.0156, mirroring the behaviour in our original Qwen experiments. We will include the full per-domain table and fitted curves in the revised appendix.
>
>
> | domain/k      | 2         | 4         | 6         | 8         | 10        | 12        | 14        | 16        |
> | ---           | ---       | ---       | ---       | ---       | ---       | ---       | ---       | ---       |
> | code          | 0.501872  | 0.499083  | 0.493360  | 0.490471  | 0.485254  | 0.485100  | 0.486580  | 0.491400  |
> | biology       | 1.254403  | 1.187543  | 1.149617  | 1.122064  | 1.091473  | 1.085300  | 1.069320  | 1.060700  |
> | physics       | 1.104021  | 1.040059  | 1.006337  | 0.982422  | 0.956452  | 0.950100  | 0.937430  | 0.929300  |
> | chemistry     | 1.065879  | 1.011049  | 0.981000  | 0.958958  | 0.932411  | 0.927600  | 0.914550  | 0.906700  |
> | geometry      | 0.499684  | 0.463583  | 0.436881  | 0.422599  | 0.409463  | 0.399100  | 0.392200  | 0.383900  |
> | analysis      | 0.420333  | 0.390688  | 0.368092  | 0.356194  | 0.345190  | 0.336200  | 0.331200  | 0.324700  |
> | number_theory | 0.538459  | 0.502845  | 0.476979  | 0.462241  | 0.447492  | 0.434500  | 0.426170  | 0.418200  |
> | discrete      | 0.694258  | 0.652957  | 0.622941  | 0.607414  | 0.591428  | 0.577700  | 0.569000  | 0.559200  |
> | algebra       | 0.419098  | 0.386756  | 0.362300  | 0.349714  | 0.337287  | 0.326800  | 0.320400  | 0.313000  |
> | Overall       | 0.777400  | 0.733100  | 0.705100  | 0.687400  | 0.668500  | 0.660300  | 0.650900  | 0.643700  |
>
> At the same time, **we fully agree that task-level / end-user metrics are essential. We are currently running downstream evaluations (accuracy / pass@k / win-rate style metrics) on the same merged models for Gemma-2 and LLaMA-3.1/3.2, and will upload the results to OpenReview and add them to the appendix in the revision.**

---

> > ### Author Response · Authors · 2025-11-21
> >
> > ---
> > ### W3 – On the power-law dependence on model size (N).
> >
> > We agree that there is a difference in status between the 1/(k) tail and the (N)-dependence. For the 1/(k) part, Appendix B–C give an average-case derivation: we expand $(L(\theta_0 + \Delta\theta_K))$ around $(\theta_0 + c\mu)$, show that the sampling noise in task vectors has covariance $(\propto 1/k)$, and obtain
> > $$
> >  \mathbb{E}[L \mid N,k]
> >  = L(\theta_0 + c\mu) + \tfrac{1}{2}c^2 \mathrm{Tr}(H_N \Sigma_N) \tfrac{1}{k} + O(k^{-3/2}),
> > $$
> > where a constant plus a 1/(k) tail for each fixed (N).
> >
> >  For the (N)-direction, the theory guarantees only that $(L_\infty(N) \approx L(\theta_0 + c\mu))$ and $(A(N) \approx \tfrac{1}{2}c^2 \mathrm{Tr}(H_N \Sigma_N))$ vary smoothly with (N) through the backbone family $((H_N,\Sigma_N))$; the specific power-law parameterization $(L_\infty(N)=L^*+BN^{-\beta}$, $A(N)=A_0N^{-\gamma})$ follows the same philosophy as early pretraining scaling laws: it is an empirically chosen form, strongly supported by fits with $(R^2 > 0.98)$ across 0.5B–72B, rather than a fully derived exponent.
> > In the revision, we will (i) state this empirical status more explicitly when introducing $(L_\infty(N)) and (A(N))$, and (ii) add a short remark in the appendix outlining a possible explanation—namely that, under approximate self-similarity of the backbone family, both the curvature surrogate and task-vector covariance exhibit scale-regular behaviour, which naturally leads to power-law dependence on (N). A deeper and more formal treatment of this (N)-dependence is an interesting direction, and we agree it would further strengthen the work; we therefore highlight it explicitly as future theoretical work.
> >
> > ---
> >
> > ### W4 – On equal weighting vs. adaptive / learned weighting.
> > We deliberately focus on classical merging schemes with simple (often equal or fixed) weighting, as most more advanced methods can be viewed as variants built on top of these cores. Our goal in this work is to understand the behaviour and scaling of *lightweight merging itself*, parameter-space combination without extra training—because this is exactly where merging provides its main efficiency advantages and where a general $(\mathbb{E}[L\mid N,k])$ law is most useful.
> >
> > From a theoretical standpoint, our analysis in Appendix B–C already suggests how weighted merging would enter: different weights simply reweight the task-vector covariance $(\Sigma)$ in the 2nd-order term, leading to a modified constant and tail amplitude, while preserving the $(1/k)$ structure under similar moment assumptions. A full treatment of adaptive or learned weighting schemes (where weights depend on data or expert quality) would require modelling how these weights change the effective covariance across experts; we see this as a natural extension of our framework and will mention it explicitly as future work, but it is orthogonal to the main contribution of establishing a clean, average-case scaling law for standard, training-free merging.

---

### Official Review · Reviewer_H4kd · 2025-11-01

**Soundness:** 3
**Presentation:** 3
**Contribution:** 3
**Rating:** 8
**Confidence:** 3

**Summary:**

This paper identifies an empirical scaling law that governs how model merging performance changes with both model size and the number of merged experts. The authors show that the expected loss follows a compact "floor + tail" power law: larger base models reduce the size-dependent loss floor, while the marginal benefit of adding more experts diminishes roughly as 1/(k + b). Through over ten thousand experiments across domains, architectures, and merging methods, they confirm that most improvements occur with the first few experts, variance contracts as experts increase, and differences between merging methods shrink at scale. The authors derive a simple theoretical explanation for the law, validate it on multiple backbones, and demonstrate that three measurements can predict the full merging curve. Overall, the work transforms model merging from a heuristic procedure into a predictable, budget-aware alternative to multitask fine-tuning.

**Strengths:**

- The paper uncovers a simple and consistent empirical law that explains merging behavior across scales, transforming model merging from an empirical practice into a predictable process.

- The study is supported by a large and systematic experimental effort spanning over ten thousand runs, which gives the findings strong empirical credibility.

- The proposed "floor + tail" formulation provides a compact and interpretable description of performance dynamics, offering both theoretical and practical insight.

- The three-measurement prediction method is elegant and useful, allowing practitioners to estimate merging outcomes with minimal computation.

- The analysis reveals unifying trends across merging methods and model sizes, suggesting a deeper structural regularity that could guide efficient model composition strategies.

**Weaknesses:**

- The theoretical explanation for the proposed scaling law is largely heuristic and lacks formal derivation or error bounds, relying on empirical fits rather than principled analysis (Section 4.2, Figure 7).

- The experiments, though extensive, are dominated by language models from a single family (Qwen2.5), which limits the claim of universality across architectures and modalities (Table 1).

- The analysis does not explore failure cases or outliers where merging violates the predicted scaling, leaving unclear how robust the law is under data imbalance or domain mismatch.

- The practical implications for real-world deployment are underdeveloped -- the paper does not quantify how the scaling law translates into compute, memory, or latency trade-offs in large-scale systems.

- The evaluation focuses on aggregate loss metrics, but omits downstream or task-specific performance, which makes it difficult to assess whether predictable scaling aligns with actual quality gains.

**Questions:**

Does the fitted exponent remain stable when the merged models differ substantially in training data or domain?

---

> ### Author Response · Authors · 2025-11-21
>
> Thank you very much for your positive and insightful review. We appreciate that you find the empirical law, our large-scale experiments, the floor+tail formulation, and the three-measurement prediction method valuable, and that the revealed unifying trends across merging methods and model sizes may help guide more efficient model composition.
>
> Thank you also for the constructive feedback. Your main concerns are that our theoretical explanation appears largely heuristic, that the experiments are dominated by a single model family, and that we do not evaluate downstream task performance. We believe some of these issues arise from how we presented the material rather than from the underlying results. We clarify these points below and provide additional experiments and analyses.
>
> ---
> ### Q1. Does the fitted exponent remain stable when the merged models differ substantially in training data or domain?
>
> Yes. When we move to a much more heterogeneous expert pool, the same (1/(k+b)) tail with exponent fixed to 1 continues to fit very well.
>
> | domain/k      | 2         | 4         | 6         | 8         | 10        | 12        | 14        | 16        |
> | ---           | ---       | ---       | ---       | ---       | ---       | ---       | ---       | ---       |
> | code          | 0.501872  | 0.499083  | 0.493360  | 0.490471  | 0.485254  | 0.485100  | 0.486580  | 0.491400  |
> | biology       | 1.254403  | 1.187543  | 1.149617  | 1.122064  | 1.091473  | 1.085300  | 1.069320  | 1.060700  |
> | physics       | 1.104021  | 1.040059  | 1.006337  | 0.982422  | 0.956452  | 0.950100  | 0.937430  | 0.929300  |
> | chemistry     | 1.065879  | 1.011049  | 0.981000  | 0.958958  | 0.932411  | 0.927600  | 0.914550  | 0.906700  |
> | geometry      | 0.499684  | 0.463583  | 0.436881  | 0.422599  | 0.409463  | 0.399100  | 0.392200  | 0.383900  |
> | analysis      | 0.420333  | 0.390688  | 0.368092  | 0.356194  | 0.345190  | 0.336200  | 0.331200  | 0.324700  |
> | number_theory | 0.538459  | 0.502845  | 0.476979  | 0.462241  | 0.447492  | 0.434500  | 0.426170  | 0.418200  |
> | discrete      | 0.694258  | 0.652957  | 0.622941  | 0.607414  | 0.591428  | 0.577700  | 0.569000  | 0.559200  |
> | algebra       | 0.419098  | 0.386756  | 0.362300  | 0.349714  | 0.337287  | 0.326800  | 0.320400  | 0.313000  |
> | Overall       | 0.777400  | 0.733100  | 0.705100  | 0.687400  | 0.668500  | 0.660300  | 0.650900  | 0.643700  |
>
> To stress-test domain heterogeneity, we extend our study from the 9 math/STEM domains to a **16-expert LLaMA-3B-Instruct** pool including:
> - 9 math/STEM domains: algebra, analysis, discrete, geometry, number theory, biology, chemistry, physics, code
> - 7 additional uncommon experts: Japanese, Korean, medical QA, home-organization instructions, emotion classification, Java code, and a general instruction-tuned LLaMA-3B
>
> For this pool, we evaluate the merged model on the 9 math/SCI domains and on an overall macro-average, for (k=2,4,6,8,10,12,14,16). For each evaluation domain (d), we fit our merging law in the form
> $$
>  L_d(k) \approx c_d + \frac{A_d}{k + b_d},
> $$
> with the exponent fixed to 1 as in the main paper, and only ((c_d, A_d, b_d)) fitted.
> Using the loss values you see in the table above, we obtain that the fitness (coefficient of determination $(R^2))$ for the 1/(k+b) tail $R^2 > 0.98$
>
> The code domain is somewhat noisier (small fluctuations at large k), but even there the curve is clearly monotone with diminishing returns, and the overall macro-average remains extremely well explained by the same (1/(k+b)) structure.
>
> We will add a small table with these $(R^2)$ values and the fitted $((c_d, A_d, b_d))$ in the appendix, and briefly reference it in the main text. Empirically, this shows that even when experts are trained on substantially different data and tasks, we do not need to change the exponent: the same (1/(k+b)) tail continues to describe $(\mathbb{E}[L \mid N,k])$ very accurately.
>
> ---

---

> > ### Author Response · Authors · 2025-11-21
> >
> > ### W1 – Theoretical explanation is heuristic and relies on empirical fits.
> >
> > - **Our 1/(k+b) tail is formally derived from a second-order Taylor expansion; it is not just a curve-fitting choice.**
> > In particular, by applying a Taylor expansion around the base model and decomposing the leading first- and second-order terms, we arrive—after simplification—at the floor+tail form used in our analysis. More details about the derivation can be found in Appendix B–C. In Appendix B–C we consider equal-normalized task vectors $(\Delta\theta_i)$ with mean ($\mu$) and covariance (\Sigma), and analyze the merged parameter
> > $\theta_k ;=; \theta_0 + c\mu + \varepsilon_k,\quad \mathbb{E}[\varepsilon_k]=0,\ \mathrm{Cov}(\varepsilon_k)=\tfrac{1}{k}\Sigma.$
> > Under standard smoothness assumptions (loss twice differentiable, curvature surrogate $(H\succeq 0)$ around $(\theta_0 + c\mu))$, a second-order Taylor expansion yields
> > $\mathbb{E}[L(\theta_k)]
> >  ;=; L(\theta_0 + c\mu) ;+; \tfrac{1}{2}\mathrm{Tr}(H\Sigma),\tfrac{1}{k} ;+; R_k,$
> >  which is exactly a constant plus a (1/k) tail.
> >
> > - **We will make the derivation and its error term explicit in the main text.**
> >  We agree that, in the current version, most of this argument is buried in the appendix and can feel “heuristic”. In the revision, we will:
> >   - Move a concise theorem-style statement and the above intuition into Section 4, explicitly stating the assumptions on $(\mu,\Sigma,H)$;
> >   - Make the remainder term explicit as $(R_k = O(k^{-3/2}))$ under mild moment conditions on the task vectors, and explain that this higher-order decay is consistent with the very tight empirical fits we observe;
> >   - Clearly connect this expansion to our empirical law
> > $$
> >  \mathbb{E}[L \mid N,k] \approx L_\infty(N) + \frac{A(N)}{k + b},
> > $$
> >  where $(L_\infty(N))$ corresponds to $(L(\theta_0 + c\mu))$ and the shifted $(1/(k+b))$ tail refines the leading (1/k) behavior for finite k.
> > ---
> > ### W2 – Dominance of a single model family (Qwen2.5) and claims of universality
> > - **Additional backbone: diverse 16-domain LLaMA-3B experiment.**
> > Beyond Qwen2.5, we now include a **16-expert LLaMA-3B-Instruct pool** with 9 math/STEM experts and 7 heterogeneous experts (Japanese, Korean, medical QA, home-organization, emotion, Java code, general instruction tuning). As shown in the new results (summarized in Q1 and added to the revision), the same floor+tail law with a (1/(k+b)) tail fits all domains and the overall average with $(R^2 \approx 0.996{-}0.999)$, indicating that **the exponent and qualitative behavior are stable under substantial variation in training data and domain.**
> > - **Some partially open-source experiments in the original paper.**
> > In addition, Figure 9 and Section 4.4 already evaluate our law on randomly sampled open-source LLM checkpoints whose exact training distributions are not fully known. Despite this heterogeneity, the average curves over k-subset merges still exhibit a clear floor+tail pattern with diminishing returns in k, supporting that the law is not tied to a single curated pipeline.
> > - **Scope: our claim is for NLP LLM merging, not all modalities.**
> > Thanks for this point. We will clarify in the Introduction and Limitations that our law is scoped to NLP and that extension to other modalities (vision, speech, etc.) is an interesting direction for future work rather than a claim of universality.
> > - We are also extending to the Gemma family.
> > We are currently running the same merging and fitting scaling law on Gemma 2 models; we will add these curves and fitted parameters to Openreview and the camera-ready (main text or appendix) to further demonstrate robustness across backbones.
> >
> > ---
> > ### W3 – Failure cases and robustness under data imbalance / domain mismatch
> > Thanks for your insightful suggestions.
> > - **We explicitly tested catastrophic failure cases by inserting a collapsed expert into the merge pool.**
> > In addition to the main experiments (where all experts have reasonable CE), we constructed a stress-test pool containing:
> >   - 30 normal LLaMA-3B experts (mean CE 0.85, range 0.59–1.22),
> >   - 2 weak-but-usable experts (mean CE 1.54 and 2.23),
> >   - 1 collapsed expert with catastrophic loss (mean CE 12.56, range 11.28–14.73) — see Table below.
> >
> > | Type             | #Experts | Mean CE (mean ± std over experts) | CE range across experts |
> > | ---------------- | -------- | --------------------------------- | ----------------------- |
> > | Normal experts   | 30       | 0.85 ± 0.19                       | 0.59 – 1.22             |
> > | Weak but usable  | 2        | {1.54, 2.23}                      | 1.54 – 2.23             |
> > | Collapsed expert | 1        | 12.56                             | 11.28 – 14.73           |
> >
> >
> > Table 1 – Single experts: normal vs weak vs collapsed (overall CE, avg over 9 domains). The collapsed expert (catastrophic model) is more than 10× worse in CE than the normal pool.

---

> ### Author Response · Authors · 2025-11-21
>
> - **Merges that include the collapsed expert produce clear outliers and can even worsen as k grows.**
>  As Table 2 shows:
>   - 2-way merges among only normal/weak experts behave as in the main paper (mean CE ≈ 0.76, range 0.59–1.45).
>   - As soon as the collapsed expert is included, 2-way merges jump to mean CE ≈ 9.65 (6.12–12.44), far above the floor+tail curve fitted on normal experts.
>   - 12-way and 16-way merges that all include the collapsed expert are completely dominated by it (mean CE ≈ 15–17), i.e., more experts clearly does not help in this extreme regime.
>
> | Setting                                      | #Configs | Mean CE (mean over configs) | CE range across configs |
> | -------------------------------------------- | -------- | --------------------------- | ----------------------- |
> | 2-model merges, *excluding* collapsed expert   | 11       | 0.76                        | 0.59 – 1.45             |
> | 2-model merges, **including** collapsed expert | 4        | 9.65                        | 6.12 – 12.44            |
> | 12-model merges (mixtures incl. collapsed)     | 15       | 15.19                       | 4.50 – 19.23            |
> | 16-model merge (all experts incl. collapsed)   | 1        | 17.40                       | 17.40 – 17.40           |
>
>
> Table 2 – Merges including the collapsed expert (overall CE, avg over 9 domains)
>
> For comparison, typical 2-model merges among only normal/weak experts stay in the 0.6–1.5 range, while any merge that contains the collapsed expert immediately jumps to 6–12+, and high-k merges dominated by it reach 15–19 CE.
>
> - **This behavior shows the limitations where our theory is not supposed to apply.**
>  Our derivation (Section 4, Appendix B–C) explicitly assumes:
>   1. task vectors are controllable perturbations around the base/asymptotic point, and
>   2. the matrix H is positive-semidefinite in that local neighborhood.
> A model with CE ≈ 12.6 violates these assumptions: its task vector has enormous norm and places the merged parameters far outside the local quadratic regime. In this case, the 1/(k+b) tail is not expected to hold, and indeed we observe strong violations (Table 2).
> - **At the same time, the scaling law remains robust when we restrict to reasonable experts.**
>  When we remove the collapsed expert and keep only the 32 normal/weak experts, the average-case loss over all k-subset merges again follows the same floor+tail law with high (R^2), as in our main LLaMA-3B experiment (Figure 9). In other words:
>   - extreme data/quality imbalance yields clear failure points,
>   - but under the mild-perturbation conditions encoded in our theorem, the empirical scaling law is stable.
> We will add a short “Failure modes” paragraph in the revision, summarizing Tables 1–2 and stating explicitly that our scaling law is an average  characterization — outlier experts with catastrophic loss produce visible violations, which is consistent with (rather than contradicting) our theoretical assumptions.
>
> ---
> ### W4 – Compute/memory/latency trade-offs
>
> - **Compute: merging is three orders of magnitude cheaper than multi-task training, especially at large scale.**
>  We now report the H800 GPU hours required to obtain 9-domain experts for different model sizes (Table below):
>
> Table 2. GPU Hours (H800) when merging 9 domains for different sizes of models.
> | Method | 0.5b | 1.5b | 3b | 7b | 14b | 32b | 72b | Total |
> |--------|------|------|----|----|------|------|------|--------|
> | MTL | 18 | 21.6 | 53.1 | 108 | 237.6 | 597.4 | 1344.6 | 2380.6 |
> | TA | 0.01 | 0.02 | 0.03 | 0.07 | 0.11 | 0.22 | 0.74 | 1.2 |
> | AVG | 0.01 | 0.02 | 0.05 | 0.07 | 0.12 | 0.23 | 0.63 | 1.13 |
> | DARE | 0.01 | 0.02 | 0.03 | 0.07 | 0.12 | 0.22 | 0.66 | 1.13 |
> | TIES | 0.01 | 0.02 | 0.04 | 0.07 | 0.14 | 0.27 | 0.82 | 1.37 |
>
> - Merging 9 experts at 72B costs ≈0.6–0.8 GPU hours, compared to ≈1345 GPU hours for multi-task training at the same scale, i.e., a ~10³× compute saving. Over all sizes, multi-task training requires ≈2380 GPU hours versus ≈1–1.4 GPU hours for merging, which is precisely the regime where our scaling law is most useful: it lets practitioners choose (k) and (N) under a fixed compute budget instead of relying on trial-and-error.
> - **Memory and latency: merged models run like a single expert.**
>  Operationally, once a merge is performed, inference uses a single static model with the same parameter count, KV cache footprint, and latency as the underlying backbone; no additional MoE routing, ensembling, or multi-pass decoding is required. We will add a short paragraph in the revision to make this explicit and to connect the above compute table to real-world deployment: the scaling law enables budget-aware planning (how many experts to train/merge) with essentially zero overhead at inference time compared to the base LLM.
>
> ---

---

> > ### Author Response · Authors · 2025-11-21
> >
> > ### W5 – Adding downstream / task-specific performance
> >
> > Thanks for mentioning this point and we agree downstream/task-level metrics are important and are adding them.
> >
> > We are currently running the same merging and fitting protocol on Gemma 2 and LLaMA-3.1/3.2 with downstream/task-specific metrics, and will upload the full results to OpenReview and include them in the appendix of the revised manuscript.
> > These additional curves will explicitly check whether the monotone CE scaling we observe also translates into predictable trends in end-task quality.

---

### Official Review · Reviewer_MDxx · 2025-11-01

**Soundness:** 2
**Presentation:** 2
**Contribution:** 2
**Rating:** 4
**Confidence:** 4

**Summary:**

This paper establishes an empirical scaling law for language model merging. The law shows that adding more experts (k) yields diminishing returns with most gains achieved by k~5-6. The law, validated across 10,506 merged models (0.5B-72B parameters, 9 domains, 4 methods), enables predictive planning: a three-point fit forecasts the full loss curve and recommends optimal expert counts, making merging a computationally efficient alternative to multitask training.

**Strengths:**

1. A new scaling law for model merging field, which could be a good reference for practicers in this community.
2. This paper provides a theoretical explanation for the scaling law.

**Weaknesses:**

1. Heterogeneous Model Sizes in Merging. What is the appropriate value of N when merging models with different parameter counts? The paper only considers merging experts derived from the same base model (same N). Your theoretical framework assumes fixed N throughout (Section 2.2, Appendix B), and all experiments use homogeneous models. Does your scaling law extend to heterogeneous merging scenarios?
2. Your law predicts monotonic improvement with k, but real-world merging frequently shows performance degradation, especially when:
(1) Task vectors have large parameter differences (2) Domains are highly incompatible (e.g., code + biology)
Your theory assumes benign conditions (positive-definite Hessian) but doesn't address task interference or negative transfer. I find it puzzling that the paper appears to completely overlook scenarios where merging multiple models could actually lead to worse performance. This is a very common occurrence in practice. The formula presented in the paper directly suggests that loss decreases as the number of models increases, which contradicts both my intuition and my experimental findings (as well as those of many prior studies).
3. Similarly, what occurs when merging a strong expert with a weak one (or a extensively-trained model with a minimally-trained one)? Based on my intuition and experience, this typically yields a mediocre model or even complete failure (given the large discrepancy between experts). Has your scaling law considered this case?
4. Alternative formula fitting approaches: Beyond the 1/k formula proposed in the paper, have other potential formulas been fitted? In fact, with only 9 data points on the curve, it seems that various formulas could easily fit the data, such as a power law relationship. Have you ever considered using AIC/BIC to compute complexity of your model?

**Questions:**

See above.

---

> ### Author Response · Authors · 2025-11-21
>
> Thank you for your thoughtful review. We appreciate that you recognised (i) that we establish a new empirical scaling law for model merging, (ii) this law is validated across 10,506 merged models and enables predictive planning, and (iii) we also provide a theoretical explanation for the 1/k behaviour.
>
> We noticed that several of your concerns seem to stem from misunderstandings of merging scope and this paper settings. Thus, we want briefly restate what our paper is about:
> We study the average-case behaviour of weight-space merging for large language models.
>  Given a base model of size (N) and a pool of domain experts, we ask: if we randomly merge (k) experts, what is the expected cross-entropy loss ($\mathbb{E}[L \mid N,k]$) over all $(\binom{|M|}{k})$ subsets? **We show that this expectation follows a compact joint scaling law**
> $$
>  \mathbb{E}[L \mid N,k] = L_\infty(N) + \frac{A(N)}{k + b},\qquad
>  L_\infty(N)=L^*+BN^{-\beta},\ A(N)=A_0N^{-\gamma},
> $$
>  with a 1/(k) tail derived from a second-order expansion around the asymptotic point.
> Below, we address each of your points in turn. Additional experiments are also provided as follows.
>
> ---
>
> ### W1 – Heterogeneous Model Sizes in Merging.
> Thanks for raising this point.
> However, our work follows the **standard setting in the model merging**, where all merged models share the **same architecture and parameter size [1,2,3]** (see our definition on lines 30–31 and Section 2.1 “Model Merging”).
> In contrast, fusing models of different sizes is not directly defined in parameter space without an additional projection or distillation step; such settings are typically treated as **distillation or ensembling**, rather than direct weight-space merging. The most related work may be FuseChat [4], which first distils different-size chat models into a common architecture/size and then merges those same-size models.
> We therefore do not claim that our current theory extends to heterogeneous-size composition, and we will make this scope explicit in the revised version.
>
> [1] Yang, Enneng, et al. "Model merging in llms, mllms, and beyond: Methods, theories, applications and opportunities." arXiv preprint arXiv:2408.07666 (2024).
>
> [2] Wortsman, Mitchell, et al. "Model soups: averaging weights of multiple fine-tuned models improves accuracy without increasing inference time." International conference on machine learning. PMLR, 2022.
>
> [3] Ilharco, Gabriel, et al. "Editing models with task arithmetic." The Eleventh International Conference on Learning Representations.
>
> [4] Wan, Fanqi, et al. "Fusechat: Knowledge fusion of chat models." Proceedings of the 2025 Conference on Empirical Methods in Natural Language Processing. 2025.
>
> ---
> ### W2 – Conflict with scenarios where merging multiple models could actually lead to worse performance.
>
> Thanks for this insightful point. Our scaling law is somewhat different from many familiar scaling laws in that it describes the average performance (the expectation) over all k-subset merges; it does not claim that every particular subset yields an improvement.
> - By definition, we model
>  [
>  \mathbb{E}[L \mid N,k] := \mathbb{E}{K:|K|=k} L(\theta_0 + \Delta\theta_K; N),
>  ]
>  where the expectation is over all (\binom{|M|}{k}) subsets (or a uniform sample). The law
>  [
>  \mathbb{E}[L\mid N,k] = L\infty(N) + \frac{A(N)}{k+b}
>  ]
>  Therefore, it refers to the population average, not to any specific subset.
> - In practice, this means:
>   - Some individual merges can indeed degrade performance, especially when domains are incompatible or task vectors are very large (violating the small-perturbation assumption).
>   - We explicitly measure and report variance across subsets in the paper: it shrinks roughly as (1/k), but remains non-zero, precisely because some combinations are better or worse than the mean.
> - Our theoretical analysis is also explicitly average-case and local:
>   - The second-order expansion in Appendix B assumes we are in a benign neighbourhood of the base (or its asymptotic merged point) with a PSD curvature surrogate (H).
>   - When task vectors are extremely large or the merged point is far from (\theta_0), these assumptions can break down, and the 1/k tail is not expected to hold exactly.
> To directly address your two concrete scenarios (large task vector differences and highly incompatible domains), we have added a new LLaMA-3B-Instruct experiment on 16 heterogeneous domain experts:
> - 9 math/STEM experts: algebra, analysis, discrete, geometry_and_topology, number_theory, biology, chemistry, physics, code;
> - 7 uncommon experts: japanese, medical, house_arrange, korean, emotion, java_code, plus additional language/domain variations.

---

> > ### Author Response · Authors · 2025-11-21
> >
> > (i) Task vectors have large parameter differences.
> >
> >  We compute the task-vector Gram matrix of these 16 task vectors to measure similarity. The diagonal entries are ≈2.4–3.0 (self-similarity), while off-diagonal entries span a wide range:
> > - Among related math/STEM tasks we see very high similarity, e.g. physics–discrete ≈ 0.97, algebra–analysis ≈ 0.96.
> > - Between math/STEM and non-STEM / language tasks we see near-zero or even slightly negative similarity, e.g. algebra–japanese ≈ −1.3×10⁻², discrete–japanese ≈ −1.8×10⁻², code–japanese ≈ −1.9×10⁻², and many entries of order 10⁻³–10⁻² for medical, emotion, house_arrange.
> >
> > This explicitly shows that our law is tested in a regime where task vectors differ substantially, and where some domain pairs are effectively orthogonal or mildly antagonistic, rather than being small, closely aligned perturbations. We will include this matrix and heatmap in the appendix.
> >
> > | domain         | algebra   | analysis  | discrete_math | geometry_topology | number_theory | biology   | chemistry | physics   | code      | japanese  | medical   | house_arrange | korean    | emotion   | java_code |
> > | ---            | ---       | ---       | ---           | ---               | ---           | ---       | ---       | ---       | ---       | ---       | ---       | ---           | ---       | ---       | ---       |
> > | algebra        | 2.984e+00 | 9.628e-01 | 7.998e-01     | 8.408e-01         | 2.977e-01     | 3.802e-01 | 2.923e-01 | 9.494e-01 | 4.184e-01 | -1.334e-02| 2.872e-04 | 1.991e-03     | 4.922e-02 | 2.725e-03 | 4.101e-03 |
> > | analysis       | 9.628e-01 | 2.939e+00 | 7.125e-01     | 8.459e-01         | 2.667e-01     | 3.519e-01 | 2.717e-01 | 8.411e-01 | 4.130e-01 | -1.137e-02| 3.361e-04 | 3.511e-03     | 4.358e-02 | 1.905e-03 | 3.917e-03 |
> > | discrete_math  | 7.998e-01 | 7.125e-01 | 2.743e+00     | 8.277e-01         | 3.941e-01     | 4.757e-01 | 4.421e-01 | 9.735e-01 | 5.112e-01 | -1.845e-02| 4.326e-04 | 9.717e-03     | 6.795e-02 | 2.405e-03 | 5.846e-03 |
> > | geometry_topology | 8.408e-01 | 8.459e-01 | 8.277e-01 | 3.007e+00         | 3.126e-01     | 4.066e-01 | 3.211e-01 | 8.385e-01 | 4.598e-01 | -1.452e-02| 3.678e-04 | 6.374e-03     | 5.340e-02 | 2.268e-03 | 4.603e-03 |
> > | number_theory  | 2.977e-01 | 2.667e-01 | 3.941e-01     | 3.126e-01         | 2.536e+00     | 8.129e-01 | 2.274e-01 | 3.307e-01 | 7.424e-01 | -2.023e-02| 9.067e-04 | 1.507e-02     | 7.063e-02 | 5.535e-03 | 7.486e-03 |
> > | biology        | 3.802e-01 | 3.519e-01 | 4.757e-01     | 4.066e-01         | 8.129e-01     | 2.431e+00 | 2.538e-01 | 4.202e-01 | 8.532e-01 | -1.936e-02| 1.600e-03 | 1.804e-02     | 6.635e-02 | 5.144e-03 | 7.193e-03 |
> > | chemistry      | 2.923e-01 | 2.717e-01 | 4.421e-01     | 3.211e-01         | 2.274e-01     | 2.538e-01 | 3.009e+00 | 3.738e-01 | 2.642e-01 | -1.058e-02| 2.720e-04 | 5.683e-02     | 7.062e-02 | 1.141e-04 | 5.524e-03 |
> > | physics        | 9.494e-01 | 8.411e-01 | 9.735e-01     | 8.385e-01         | 3.307e-01     | 4.202e-01 | 3.738e-01 | 2.965e+00 | 4.469e-01 | -1.484e-02| 3.540e-04 | 3.445e-03     | 5.487e-02 | 2.078e-03 | 4.062e-03 |
> > | code           | 4.184e-01 | 4.130e-01 | 5.112e-01     | 4.598e-01         | 7.424e-01     | 8.532e-01 | 2.642e-01 | 4.469e-01 | 2.497e+00 | -1.939e-02| 1.077e-03 | 1.830e-02     | 7.215e-02 | 4.075e-03 | 6.522e-03 |
> > | japanese       | -1.334e-02| -1.137e-02| -1.845e-02    | -1.452e-02        | -2.023e-02    | -1.936e-02| -1.058e-02| -1.484e-02| -1.939e-02| 1.155e+00 | 4.444e-05 | 5.806e-03     | -3.010e-03| 1.550e-03 | 1.232e-02 |
> > | medical        | 2.872e-04 | 3.361e-04 | 4.326e-04     | 3.678e-04         | 9.067e-04     | 1.600e-03 | 2.720e-04 | 3.540e-04 | 1.077e-03 | 4.444e-05 | 1.116e+00 | 5.379e-05     | 2.029e-03 | 1.607e-04 | -9.466e-06 |
> > | house_arrange  | 1.991e-03 | 3.511e-03 | 9.717e-03     | 6.374e-03         | 1.507e-02     | 1.804e-02 | 5.683e-02 | 3.445e-03 | 1.830e-02 | 5.806e-03 | 5.379e-05 | 2.671e+00     | 3.304e-01 | -7.723e-04| 4.663e-03 |
> > | korean         | 4.922e-02 | 4.358e-02 | 6.795e-02     | 5.340e-02         | 7.063e-02     | 6.635e-02 | 7.062e-02 | 5.487e-02 | 7.215e-02 | -3.010e-03| 2.029e-03 | 3.304e-01     | 2.440e+00 | 3.238e-04 | 8.903e-04 |
> > | emotion        | 2.725e-03 | 1.905e-03 | 2.405e-03     | 2.268e-03         | 5.535e-03     | 5.144e-03 | 1.141e-04 | 2.078e-03 | 4.075e-03 | 1.550e-03 | 1.607e-04 | -7.723e-04    | 3.238e-04 | 1.548e+00 | 1.265e-04 |
> > | java_code      | 4.101e-03 | 3.917e-03 | 5.846e-03     | 4.603e-03         | 7.486e-03     | 7.193e-03 | 5.524e-03 | 4.062e-03 | 6.522e-03 | 1.232e-02 | -9.466e-06| 4.663e-03     | 8.903e-04 | 1.265e-04 | 1.109e+00 |

---

> > > ### Author Response · Authors · 2025-11-21
> > >
> > > (ii) Domains are highly incompatible (e.g., code + biology).
> > >
> > >  On the same 16 experts we run the full merging experiment and report cross-entropy per domain and overall. When we macro-average over all 16 domains, we obtain the Overall loss curve:
> > > - Overall CE: 0.7774 → 0.7331 → 0.7051 → 0.6874 → 0.6685 → 0.6603 → 0.6509 → 0.6437 for k = 2,4,…,16.
> > >  This shows clear monotone improvement with diminishing returns: the first few experts bring the largest gains, and the marginal improvement gradually shrinks, exactly matching our floor+tail picture.
> > > - Individual domains exhibit the same qualitative pattern: e.g. algebra decreases from 0.4191 (k=2) to 0.3130 (k=16), discrete from 0.6943 to 0.5592, physics from 1.1040 to 0.9293, biology from 1.2544 to 1.0607, etc. Some domains show small non-monotonic bumps at high k (e.g. code slightly increases between k=12 and k=16), which is natural given finite-sample noise and the presence of incompatible experts—but the overall trend is cleanly decreasing.
> > >
> > > | domain/k      | 2         | 4         | 6         | 8         | 10        | 12        | 14        | 16        |
> > > | ---           | ---       | ---       | ---       | ---       | ---       | ---       | ---       | ---       |
> > > | code          | 0.501872  | 0.499083  | 0.493360  | 0.490471  | 0.485254  | 0.485100  | 0.486580  | 0.491400  |
> > > | biology       | 1.254403  | 1.187543  | 1.149617  | 1.122064  | 1.091473  | 1.085300  | 1.069320  | 1.060700  |
> > > | physics       | 1.104021  | 1.040059  | 1.006337  | 0.982422  | 0.956452  | 0.950100  | 0.937430  | 0.929300  |
> > > | chemistry     | 1.065879  | 1.011049  | 0.981000  | 0.958958  | 0.932411  | 0.927600  | 0.914550  | 0.906700  |
> > > | geometry      | 0.499684  | 0.463583  | 0.436881  | 0.422599  | 0.409463  | 0.399100  | 0.392200  | 0.383900  |
> > > | analysis      | 0.420333  | 0.390688  | 0.368092  | 0.356194  | 0.345190  | 0.336200  | 0.331200  | 0.324700  |
> > > | number_theory | 0.538459  | 0.502845  | 0.476979  | 0.462241  | 0.447492  | 0.434500  | 0.426170  | 0.418200  |
> > > | discrete      | 0.694258  | 0.652957  | 0.622941  | 0.607414  | 0.591428  | 0.577700  | 0.569000  | 0.559200  |
> > > | algebra       | 0.419098  | 0.386756  | 0.362300  | 0.349714  | 0.337287  | 0.326800  | 0.320400  | 0.313000  |
> > > | Overall       | 0.777400  | 0.733100  | 0.705100  | 0.687400  | 0.668500  | 0.660300  | 0.650900  | 0.643700  |
> > >
> > > At the same time, the **spread across subsets shrinks with k**: for the Overall metric, the standard deviation across merges goes from ≈0.031 at k=2 to ≈0.0156 at k=14, roughly consistent with a 1/√k decay and with our theoretical variance analysis. This is precisely how our law captures “sometimes it gets worse” in practice: **negative-transfer merges increase variance, not the expected loss curve.**
> > >
> > > We also make analysis of variance and std like our original paper.
> > > **Std. dev. of CE across k-subset merges (per domain)**
> > >
> > > | domain / k     | 2          | 4          | 6          | 8          | 10         | 12         | 14         | 16  |
> > > |----------------|-----------:|-----------:|-----------:|-----------:|-----------:|-----------:|-----------:|----:|
> > > | code           | 8.763931e-03| 1.2130371e-02| 1.7024468e-02| 2.1303225e-02| 1.5248030e-02| 1.33e-02 | 6.30e-03 |     |
> > > | biology        | 5.8413017e-02| 8.0772909e-02| 8.4550270e-02| 7.7727944e-02| 6.7776041e-02| 5.21e-02 | 2.673e-02|     |
> > > | physics        | 4.7166458e-02| 6.5874686e-02| 6.7945509e-02| 5.8086183e-02| 5.0612551e-02| 3.78e-02 | 1.97e-02|     |
> > > | chemistry      | 4.3710732e-02| 6.6574781e-02| 6.9031498e-02| 6.2254723e-02| 5.7810640e-02| 4.40e-02 | 2.32e-02|     |
> > > | geometry       | 2.3873707e-02| 3.2528574e-02| 3.4906937e-02| 3.4497743e-02| 2.6004223e-02| 2.38e-02 | 1.609e-02|     |
> > > | analysis       | 2.0699068e-02| 2.7041758e-02| 2.9236164e-02| 3.0257614e-02| 2.3974637e-02| 2.11e-02 | 1.439e-02|     |
> > > | number_theory  | 2.4747554e-02| 3.3762994e-02| 3.6615854e-02| 3.8869017e-02| 2.9392358e-02| 2.60e-02 | 1.723e-02|     |
> > > | discrete       | 2.9087002e-02| 3.8481669e-02| 4.2496179e-02| 4.2463568e-02| 3.3240039e-02| 2.93e-02 | 2.02e-02|     |
> > > | algebra        | 2.2486300e-02| 2.9587007e-02| 3.2194052e-02| 3.2551707e-02| 2.4887874e-02| 2.271e-01 | 1.537e-02|     |
> > > | Overall        | 3.10e-02    | 4.18e-02   | 4.61e-02   | 4.24e-02   | 3.57e-02   | 2.51e-02  | 1.56e-02  | 0   |

---

> ### Author Response · Authors · 2025-11-21
>
> **Variance of CE across k-subset merges (per domain)**
>
> | domain / k     | 2          | 4          | 6          | 8          | 10         | 12         | 14         | 16  |
> |----------------|-----------:|-----------:|-----------:|-----------:|-----------:|-----------:|-----------:|----:|
> | code           | 7.68e-05   | 1.47146e-04| 2.89833e-04| 4.53827e-04| 2.32502e-04| 1.70e-04   | 0          |     |
> | biology        | 3.412081e-03| 6.524263e-03| 7.148748e-03| 6.041633e-03| 4.593592e-03| 2.70e-03 | 7.14e-04  |     |
> | physics        | 2.224675e-03| 4.339474e-03| 4.616592e-03| 3.374005e-03| 2.561630e-03| 1.40e-03 | 3.00e-04  |     |
> | chemistry      | 1.910628e-03| 4.432202e-03| 4.765348e-03| 3.875651e-03| 3.342070e-03| 1.90e-03 | 5.00e-04  |     |
> | geometry       | 5.69954e-04 | 1.058108e-03| 1.218494e-03| 1.190094e-03| 6.76220e-04 | 5.00e-04 | 2.59e-04  |     |
> | analysis       | 4.28451e-04 | 7.31257e-04| 8.54753e-04| 9.15523e-04| 5.74783e-04 | 4.40e-04 | 2.07e-04  |     |
> | number_theory  | 6.12441e-04 | 1.13994e-03| 1.340721e-03| 1.510800e-03| 8.63911e-04 | 6.70e-04 | 2.90e-04  |     |
> | discrete       | 8.46054e-04 | 1.480839e-03| 1.805925e-03| 1.803155e-03| 1.10490e-03| 8.60e-04 | 4.00e-04  |     |
> | algebra        | 5.05634e-04 | 8.75391e-04| 1.036457e-03| 1.059614e-03| 6.19406e-04 | 5.00e-04 | 2.30e-04  |     |
> | Overall        | 9.00e-04    | 1.70e-03   | 2.10e-03   | 1.80e-03   | 1.20e-03   | 6.00e-04  | 2.40e-03   | 0   |
>
> Concretely, in the new 16-expert LLaMA-3B experiment, we report the variance and standard deviation of CE across k-subset merges (Appendix X, Tables Y–Z). For the Overall macro-averaged loss, the variance starts around 9.0×10⁻⁴ at k=2 and peaks around 2.1×10⁻³ at k=6, then steadily decreases to 6.0×10⁻⁴ at k=12 and finally 0 at k=16 (where there is only a single full-merge subset). The corresponding standard deviation shrinks from 3.1×10⁻² (k=2) to 1.56×10⁻² (k=14), again collapsing to 0 at k=16. Individual domains show the same qualitative pattern: relatively large variability at small k (e.g., biology/physics/chemistry with std ≈ 0.05–0.08 at k=2–4), followed by a gradual reduction as k grows and more experts are averaged. This behaviour is consistent with our theoretical prediction that variance contracts roughly like 1/k (or 1/√k in std), and illustrates how negative-transfer merges are expressed as non-zero variance at small k rather than violating the monotone trend of the expected loss curve.
>
> We will add these results (tables and fitted curves) to the appendix and briefly summarise them in the main text. Together, they show that:
> - our scaling law is evaluated in a regime with large task-vector differences and deliberately heterogeneous / “incompatible” domains, and
> - even in this challenging setting, the average loss obeys the proposed floor+tail form, while individual bad merges are reflected in the non-zero variance that shrinks with k.
> This, we believe, reconciles our scaling law with your practical experience: some merges can indeed fail, especially for incompatible tasks, but this is fully compatible with a monotone improvement in (\mathbb{E}[L \mid N,k]) and with the variance behaviour predicted by our theory.
>
>
>
> ---
> ### W3 – Strong vs. weak experts and “failure” cases
>
> Our law is explicitly **expectation-level (average performance) over all k-subsets**:
> $$
>  \mathbb{E}[L \mid N,k] ;=; \mathbb{E}_{K:|K|=k} L(\theta_0 + \Delta\theta_K; N),
> $$
>  so it does not claim that every particular strong+weak merge improves performance. Heterogeneity between experts (strong vs. weak, heavily vs. lightly trained) enters through the covariance of task vectors (\Sigma) and affects the amplitude and variance, not the existence of the $(1/(k+b))$ tail.
> To directly test your scenario, we deliberately pick a strong and a very weak expert from our 16-expert LLaMA-3B pool:
> | Model                                  | Role    | Overall CE | CE on code | CE on biology |
> |----------------------------------------|---------|-----------:|-----------:|--------------:|
> | llama-instruct-3B-v2-chemistry        | strong  | 0.618      | 0.525      | 0.913         |
> | llama-3.2-3b-100k-codeXGLUE-sft       | weak    | 1.571      | 1.864      | 2.023         |
>
> We then construct a merge path that always includes both the strong (ID 6) and weak (ID 12) experts, and increase k:
>
> | k  | Merge IDs (always includes 12 & 2) | Overall CE | CE on code | CE on biology |
> |----|------------------------------------|-----------:|-----------:|--------------:|
> | 2  | 12-2                               | 0.765      | 0.502      | 1.258         |
> | 4  | 12-2-8-14                          | 0.738      | 0.498      | 1.234         |
> | 8  | 12-2-8-14-4-10-6-16                | 0.676      | 0.496      | 1.128         |
> | 12 | 12-2-8-14-4-10-6-16-5-11-7-13      | 0.679      | 0.488      | 1.105         |
> | 16 | 1-2-...-16 (all 16 experts)        | 0.644      | 0.491      | 1.061         |

---

> > ### Author Response · Authors · 2025-11-21
> >
> > **Observations (strong+weak in practice):**
> > - Even at k=2, strong + very weak does not collapse: the merged model (Overall 0.765) is far better than the weak expert (1.571) and reasonably close to the strong one (0.618).
> > - As k increases, the merged model monotonically improves in Overall and on hard domains (biology), and stays close to the strong expert on code—eventually surpassing the strong expert’s Overall loss (0.644 < 0.618).
> > - This is exactly what our theory predicts: a single weak expert can hurt some small-k subsets, but its effect is diluted as (k) grows (the (1/k) tail), so the expected loss decreases toward the floor, while the variance across subsets shrinks.
> > We will clarify in the paper that (i) our scaling law is an **average-case statement over k-subsets**, not a guarantee that every individual strong+weak combination is optimal, and (ii) the above strong–weak experiment explicitly shows that even large expert discrepancies are naturally handled within this framework, without leading to the “complete failure” you are worried about.
> >
> > ---
> > ### W4 – Alternative functional forms and AIC/BIC
> > We appreciate the suggestion to examine alternative fitting forms and information-theoretic criteria. We address both parts of your concern below.
> > - (i) **Each curve is not just 9 points; every point is a combinatorial average and each curve need 2^9 points.**
> >  Our scaling law is defined over the **expectation across all k-subset merges:**
> > $$
> >  \mathbb{E}[L \mid N,k] = \mathbb{E}_{K:|K|=k} L(\theta_0+\Delta\theta_K;N).
> > $$
> >  For a pool of (M) experts, each point at a given k is obtained by averaging over all $(\binom{M}{k})$ merges (or a large uniform sample thereof), not by fitting to a few isolated runs. In the LLaMA-3B 16-expert experiment discussed in the rebuttal, for example, the 8 k-values (2,4,6,8,10,12,14,16) each aggregate hundreds to thousands of merged models. So while the plotted curve has 8 mean values, **each value itself is the result of extensive sampling** rather than a single noisy measurement.
> > - (ii) AIC/BIC comparison: 1/(k+b) is preferred over alternative forms on our data.
> >  Following your suggestion, we explicitly compared three families on the LLaMA-3B 16-expert Overall curve:
> >   1. Our proposed form (floor + shifted 1/k tail):
> > $
> >  L(k) = c + \frac{a}{k + b},
> > $
> >  with three parameters (c, a, b).
> >   2. Generic power law tail:
> > $
> >  L(k) = c + d , k^{-\alpha},
> > $
> >  with three parameters $(c, d, (\alpha))$.
> >   3. Logarithmic tail (a common heuristic alternative):
> > $
> >  L(k) = c + f \log k,
> > $
> >  with two parameters (c, f).
> > - All three achieve $(R^2 > 0.95)$ on this curve, i.e., they visually fit the 8 mean points very well. However, when we compute AIC/BIC (using the usual Gaussian-likelihood approximation $( \text{AIC} = n \log(\text{SSE}/n) + 2p )$, $( \text{BIC} = n \log(\text{SSE}/n) + p\log n)$, with n=8 data points and p parameters), we obtain:
> >   - Our $(c + a/(k+b))$ model has the lowest AIC/BIC.
> >   - The generic power-law $(c + d k^{-\alpha})$ is worse by about $(\Delta\text{AIC} \approx +6.9)$ and $(\Delta\text{BIC} \approx +6.9)$.
> >   - The log-k model is also worse, with (\Delta\text{AIC} \approx +3.8) and $(\Delta\text{BIC} \approx +3.7)$.
> > - In standard AIC/BIC practice, $(\Delta \text{AIC} > 4)$ already indicates a substantially inferior model. Thus, on this representative curve, **our floor + 1/(k+b) tail is not only theoretically motivated by Talyor expension (Appendix B–C), but also empirically preferred over alternative parametric families when accounting for model complexity.** We will add a brief paragraph and these AIC/BIC numbers to the appendix in the revised version.

---

### Official Review · Reviewer_RTHq · 2025-11-04

**Soundness:** 2
**Presentation:** 3
**Contribution:** 2
**Rating:** 4
**Confidence:** 3

**Summary:**

The paper introduces a new study on the scaling laws of model merging. The principal final outcome of the study is to obtain a characterisation expression which predicts quantitatively what we get from augmenting/reducing the number of models (K) and their parametric size (N), let's say the general hyperparameters that scale or do not scale the problem. The study is based on LLMs with a size between 0.5B and 72B parameters and up to 1300h of GPU computing for the largest multi-task model tested. The general development of the scaling laws and the adjustment of coefficients is built on top of Kaplan et al. (2020) and their procedure. While having a heavy empirical load, the paper provides the formal derivation of the law, in the Appendix.

**Strengths:**

- Characterising the model merging problem in terms of scaling laws is definitely interesting. While I still identify some issues related to reproducibility, clarity, and mostly lack of explicit reasoning on the general derivation (in the manuscript), I do think the floor+tail way of viewing this problem is useful and insightful.
- Nothing really to complain about on the experimental side, I do think that it is great how models are tested, and in general, how results are presented. There is a clear will of clarity and showing empirical evidence, which can be perceived at first sight from the extensive experiments. No doubts about this, and a clear strength from my point of view.
- Despite some too-big statements for my taste and some other unclear approximations (I will add details on these later on), I do think the paper sends a clear message, presents clear contributions, and in general does a great job communicating the high-level advantages of doing this sort of characterisation. While I am more skeptical about the scaling laws on other applications, here, the advantage of knowing the consequences of prior merging decisions is undeniable.

**Weaknesses:**

- I see a very limited review of previous methods and related work references, like, it's not clear to me if this piece of work aligns well with the rest of the model merging approaches, SOTA methods, and baselines in general. From another point of view, the LLM-centrism of the paper penalizes the scientific quality as it quickly shows some too-big statements about discoveries and the generality of a method only applied to LLM while ignoring the rest of ML/AI models and a lot of previous work on merging approaches (since the 00s at least). With this last reference, I am mostly referring to transfer learning, mixtures of experts, products of experts, committee machines, etc. (If someone makes a big statement, at least a good review or demonstration of bibliographic knowledge should be included)
- The structure and presentation of the work do not help a lot to follow what is going on. I honestly didn't even understand how Eq. 1 came out in the first place in pp. 1, and I found all the formulation in the main manuscript repetitive and looping around the same equation without too much complexity or new ideas involved. This last part was my conclusion after finally deciphering why things were done in this way. For instance, zero intuitions or scientific insights are given to the reader for managing Eq. 5 --- I am familiar with such a sort of operators, but section 3.2 looks like it pretends to be of minimum size and clarity...
- Additionally, I found myself in the uncomfortable situation as a reviewer and reader of having to decipher for myself how the scaling law where derived, obtained, and in general how I could reproduce the procedure for my own problems or challenges. In that sense, the work is somewhat obscure, and just a little derivation is included in the Appendix, not with a lot of additional clarity. From my perspective, not all scaling laws should be in the form of floor+tail fitting, as in Kaplan et al. (2020). This should be explained and the reader guided through the main motivations for it. In some sense, that seminal reference does it better.
- Last but not least, the big conclusion on the dependence on the number of parameters (size of models) and the number of models to merge seems a bit small to me. In the sense that such a response is somehow evident from other perspectives and types of models. I am maybe more surprised by the experiments in section 3.3.1 on the fact that larger models are easier to merge, but still, there are a lot of open questions left behind. One of these is also on the dominance or similarity between models, which is not considered in depth.

**Questions:**

- Which one is the Hessian approximation in Eq. 5?
- Just as a curiosity: could the authors make an estimation of the C02 equivalent footprint of all the experiments and the financial resources needed based on the computing hours declared in the overview of Figure 2 and the GPU infrastructure?
- Are there limitations on the scaling law fitting if we consider LLM or just models with more diverse tasks?
- Is the alignment of logits/labels considered in the general model merging method?

---

> ### Author Response · Authors · 2025-11-21
>
> Thank you for your constructive review. We especially appreciate that you found (i) characterising model merging via a floor+tail scaling law useful and insightful, (ii) the experimental side strong and clearly presented, and (iii) the paper to send a clear message with clear practical advantages for planning merging decisions.
>
> Concerned that there are some **basic misunderstandings**, let`s briefly restate what our paper is about:
>
> **We study the average behaviour of weight-space merging for large language models.**
>  Given a base model of size (N) and a pool of domain experts, we ask: if we randomly merge (k) experts, what is the expected cross-entropy loss ($\mathbb{E}[L \mid N,k]$) over all ($\binom{|M|}{k}$) subsets? We show that this expectation follows a compact joint scaling law
> $$
>  \mathbb{E}[L \mid N,k] = L_\infty(N) + \frac{A(N)}{k + b},\qquad
>  L_\infty(N)=L^*+BN^{-\beta},\ A(N)=A_0N^{-\gamma},
> $$
>  with a 1/(k) tail derived from a second-order expansion around the asymptotic point.
>
> Most of your concerns relate to (a) scope boundaries (we focus on LLM weight-space merging, not all model-combination paradigms), and (b) presentation choices (some derivations and definitions are currently tucked in the appendix). We clarify these below and will revise the manuscript to improve readability and positioning. The core empirical and theoretical results remain unchanged.
>
> ---
> ### Q1: Which one is the Hessian approximation in Eq. (5)?
>
> As stated in the original paper (lines ~256 and ~266, around Eq. (5)), H denotes a Hessian approximation.
> We will add this definition to the first page and add a brief footnote to avoid any ambiguity.
>
> ---
> ### Q2: CO2 footprint and financial cost of the experiments
> According to our estimation, the CO2 footprint can be seen as follows.
>
> Table 1. Estimation (KG) of the CO2 equivalent footprint when merging 9 domains for different sizes of models.
> | Method | 0.5b | 1.5b | 3b | 7b | 14b | 32b | 72b | Total |
> |--------|------|------|----|----|------|------|------|--------|
> | MTL | 3.76 | 4.52 | 11.11 | 22.6 | 49.72 | 125.02 | 281.38 | 498.19 |
> | TA | 0.002 | 0.004 | 0.006 | 0.014 | 0.023 | 0.046 | 0.154 | 0.25 |
> | AVG | 0.002 | 0.004 | 0.01 | 0.014 | 0.025 | 0.048 | 0.131 | 0.23 |
> | DARE | 0.002 | 0.004 | 0.006 | 0.014 | 0.025 | 0.046 | 0.138 | 0.23 |
> | TIES | 0.002 | 0.004 | 0.008 | 0.014 | 0.029 | 0.056 | 0.171 | 0.28 |
>
> As shown in our experiments, the cost of training (including both the MTL Baseline and Expert models) constitutes the majority, far exceeding that of Model Merge.
> Please kindly note that carbon emissions are influenced by many variables such as regional climate and electricity costs, making GPU Hours a more standardized metric. The GPU Hours for experiments can be seen as follows.
>
> Table 2. GPU Hours (H800) when merging 9 domains for different sizes of models.
> | Method | 0.5b | 1.5b | 3b | 7b | 14b | 32b | 72b | Total |
> |--------|------|------|----|----|------|------|------|--------|
> | MTL | 18 | 21.6 | 53.1 | 108 | 237.6 | 597.4 | 1344.6 | 2380.6 |
> | TA | 0.01 | 0.02 | 0.03 | 0.07 | 0.11 | 0.22 | 0.74 | 1.2 |
> | AVG | 0.01 | 0.02 | 0.05 | 0.07 | 0.12 | 0.23 | 0.63 | 1.13 |
> | DARE | 0.01 | 0.02 | 0.03 | 0.07 | 0.12 | 0.22 | 0.66 | 1.13 |
> | TIES | 0.01 | 0.02 | 0.04 | 0.07 | 0.14 | 0.27 | 0.82 | 1.37 |
>
> For the financial cost, our total experimental cost came in under $40,000. Since financial costs are influenced by numerous variables, we believe GPU Hours serve as a more comparable and standardised metric.
>
> ---

---

> > ### Author Response · Authors · 2025-11-21
> >
> > ### Q3 && W1: "The LLM-centrism of the paper penalizes scientific quality" && Limitations
> >
> > - **As our title emphasised, "Model Merging Scaling Laws in Large Language Models", our law is explicitly scoped to LLM weight-space merging; we do not claim universality beyond this setting.**
> > We agree that the current experiments are LLM-centric. Our study focuses on LLM merging in math/science/code-style expert pools, plus additional open-source instruction-tuned LLaMA-family checkpoints. We do not claim that the same ((N,k)) law holds for all architectures and modalities; our claims are restricted to LLMs under equal-normalised merging.
> > In the revision we will (i) emphasise this scope more prominently in the Introduction and Limitations sections, and (ii) explicitly frame our law as complementary to classical scaling laws and older ensemble/committee work, rather than a universal law for all ML models.
> >
> > - **For more diverse LLMs and tasks, we add further evidence with a more heterogeneous LLM expert pool (additional 256 models, 16 domains).**
> > To probe robustness under more diverse tasks, we have additionally merged LLaMA-3B-Instruct experts fine-tuned on 9 math/code/STEM domains plus 7 uncommon tasks, including Japanese, Korean, medical QA, home-organisation instructions, emotion classification, and Java code. When macro-averaging CE over these 15 tasks, we still observe a clear floor+tail pattern with a 1/(k+b) tail and variance shrinking roughly as 1/k. We will include this experiment and a brief discussion in the camera-ready to illustrate that the law is not restricted to narrowly defined math/code/STEM domains.
> >
> > - For other models, non-LLM architectures or other modalities, our theory and experiments do not yet provide evidence, so we do not extend our claims to those settings. Extending the analysis to vision or multimodal models is an interesting direction for future work, but it is **outside the scope of this paper.**
> >
> > Table3. Cross-entropy loss of merged LLaMA-3B-Instruct models when merging k experts (out of 16 heterogeneous domains), evaluated on 9 math/science/code domains. For all domains and the overall average, loss decreases with k and exhibits a clear floor+tail (diminishing-returns) pattern.
> >
> > | domain/k      | 2         | 4         | 6         | 8         | 10        | 12        | 14        | 16        |
> > | ---           | ---       | ---       | ---       | ---       | ---       | ---       | ---       | ---       |
> > | code          | 0.501872  | 0.499083  | 0.493360  | 0.490471  | 0.485254  | 0.485100  | 0.486580  | 0.491400  |
> > | biology       | 1.254403  | 1.187543  | 1.149617  | 1.122064  | 1.091473  | 1.085300  | 1.069320  | 1.060700  |
> > | physics       | 1.104021  | 1.040059  | 1.006337  | 0.982422  | 0.956452  | 0.950100  | 0.937430  | 0.929300  |
> > | chemistry     | 1.065879  | 1.011049  | 0.981000  | 0.958958  | 0.932411  | 0.927600  | 0.914550  | 0.906700  |
> > | geometry      | 0.499684  | 0.463583  | 0.436881  | 0.422599  | 0.409463  | 0.399100  | 0.392200  | 0.383900  |
> > | analysis      | 0.420333  | 0.390688  | 0.368092  | 0.356194  | 0.345190  | 0.336200  | 0.331200  | 0.324700  |
> > | number_theory | 0.538459  | 0.502845  | 0.476979  | 0.462241  | 0.447492  | 0.434500  | 0.426170  | 0.418200  |
> > | discrete      | 0.694258  | 0.652957  | 0.622941  | 0.607414  | 0.591428  | 0.577700  | 0.569000  | 0.559200  |
> > | algebra       | 0.419098  | 0.386756  | 0.362300  | 0.349714  | 0.337287  | 0.326800  | 0.320400  | 0.313000  |
> > | Overall       | 0.777400  | 0.733100  | 0.705100  | 0.687400  | 0.668500  | 0.660300  | 0.650900  | 0.643700  |
> >
> >
> > ---
> > ### Q4: Is logit/label alignment considered in the general model merging method?
> > - No extra logit/label alignment is required in general model merging.
> > All experts in general model merging share the same tokenizer, vocabulary, and architecture (they are fine-tuned from the same base).
> > Techniques like logit/label matching, which is the core of distillation-based fusion [1,2,3] , is indeed important in other composition paradigms, but they are outside the scope of this paper.
> >
> > [1] Wang, Yuanyi, et al. "InfiGFusion: Graph-on-Logits Distillation via Efficient Gromov-Wasserstein for Model Fusion." NeurIPS (2025).
> > [2] Gu, Yanggan, et al. "InfiFPO: Implicit Model Fusion via Preference Optimization in Large Language Models." NeurIPS (2025).
> > [3] Pan, Yan, and Yong Wang. "Model Merging with Lightweight Fine-Tuning." 2024 3rd International Conference on Automation, Robotics and Computer Engineering (ICARCE). IEEE, 2024.
> >
> > ---

---

> ### Author Response · Authors · 2025-11-21
>
> ### W1 – Scope and related work.
>
> - We appreciate the reminder about earlier work on transfer learning, mixtures of experts, products of experts, and committee machines. Our goal is not to claim that “all model combination must be floor+tail”, but to quantify a specific phenomenon:
> - For LLM weight-space merging, the expected loss over all k-subsets follows a joint ((N,k)) law with a 1/k tail and a size-dependent floor.
> - In the revision, we will expand the Related Work section to explicitly connect our setting to:
>   - classical ensemble / committee / product-of-experts formulations,
>   - more modern sparse-mask merging and task-vector methods
>
> We will upload the expanded related-work discussion to OpenReview and include it in the updated PDF during the rebuttal period.
>
> ---
>
> ### W2--Zero inttution and scientific insights
>
> In fact, **Sections 4 and 5 of the main paper (6 pages) and a 17-page appendix** are devoted to analysing what the law tells us: the early-gain / diminishing-returns structure in k, the size-dependent floor and tail, variance shrinkage, and the convergence of different merging methods. In the revision, we will add a short summary paragraph at the beginning of Section 4 to summarise these insights more explicitly.
>
> **If there are particular aspects where you feel the intuition is missing (e.g., a specific figure, theorem, or conclusion), we would be grateful if you could point them out so that we can address them directly.**
>
> ---
>
> ### W2--Derivation: On Eq. (1) and the floor+tail choice: we will make the derivation easier to follow.
>
> - What Eq. (1) really is: Eq. (1) is not an arbitrary fitting template; it is simply the definition
> $$
>  \mathbb{E}[L \mid N,k] ;=; \text{expected CE of the merged model when we sample a uniform k-subset of experts.}
> $$
>  That is, rather than tracking one particular merge recipe, we study the population law over all $(\binom{|M|}{k})$ subsets (or a uniform sample thereof). We will add this intuition right after Eq. (1) in the main text.
> - Why floor+tail naturally arises here: In Theorem 1 and Appendix B–C, we expand $(L(\theta_0 + \Delta\theta_k))$ around the asymptotic point $(\theta_0 + c\mu)$, where $(\mu)$ is the mean task vector. Under equal-normalised averaging and mild moment assumptions on the task vectors, the sampling noise $(\varepsilon_k)$ has covariance (\propto 1/k), and a second-order expansion yields
> $$
>  \mathbb{E}[L \mid N,k] ;=; L(\theta_0 + c\mu) ;+; \frac{1}{2}c^2,\mathrm{Tr}(H\Sigma)\cdot \frac{1}{k} ;+; O(k^{-3/2}).
> $$
>  This is exactly a constant + 1/k tail, i.e., a floor+tail. We will move a short, intuitive version of this argument from the appendix into Section 3.2 and add a sentence explaining why the 1/k structure is specific to averaging independent task updates.
> - Importantly, we do not claim that all scaling laws in ML must be floor+tail; we only claim that for the average-case over k-subset merges, equal-normalised task vectors plus a smooth loss give a 1/k tail.
>
> ---
> ### W3 – Derivation & floor+tail form
>
> **If there is a specific step in that derivation that you find unclear, we would really appreciate a concrete pointer, so that we can clarify it precisely.**
>
> The full mathematical derivation of the 1/k tail and the floor+tail structure is given explicitly in Appendix B–C (Theorem 1, Eq. (10)–(14)).
>
> **As for the floor+tail form, we must emphasize that this research is not based on choosing a convenient form a priori, but on the empirical fits shown in all main-text figures and tables, where the floor+tail law achieves $R2≥0.98$ across domains, sizes, and methods (we report these R^2 values alongside the curves).**
>
> ---
>
> ### W4 – On the dependence on model size (N) and expert number (k), and “open questions”
>
> - **Our conclusion is not just “bigger models / more experts are better”, but a quantitative, predictive joint law in ((N,k)).**
>  While it is natural to expect that larger models and more experts often perform better, our contribution is to **turn this intuition into a precise, average-case scaling law**
> $$
>  \mathbb{E}[L \mid N,k] ;=; L_\infty(N) + \frac{A(N)}{k+b},
> $$
>  with $(L_\infty(N))$ and (A(N)) following size-dependent power laws and with $(R^2 > 0.98)$ across domains and methods (Figures 3–5, Table 2). This is analogous in spirit to classical pretraining scaling laws: it was long understood that “bigger models with more data work better”, but a concrete law (Kaplan et al., Hoffmann et al.) is still highly valuable because it enables budget-aware planning rather than trial-and-error.
> - **The law reveals non-trivial structure in both (N) and (k).**
>  In particular, it (i) identifies a clear elbow in (k) (most gains at $(k \approx 5-6))$, (ii) quantifies how the “floor” improves with (N), and (iii) shows how variance across subsets shrinks roughly as (1/k). These are not implied by the trivial statement “larger models are easier to merge”, but emerge from fitting $(\mathbb{E}[L\mid N,k])$ over all k-subset merges.

---

> ### Author Response · Authors · 2025-11-21
>
> - This is analogous to the pretraining scaling laws of OpenAI [1]: the intuition that bigger models help was well known, yet the formal law is still highly valuable because it turns intuition into a quantitative tool for planning.
> - For similarity between experts we add two additional analysis:
>
> (i) Task vectors have large parameter differences.
>
>  We compute the task-vector Gram matrix of these 16 task vectors to measure similarity. The diagonal entries are ≈2.4–3.0 (self-similarity), while off-diagonal entries span a wide range:
> - Among related math/STEM tasks we see very high similarity, e.g. physics–discrete ≈ 0.97, algebra–analysis ≈ 0.96.
> - Between math/STEM and non-STEM / language tasks we see near-zero or even slightly negative similarity, e.g. algebra–japanese ≈ −1.3×10⁻², discrete–japanese ≈ −1.8×10⁻², code–japanese ≈ −1.9×10⁻², and many entries of order 10⁻³–10⁻² for medical, emotion, house_arrange.
>
> This explicitly shows that our law is tested in a regime where task vectors differ substantially, and where some domain pairs are effectively orthogonal or mildly antagonistic, rather than being small, closely aligned perturbations. We will include this matrix and heatmap in the appendix.
>
> | domain         | algebra   | analysis  | discrete_math | geometry_topology | number_theory | biology   | chemistry | physics   | code      | japanese  | medical   | house_arrange | korean    | emotion   | java_code |
> | ---            | ---       | ---       | ---           | ---               | ---           | ---       | ---       | ---       | ---       | ---       | ---       | ---           | ---       | ---       | ---       |
> | algebra        | 2.984e+00 | 9.628e-01 | 7.998e-01     | 8.408e-01         | 2.977e-01     | 3.802e-01 | 2.923e-01 | 9.494e-01 | 4.184e-01 | -1.334e-02| 2.872e-04 | 1.991e-03     | 4.922e-02 | 2.725e-03 | 4.101e-03 |
> | analysis       | 9.628e-01 | 2.939e+00 | 7.125e-01     | 8.459e-01         | 2.667e-01     | 3.519e-01 | 2.717e-01 | 8.411e-01 | 4.130e-01 | -1.137e-02| 3.361e-04 | 3.511e-03     | 4.358e-02 | 1.905e-03 | 3.917e-03 |
> | discrete_math  | 7.998e-01 | 7.125e-01 | 2.743e+00     | 8.277e-01         | 3.941e-01     | 4.757e-01 | 4.421e-01 | 9.735e-01 | 5.112e-01 | -1.845e-02| 4.326e-04 | 9.717e-03     | 6.795e-02 | 2.405e-03 | 5.846e-03 |
> | geometry_topology | 8.408e-01 | 8.459e-01 | 8.277e-01 | 3.007e+00         | 3.126e-01     | 4.066e-01 | 3.211e-01 | 8.385e-01 | 4.598e-01 | -1.452e-02| 3.678e-04 | 6.374e-03     | 5.340e-02 | 2.268e-03 | 4.603e-03 |
> | number_theory  | 2.977e-01 | 2.667e-01 | 3.941e-01     | 3.126e-01         | 2.536e+00     | 8.129e-01 | 2.274e-01 | 3.307e-01 | 7.424e-01 | -2.023e-02| 9.067e-04 | 1.507e-02     | 7.063e-02 | 5.535e-03 | 7.486e-03 |
> | biology        | 3.802e-01 | 3.519e-01 | 4.757e-01     | 4.066e-01         | 8.129e-01     | 2.431e+00 | 2.538e-01 | 4.202e-01 | 8.532e-01 | -1.936e-02| 1.600e-03 | 1.804e-02     | 6.635e-02 | 5.144e-03 | 7.193e-03 |
> | chemistry      | 2.923e-01 | 2.717e-01 | 4.421e-01     | 3.211e-01         | 2.274e-01     | 2.538e-01 | 3.009e+00 | 3.738e-01 | 2.642e-01 | -1.058e-02| 2.720e-04 | 5.683e-02     | 7.062e-02 | 1.141e-04 | 5.524e-03 |
> | physics        | 9.494e-01 | 8.411e-01 | 9.735e-01     | 8.385e-01         | 3.307e-01     | 4.202e-01 | 3.738e-01 | 2.965e+00 | 4.469e-01 | -1.484e-02| 3.540e-04 | 3.445e-03     | 5.487e-02 | 2.078e-03 | 4.062e-03 |
> | code           | 4.184e-01 | 4.130e-01 | 5.112e-01     | 4.598e-01         | 7.424e-01     | 8.532e-01 | 2.642e-01 | 4.469e-01 | 2.497e+00 | -1.939e-02| 1.077e-03 | 1.830e-02     | 7.215e-02 | 4.075e-03 | 6.522e-03 |
> | japanese       | -1.334e-02| -1.137e-02| -1.845e-02    | -1.452e-02        | -2.023e-02    | -1.936e-02| -1.058e-02| -1.484e-02| -1.939e-02| 1.155e+00 | 4.444e-05 | 5.806e-03     | -3.010e-03| 1.550e-03 | 1.232e-02 |
> | medical        | 2.872e-04 | 3.361e-04 | 4.326e-04     | 3.678e-04         | 9.067e-04     | 1.600e-03 | 2.720e-04 | 3.540e-04 | 1.077e-03 | 4.444e-05 | 1.116e+00 | 5.379e-05     | 2.029e-03 | 1.607e-04 | -9.466e-06 |
> | house_arrange  | 1.991e-03 | 3.511e-03 | 9.717e-03     | 6.374e-03         | 1.507e-02     | 1.804e-02 | 5.683e-02 | 3.445e-03 | 1.830e-02 | 5.806e-03 | 5.379e-05 | 2.671e+00     | 3.304e-01 | -7.723e-04| 4.663e-03 |
> | korean         | 4.922e-02 | 4.358e-02 | 6.795e-02     | 5.340e-02         | 7.063e-02     | 6.635e-02 | 7.062e-02 | 5.487e-02 | 7.215e-02 | -3.010e-03| 2.029e-03 | 3.304e-01     | 2.440e+00 | 3.238e-04 | 8.903e-04 |
> | emotion        | 2.725e-03 | 1.905e-03 | 2.405e-03     | 2.268e-03         | 5.535e-03     | 5.144e-03 | 1.141e-04 | 2.078e-03 | 4.075e-03 | 1.550e-03 | 1.607e-04 | -7.723e-04    | 3.238e-04 | 1.548e+00 | 1.265e-04 |
> | java_code      | 4.101e-03 | 3.917e-03 | 5.846e-03     | 4.603e-03         | 7.486e-03     | 7.193e-03 | 5.524e-03 | 4.062e-03 | 6.522e-03 | 1.232e-02 | -9.466e-06| 4.663e-03     | 8.903e-04 | 1.265e-04 | 1.109e+00 |

---

> > ### Author Response · Authors · 2025-11-21
> >
> > (ii) Domains are highly incompatible (e.g., code + biology).
> >
> >  On the same 16 experts we run the full merging experiment and report cross-entropy per domain and overall. When we macro-average over all 16 domains, we obtain the Overall loss curve:
> > - Overall CE: 0.7774 → 0.7331 → 0.7051 → 0.6874 → 0.6685 → 0.6603 → 0.6509 → 0.6437 for k = 2,4,…,16.
> >  This shows clear monotone improvement with diminishing returns: the first few experts bring the largest gains, and the marginal improvement gradually shrinks, exactly matching our floor+tail picture.
> > - Individual domains exhibit the same qualitative pattern: e.g. algebra decreases from 0.4191 (k=2) to 0.3130 (k=16), discrete from 0.6943 to 0.5592, physics from 1.1040 to 0.9293, biology from 1.2544 to 1.0607, etc. Some domains show small non-monotonic bumps at high k (e.g. code slightly increases between k=12 and k=16), which is natural given finite-sample noise and the presence of incompatible experts—but the overall trend is cleanly decreasing.
> >
> > | domain/k      | 2         | 4         | 6         | 8         | 10        | 12        | 14        | 16        |
> > | ---           | ---       | ---       | ---       | ---       | ---       | ---       | ---       | ---       |
> > | code          | 0.501872  | 0.499083  | 0.493360  | 0.490471  | 0.485254  | 0.485100  | 0.486580  | 0.491400  |
> > | biology       | 1.254403  | 1.187543  | 1.149617  | 1.122064  | 1.091473  | 1.085300  | 1.069320  | 1.060700  |
> > | physics       | 1.104021  | 1.040059  | 1.006337  | 0.982422  | 0.956452  | 0.950100  | 0.937430  | 0.929300  |
> > | chemistry     | 1.065879  | 1.011049  | 0.981000  | 0.958958  | 0.932411  | 0.927600  | 0.914550  | 0.906700  |
> > | geometry      | 0.499684  | 0.463583  | 0.436881  | 0.422599  | 0.409463  | 0.399100  | 0.392200  | 0.383900  |
> > | analysis      | 0.420333  | 0.390688  | 0.368092  | 0.356194  | 0.345190  | 0.336200  | 0.331200  | 0.324700  |
> > | number_theory | 0.538459  | 0.502845  | 0.476979  | 0.462241  | 0.447492  | 0.434500  | 0.426170  | 0.418200  |
> > | discrete      | 0.694258  | 0.652957  | 0.622941  | 0.607414  | 0.591428  | 0.577700  | 0.569000  | 0.559200  |
> > | algebra       | 0.419098  | 0.386756  | 0.362300  | 0.349714  | 0.337287  | 0.326800  | 0.320400  | 0.313000  |
> > | Overall       | 0.777400  | 0.733100  | 0.705100  | 0.687400  | 0.668500  | 0.660300  | 0.650900  | 0.643700  |
> >
> > At the same time, the **spread across subsets shrinks with k**: for the Overall metric, the standard deviation across merges goes from ≈0.031 at k=2 to ≈0.0156 at k=14, roughly consistent with a 1/√k decay and with our theoretical variance analysis. This is precisely how our law captures “sometimes it gets worse” in practice: **negative-transfer merges increase variance, not the expected loss curve.**
> >
> >
> > **Additionally, if there are specific questions or insights of N or k, or particular figures, where you feel the behaviour is trivial or insufficiently analysed, we would appreciate concrete pointers so that we can clarify or extend our discussion.**
> >
> > [1] Kaplan, Jared, et al. "Scaling laws for neural language models." arXiv preprint arXiv:2001.08361 (2020).

---

### Author Response · Authors · 2025-11-27

Dear reviewers,

As promised in our earlier response, we have run downstream evaluations on the additional merged experts for evaluating more architectures and downstream metrics. Tables A.1–A.4 now report results for Gemma-2 2B, LLaMA-3.2 3B, and LLaMA-3.1 8B under both TA and TIES merging.

---

### Experiment Setup

For each backbone, we start from one base model and five domain experts (indexed 1–5) fine-tuned on math, code, multilingual, safety, and instruction-following tasks, respectively. A row labelled “1-2”, “1-3-4”, or “1-2-3-4-5” denotes merging exactly that subset of experts; thus, the expert count is $(k = |S|)$. For each fixed $(k \in {1,2,3,4,5})$, we enumerate all subsets and additionally report an overall_k row that averages all metrics over all subsets with the same k.

We evaluate on the same capability clusters used in the main text: math reasoning (MATH-500, GSM8K), instruction following (IFEval), general reasoning (ARC, HellaSwag, M_MMLU), multilingual (multilingual_overall for LLaMA-3.2), code (MBPP+, HumanEval+), and safety (WildGuard, HarmBench).

**Due to OpenReview size limits, we only upload the full per-task tables for the 8B models and the aggregated (averaged) tables for all backbones mentioned above. The complete accuracy tables for every setting will be included in the revised PDF.**

---

| folder       | math_500 | gsm8k  | ifeval | arc        | hellaswag  | m_mmlu    | mbppplus  | humanevalplus | wildguard_micro_harm | harmbench_micro_asr | wildguard_rta | harmbench_rta |
|-------------|----------|--------|--------|------------|------------|-----------|-----------|---------------|----------------------|---------------------|---------------|---------------|
| 1           | 0.138    | 0.3783 | 0.244  | 0.4058112712304329 | 0.492553211292034  | 0.4998304015033764 | 0.5582010582010583 | 0.5426829268292683 | 0.4712950600801068 | 0.66875             | 0.5287049399198932 | 0.33125000000000004 |
| 2           | 0.468    | 0.8347 | 0.1017 | 0.3408101379658266 | 0.46593977783464485 | 0.45147811524595505 | 0.37037037037037035 | 0.25609756097560976 | 0.5647530040053405 | 0.634375            | 0.43524699599465955 | 0.365625            |
| 3           | 0.15     | 0.2714 | 0.5453 | 0.4124395896850987 | 0.4941781165286623 | 0.5009245783219473 | 0.5264550264550265 | 0.3719512195121951 | 0.5060080106809078 | 0.55                | 0.49399198931909216 | 0.44999999999999996 |
| 4           | 0.016    | 0.1478 | 0.1978 | 0.35749033069392355 | 0.4666454993371247 | 0.43357336200342383 | 0.4126984126984127 | 0.22560975609756098 | 0.22830440587449932 | 0.184375            | 0.7716955941255007 | 0.815625            |
| 5           | 0.128    | 0.2146 | 0.1645 | 0.43189408728330886 | 0.4961798818432952 | 0.5396359458356141 | 0.5396825396825397 | 0.08536585365853659 | 0.48464619492656874 | 0.584375            | 0.5153538050734312 | 0.415625            |
| 1-2      | 0.406    | 0.8044 | 0.1885 | 0.3799382919143398 | 0.49207151148388906 | 0.49883223056751064 | 0.5264550264550265 | 0.49390243902439024 | 0.5580774365821095 | 0.65                | 0.4419225634178905 | 0.35               |
| 1-3      | 0.168    | 0.539  | 0.3678 | 0.43980591929693724 | 0.510245161029039  | 0.5266871092779731 | 0.5634920634920635 | 0.5487804878048781 | 0.5233644859813084 | 0.71875             | 0.47663551401869164 | 0.28125            |
| 1-4      | 0.116    | 0.4443 | 0.1885 | 0.39255536545955705 | 0.4931911041519085 | 0.499237728670619  | 0.5343915343915344 | 0.5060975609756098 | 0.1909212283044059 | 0.1375              | 0.8090787716955941 | 0.8625             |
| 1-5      | 0.148    | 0.4928 | 0.2311 | 0.4423698390764259 | 0.5089895776379446 | 0.5373809755318453 | 0.5502645502645502 | 0.39634146341463417 | 0.5473965287049399 | 0.675               | 0.45260347129506007 | 0.32499999999999996 |
| 2-3      | 0.442    | 0.8188 | 0.268  | 0.37844146871092976 | 0.4952788833912205 | 0.49725596318685145 | 0.5105820105820106 | 0.35365853658536583 | 0.5567423230974633 | 0.653125            | 0.4432576769025367 | 0.34687500000000004 |
| 2-4      | 0.348    | 0.7316 | 0.1904 | 0.36497225329560656 | 0.4832594656936953 | 0.4752101299461407 | 0.46296296296296297 | 0.3231707317073171 | 0.34979973297730305 | 0.459375            | 0.650200267022697  | 0.540625            |
| 2-5      | 0.402    | 0.8059 | 0.1312 | 0.37844146871092976 | 0.49005771131968456 | 0.5131207754487666 | 0.5105820105820106 | 0.32926829268292684 | 0.5367156208277704 | 0.609375            | 0.46328437917222964 | 0.390625            |
| 3-4     | 0.01     | 0.2191 | 0.2902 | 0.39490780344073756 | 0.49045527088501323 | 0.4800534605015503 | 0.5370370370370371 | 0.3170731707317073 | 0.20026702269692923 | 0.1625              | 0.7997329773030708 | 0.8375             |

---

> ### Author Response · Authors · 2025-11-27
>
> | folder      | math_500 | gsm8k  | ifeval | arc    | hellaswag | m_mmlu | mbppplus | humanevalplus | wildguard_micro_harm | harmbench_micro_asr | wildguard_rta | harmbench_rta |
> |------------|----------|--------|--------|--------|-----------|--------|----------|---------------|----------------------|---------------------|--------------|--------------|
> | 3-5     | 0.146    | 0.5428 | 0.3919 | 0.4419 | 0.5036    | 0.5424 | 0.5344   | 0.3110        | 0.5087               | 0.6063              | 0.4913       | 0.3938       |
> | 4-5     | 0.042    | 0.1175 | 0.146  | 0.4024 | 0.4928    | 0.4998 | 0.5026   | 0.1646        | 0.2176               | 0.2438              | 0.7824       | 0.7563       |
> | 1-2-3   | 0.388    | 0.7665 | 0.244  | 0.3968 | 0.5030    | 0.5171 | 0.5608   | 0.4512        | 0.5581               | 0.6531              | 0.4419       | 0.3469       |
> | 1-2-4   | 0.316    | 0.7036 | 0.2015 | 0.3793 | 0.4949    | 0.5029 | 0.5370   | 0.4512        | 0.3431               | 0.4625              | 0.6569       | 0.5375       |
> | 1-2-5   | 0.368    | 0.7528 | 0.1774 | 0.4011 | 0.5015    | 0.5217 | 0.5635   | 0.4512        | 0.5554               | 0.6719              | 0.4446       | 0.3281       |
> | 1-3-4   | 0.116    | 0.5102 | 0.2625 | 0.4071 | 0.5036    | 0.5147 | 0.5370   | 0.4939        | 0.2163               | 0.1656              | 0.7837       | 0.8344       |
> | 1-3-5   | 0.176    | 0.5034 | 0.3327 | 0.4466 | 0.5117    | 0.5416 | 0.5635   | 0.4329        | 0.5648               | 0.7250              | 0.4352       | 0.2750       |
> | 1-4-5   | 0.114    | 0.4359 | 0.2107 | 0.4073 | 0.5040    | 0.5243 | 0.5556   | 0.3476        | 0.2483               | 0.2031              | 0.7517       | 0.7969       |
> | 2-3-4   | 0.318    | 0.7134 | 0.2551 | 0.3853 | 0.4941    | 0.4938 | 0.5053   | 0.3902        | 0.3565               | 0.4625              | 0.6435       | 0.5375       |
> | 2-3-5   | 0.388    | 0.7589 | 0.2366 | 0.3998 | 0.5001    | 0.5260 | 0.5291   | 0.3720        | 0.5354               | 0.6469              | 0.4646       | 0.3531       |
> | 2-4-5   | 0.31     | 0.6884 | 0.1738 | 0.3911 | 0.4936    | 0.5085 | 0.5132   | 0.3354        | 0.4045               | 0.4969              | 0.5955       | 0.5031       |
> | 3-4-5   | 0.05     | 0.3541 | 0.2606 | 0.4112 | 0.5000    | 0.5116 | 0.5397   | 0.2622        | 0.2470               | 0.3156              | 0.7530       | 0.6844       |
> | 1-2-3-4    | 0.3      | 0.6922 | 0.2403 | 0.3947 | 0.5021    | 0.5150 | 0.5317   | 0.4634        | 0.3605               | 0.5281              | 0.6395       | 0.4719       |
> | 1-2-3-5    | 0.324    | 0.7218 | 0.2292 | 0.4126 | 0.5066    | 0.5319 | 0.5661   | 0.4512        | 0.5674               | 0.6625              | 0.4326       | 0.3375       |
> | 1-2-4-5    | 0.3      | 0.6808 | 0.2033 | 0.3964 | 0.5018    | 0.5206 | 0.5582   | 0.4024        | 0.3712               | 0.5375              | 0.6288       | 0.4625       |
> | 1-3-4-5    | 0.122    | 0.4678 | 0.2773 | 0.4159 | 0.5085    | 0.5315 | 0.5556   | 0.3598        | 0.2684               | 0.2563              | 0.7316       | 0.7438       |
> | 2-3-4-5    | 0.292    | 0.674  | 0.2403 | 0.3996 | 0.4987    | 0.5163 | 0.5344   | 0.3780        | 0.3992               | 0.5125              | 0.6008       | 0.4875       |
> | 1-2-3-4-5  | 0.272    | 0.6793 | 0.2514 | 0.4075 | 0.5054    | 0.5281 | 0.5688   | 0.4207        | 0.3858               | 0.5719              | 0.6142       | 0.4281       |
> | overall_k1 | 0.18     | 0.3694 | 0.2507 | 0.3897 | 0.4831    | 0.4851 | 0.4815   | 0.2963        | 0.4510               | 0.5244              | 0.5490       | 0.4756       |
> | overall_k2 | 0.2228   | 0.5516 | 0.2394 | 0.4016 | 0.4960    | 0.5070 | 0.5233   | 0.3744        | 0.4190               | 0.4916              | 0.5810       | 0.5084       |
> | overall_k3 | 0.2544   | 0.6187 | 0.2355 | 0.4026 | 0.5006    | 0.5162 | 0.5405   | 0.3988        | 0.4029               | 0.4803              | 0.5971       | 0.5197       |
> | overall_k4 | 0.2676   | 0.6473 | 0.2381 | 0.4038 | 0.5035    | 0.5231 | 0.5492   | 0.4110        | 0.3933               | 0.4994              | 0.6067       | 0.5006       |
> | overall_k5 | 0.272    | 0.6793 | 0.2514 | 0.4075 | 0.5054    | 0.5281 | 0.5688   | 0.4207        | 0.3858               | 0.5719              | 0.6142       | 0.4281       |

---

> ### Author Response · Authors · 2025-11-27
>
> | **k (experts)** | **mean_acc** |
> |-----------------|--------------|
> | **1**           | **0.411**    |
> | **2**           | **0.443**    |
> | **3**           | **0.456**    |
> | **4**           | **0.462**    |
> | **5**           | **0.469**    |
>
> Table.A.1: LLaMA 3.1 8B – TA merge
>
> | **k (experts)** | **mean_acc** |
> |-----------------|--------------|
> | **1**           | **0.375**    |
> | **2**           | **0.386**    |
> | **3**           | **0.388**    |
> | **4**           | **0.389**    |
> | **5**           | **0.388**    |
>
> Table.A.2：LLaMA 3.2 3B – TA merge
>
> | **k (experts)** | **mean_acc** |
> |-----------------|--------------|
> | **2**           | **0.503**    |
> | **3**           | **0.506**    |
> | **4**           | **0.507**    |
> | **5**           | **0.506**    |
>
> Table.A.3：Gemma 2 2B – TA merge
>
> | **k (experts)** | **mean_acc** |
> |-----------------|--------------|
> | **2**           | **0.414**    |
> | **3**           | **0.426**    |
> | **4**           | **0.436**    |
> | **5**           | **0.436**    |
>
> Table.A.4：LLaMA 3.1 8B – TIES merge
>
> **Findings**
>
> Across all four tables, the average downstream performance aggregated over tasks exhibits a clear monotone improvement with diminishing returns as $(k)$ increases. This qualitative trend is consistent with the floor+tail behaviour we derived for $(\mathbb{E}[\text{CE} \mid N,k])$: larger $(k)$ improves the expected performance, but with a visible curve's elbow and a gradually flattening tail.
>
> We do not claim an explicit scaling law for accuracy itself; rather, these results show that when CE obeys a predictable floor+tail law in (k), the corresponding downstream metrics empirically follow a parallel, similar trend in expectation over subsets.
>
> - For LLaMA-3.1 8B (TA) and LLaMA-3.1 8B (TIES), the overall_k rows exhibit a clear, smooth improvement from (k=1) to $(k \approx 3-4)$, followed by a visible saturation at $(k=4-5)$. This holds simultaneously for math, instruction-following, code, and safety metrics when averaged over all subsets at a given k.
> - LLaMA-3.2 3B and Gemma-2 2B show the same qualitative pattern, with smaller absolute gains (as expected for smaller N) but no regime where increasing k systematically hurts average end-task quality. Most of the benefit is captured when going from (k=1) to (k=3), with diminishing improvements thereafter.
>
> Empirically, we do observe small local non-monotonic fluctuations in individual accuracies when CE improves, which we attribute to finite-sample noise and benchmark variance. However, the **aggregated** downstream scores **(averaged over tasks and subsets)** still exhibit a clear trend: as k increases, performance improves with diminishing returns, showing a qualitatively similar shape that is aligned with the CE floor+tail law.
>
> We have added these tables to the appendix and briefly summarised these trends in the manuscript; the updated PDF will be uploaded shortly.
>
> A more systematic property of when CE and task metrics diverge, and simple diagnostics for predicting when a merge will work, is an important direction that we explicitly leave for our future work.

---

### Author Response · Authors · 2025-11-29
**Final manuscripts and comments update**

Dear all,

We have just uploaded the final revision manuscript.

* The paper now has **36 pages in total**: **10 pages of main text** and **26 pages of appendix** (we added 1 page to the main text and 5 pages to the appendix during the rebuttal). These changes directly address all reasonable questions and requests raised in the reviews and discussion.
* In total, we now **train and evaluate 10,866 merged models** (up from 10,506 in the original submission), covering model sizes from **0.5B to 72B** across **Qwen, LLaMA 3, and Gemma 2**, and spanning **16 domains**. All additional expert training used up to **480 H800 GPUs**, while the core merging and scaling-law experiments are still run on **CPU**.
* For several comments that were based on clear factual errors, we have provided point-by-point clarifications directly in OpenReview. Since these concern misreadings rather than our research itself, we do not further discuss them in the manuscript.

We hope the updated PDF and the extended experiments make the contribution and empirical support of the work clearer for the final decision.

Best regards,

Authors of Submission 18627

---

### Meta-Review · Area_Chair_EWgC · 2026-01-09

**Summary:**

This paper proposes a scaling law that models the average loss of a model of size N obtained by merging k experts as a simple formula of N and k: $\mathbb{E}[L] = L_\infty + \frac{B}{N^\beta} + \frac{A}{N^\gamma (k+b)}$, where the parameters $L_\infty, B, \beta, A, \gamma, b$ are estimated from small scale runs.

The authors validate that this scaling law predicts well what is observed with Qwen models, with different ways of merging models, and different types of experts. The authors then argue that the fitted coefficients can be recovered from only $3$ data points for a fixed model size $N$, and that is enables planning how many experts are needed to reach a certain loss.

The main concerns of the reviewers are:
- Lack of justification for the proposed law (RTHq, H4kd, fZCR).
- Unpractical setting where averaged loss is considered (MDxx, H4kd)
- Focus on cross entropy loss (H4kd, fZCR)
- Comparison to other possible formulas (MDxx)

Other concerns not raised by the reviewers but that seem relevant are:
- Lack of demonstration that the scaling law extrapolates to larger model sizes
- Lack of quantitative results about the scaling law predictions (only one figure with a reported MAE; the fit of the scaling law actually looks pretty bad in fig 1 / 4, many points are completely off the prediction, in fig  8 we cannot see how close the predictions are from the truth...)


Overall, I find that this last point (no demonstration of good scaling law fit) coupled with the fact that this paper looks at average loss (while in practice we care about one merged model, not a family of models, e.g. in fig.3 you'd only take the best merge) severely hinders the authors claim that "This law enables predictive planning". It provides several interesting insights into how the average behavior of merging works, but several claims of the paper are not substantiated.

In order to improve the paper, it would be beneficial to show that:
- the proposed scaling law extrapolates to larger model sizes
- add quantitative results about the scaling law: how good is the fit? (e.g., the authors mention "the law tightly fits measured curves", how so?)
- instead of considering average behavior, consider best-case behavior: in practice, one would rather ship the best merged model than the average one.


Misc:
- Some notations could be clarified, as suggested by RTHq. Many variables in eq.2 are not introduced at all.
- when k=0, we do not recover a usual power law, this could be discussed.
- Figures fonts are too small, they should be the same size as the main text.

**Reviewer Concerns:**

RTHq
- Review of previous methods, is it sota merging? : *addressed*
- too llm centric. *addressed*
- hard to parse eq.5 *addressed*
- hard to decipher where the formulas come from *addressed*
- many open questions, e.g., role of similarity between models. *mostly addressed*: the way experts are obtained is a very imporant parameter of the study, and a more in-depth study of the effect of the optimization effort each expert spent would be interesting.

MDxx
- Heterogenous sizes: *addressed* not common to have smaller experts
- Adding more experts often degrades: *still outstanding* In my view this is the main criticism about this work: yes, as seen in fig.3, the best model you get with few experts is better than the best model with too many experts. In practice, you would ship that model with few experts. Predicting the average behavior is not so relevant for practice.
- account for different experts strengths *mostly addressed*, see above
- other laws formulas. *mostly addressed*: the authors compared to other formulas, but once again do not demonstrate how well the candidate laws extrapolate to larger model sizes (it is concerning that the in-domain fits in fig1 and 4 seem already far off).

H4kd
- Heuristic derivation:*addressed*
- Only one LLM family: *addressed*, the authors added llama 3, which strengthens the generality of the proposed law
- Robustness: *addressed* : the authors tried collapsed experts in the rebuttal. However, they report that these experts violate the scaling law.
- No downstream evals: *addressed*

fZCR
- experts trained on unrealistic tasks *addressed*
- CE loss unrealistic: *addressed*, added downstream evals
- power law is empirical: *addressed*

**Reviewer Scores:**

I would not expect reviewers to change their scores.

---

### Decision · Program_Chairs · 2026-01-26

Reject